

# FarmConners Wind Farm Flow Control Benchmark: Blind Test Results

Tuhfe Göçmen[1], Filippo Campagnolo[2], Thomas Duc[3], Irene Eguinoa[6], Søren Juhl Andersen[1], Vlaho Petrović[4], Lejla Imširović[2], Robert Braunbehrens[2], Ju Feng[1], Jaime Liew[1], Mads Baungaard[1], Maarten Paul van der Laan[1], Guowei Qian[5], Maria Aparicio-Sanchez[6], Rubén González-Lope[6], Vinit V. Dighe[7], Marcus Becker[7], Maarten van den Broek[7], Jan-Willem van Wingerden[7], Adam Stock[8], Matthew Cole[8], Renzo Ruisi[9], Ervin Bossanyi[9], Niklas Requate[10], Simon Strnad[10], Jonas Schmidt[10], Lukas Vollmer[10], Frédéric Blondel[11], Ishaan Sood[12], and Johan Meyers[12]

[1]DTU Wind Energy, Technical University of Denmark, Lyngby/Roskilde, Denmark
[2]Technical University Munich, Munich, Germany
[3]ENGIE Green, Montpellier, France
[4]ForWind, University of Oldenburg, Oldenburg, Germany
[5]The University of Tokyo, Tokyo, Japan
[6]CENER (Centro Nacional de Energías Renovables), Sarriguren, Spain
[7]Delft Center for Systems and Control, Delft University of Technology, Delft, The Netherlands
[8]University of Strathclyde, Glasgow, Scotland
[9]DNV, Group Research & Development, Bristol, United Kingdom
[10]Fraunhofer IWES, Bremerhaven, Germany
[11]IFP Energies nouvelles (IFPEN), Rueil-Malmaison, France
[12]Mechanical Engineering, KU Leuven, Celestijnenlaan 300, Leuven 3001, Belgium

**Correspondence:** Tuhfe Göçmen (tuhf@dtu.dk)

**Abstract.**

Wind farm flow control (WFFC) is a topic of interest at several research institutes, industry and certification agencies world-wide. For reliable performance assessment of the technology, the efficiency and the capability of the models applied to WFFC should be carefully evaluated. To address that, FarmConners consortium has launched a common benchmark for code

comparison under controlled operation to demonstrate its potential benefits such as increased power production. The benchmark builds on available data sets from previous field campaigns, wind tunnel experiments and high-fidelity simulations. Within that database, 4 blind tests are defined and 13 participants in total have submitted results for the analysis of single and multiple wake under WFFC. Some participants took part in several blind tests and some participants have implemented several models. The observations and/or the model outcomes are evaluated via direct power comparisons at the upstream and downstream

turbine(s), as well as the power gain at the wind farm level under wake steering control strategy. Additionally, wake loss reduction is also analysed to support the power performance comparison, where relevant. Majority of the participating models show good agreement with the observations or the reference high-fidelity simulations, especially for lower degrees of upstream misalignment and narrow wake sector. However, the benchmark clearly highlights the importance of the calibration procedure for control-oriented models. The potential effects of limited controlled operation data in calibration is particularly visible via

frequent model mismatch for highly deflected wakes, as well as the power loss at the controlled turbine(s). In addition to





the flow modelling, sensitivity of the predicted WFFC benefits to the turbine representation and the implementation of the controller is also underlined. FarmConners benchmark is the first of its kind to bring a wide variety of data sets, control settings and model complexities for the (initial) assessment of farm flow control benefits. It forms an important basis for more detailed benchmarks in the future with extended control objectives to assess the *true* value of WFFC.

## 1  Introduction

Wind farm flow control (WFFC) promises to mitigate the losses due to aerodynamic turbine-turbine interactions and can potentially provide several benefits to reduce the cost of energy in the design and operation of wind farms. Its most prominent benefits are the potential increase in power production and/or alleviation of turbine structural loading at wind farms by reducing wake losses and encouraging energy entrainment into the farm. The phenomenon has been thoroughly investigated, with lower

order and high-fidelity flow and structural response models (*e.g.* Gebraad et al., 2016; Munters and Meyers, 2018; Duc et al., 2019; Hulsman et al., 2020); wind tunnel tests (*e.g.* Rockel et al., 2017; Bastankhah and Porté-Agel, 2019; Campagnolo et al., 2020; Bottasso and Campagnolo, 2021); and field experiments (*e.g.* Fleming et al., 2017; Annoni et al., 2018; Doekemeijer et al., 2021; Bossanyi and Ruisi, 2021; Simley et al., 2021). A comprehensive review of the power maximisation through WFFC is presented in Kheirabadi and Nagamune (2019) and Andersson et al. (2021). To realise those benefits, the control

strategy might be 1) Axial induction control, in which some upstream turbines will lower their energy capture (also referred as curtailment, down-regulation or derating) hence increasing the wind velocity and reducing the turbulence downstream; and/or 2) Wake steering, in which some of the turbines will be misaligned to redirect the wake away from the other turbines hence mitigating the wake effects; and 3) Wake mixing where upstream turbines are dynamically up-regulated and down-regulated on short time scales to induce additional wake mixing and wake recovery, minimising the losses further downstream. A number

of control-oriented models with different levels of complexity have been proposed in literature to implement those control strategies but uncertainty remains high and a systematic validation and comparison under different control settings have been lacking.

In order to assess the performance of the WFFC technology, the capabilities of WFFC models should be evaluated. Accordingly, FarmConners consortium (FarmConners, 2019) has launched a common benchmark for code comparison; where high-

fidelity simulation results, wind tunnel experiments and field data measured at a full-scale wind farm are brought together. This unique database combines the efforts of the connected WFFC projects of different sizes all over Europe and consists of 4 blind tests in total: 1) SMARTEOLE Sole du Moulin Vieux (SMV) field measurement campaign (Duc et al., 2019), 2) CL-Windcon Wind Tunnel Experiments (CL-Windcon, 2016), 3) CL-Windcon large eddy simulations (LES) (CL-Windcon, 2016) and 4) TotalControl LES (TotalControl, 2018). Every data set is divided into a 'calibration' and a 'blind test' period to resemble field

application of WFFC models. The 'calibration' period involves both input and output features which can be used to calibrate the participating models under normal operation and limited control set-points. In the 'blind test' period, the calibrated models are to be run 'blindly' where only the input features are provided and their outputs are compared against the validation data set or the blind test reference, as well as each other.





Also promoting data sharing and standardisation for validation processes, one of the most relevant exercises with similar
structure is Wakebench (Rodrigo et al., 2014; Moriarty et al., 2014), focusing on wind farm flow modelling under normal
operation. Although sharing similar goals, the FarmConners benchmark blind tests conduct the performance evaluation exclu-
sively under controlled operation. Therefore, the two benchmarks diverge in terms of test cases, quantities of interests and the
validation metrics. However, the lessons learned from the extensive Wakebench experience (Doubrawa et al., 2020) is atten-
tively taken into consideration while preparing the framework, aiming to extend the standard verification and validation (V&V)
practices to include WFFC in wake research, globally.

The FarmConners benchmark is launched in TORQUE 2020 (Göçmen et al., 2020a) and in the end, 13 participants have
submitted the results from their models, taking part in different blind tests. The overview of participants among the benchmark
blind tests is presented in Table 1 and it should be noted that some participants have provided results from several models.
The primary quantities of interest used in the model evaluations are the direct power comparisons at upstream and downstream
turbines, as well as the power gain at the wind farm level under wake steering control strategy. Additional analysis on the
wake loss reduction has also been presented to support the evaluation of the power performance, where relevant. It should be
noted that the potential of the structural load alleviation, which was originally a quantity of interest in the benchmark, has been
excluded from this study. It is not only due to the limited number of participating models capable of providing the necessary
channels, but also to keep the focus on the most important benefit of the WFFC technology, as reported in the expert elicitation
survey (van Wingerden et al., 2020). Similarly, scenarios with axial induction control that are included in the majority of the
blind tests originally are also omitted due to limited participating model results. All the data collected for the benchmark can
potentially be made available upon request for researchers, see Data availability section for details. The notebooks, including
data snippets, where the blind tests results are produced are also publicly available, see Code availability section for details.

|   | Single Wake | | | | Multiple Wake | | | |
|---|---|---|---|---|---|---|---|---|
|   | SMARTEOLE SMV Wind Farm | CL-Windcon Wind Tunnel Experiments | CL-Windcon LES | TotalControl LES | SMARTEOLE SMV Wind Farm | CL-Windcon Wind Tunnel Experiments | CL-Windcon LES | TotalControl LES |
| P3 |   | X |   |   |   | X |   |   |
| P4 | X |   |   |   | X |   |   |   |
| P5 | X |   |   |   | X |   |   |   |
| P6 | X |   |   |   | X |   |   |   |
| P8 |   |   |   | X |   |   |   | X |
| P10 |   |   |   | X |   |   |   | X |
| P11 |   |   | X | X |   |   | X | X |
| P12 |   |   | X |   |   |   | X |   |
| P16 | X |   | X | X | X |   | X | X |
| P17 | X |   |   |   | X |   |   |   |
| P18 |   | X |   |   |   | X |   |   |
| P19 |   |   | X |   |   |   | X |   |
| P20 |   |   |   | X |   |   |   | X |

**Table 1.** Overview of the FarmConners Benchmark participants (Participant IDs P3 – P20) per test case, per blind test. WFFC-oriented model comparison is performed separately for each column. The participants are colour coded for easy access through the article.





The results of the benchmark are listed per technology readiness level (TRL) of the validation or code comparison exercise. Accordingly, Section 2 describes the SMV field campaign and presents the participating model results for single and multiple wake scenarios under upstream wake steering. Section 3 illustrates the CL-Windcon wind tunnel campaign with 3 scaled turbines under various misalignment set-points and compares the participating model results. High-fidelity simulations being a key enabler for the industrial implementation of WFFC, the subset of the extensive CL-Windcon LES database used for the FarmConners benchmark is detailed in Section 4. The participating models for the CL-Windcon LES blind tests are then evaluated for 3 and 9-turbine wind farm configurations, with 5 and 7-diameters (D) spacing, respectively. Similarly, Section 5 utilises up to 8-turbine subsets of 32-turbine layout of reference wind farm developed under TotalControl project and compares the participating model results with the reference LES database under upstream wake steering.

Here in this article, we present the results from all the blind tests together to facilitate cross-comparison of the model performances and validation/reference data set characteristics. The resulting extensive analysis, however, is broken down into blind test specific, relatively stand-alone sections with limited cross-references. Therefore, individual sections can also be read separately if preferred.

## 2 BLIND TEST #1: SMV WIND FARM FIELD DATA

The wind farm field data comes from Sole du Moulin Vieux (SMV) wind farm, located in the northern part of France (approximately midway between Paris and Lille) and operated by ENGIE Green. It consists of 7 Senvion MM82 wind turbines (diameter of 82 m, nominal power of 2050 kW, hub height of 80 m, see (Duc et al., 2019) for power and thrust coefficient, $C_T$ curves), organised in an irregular single row layout and labelled SMV1 to SMV7 from North to South. This wind farm has been used for the field tests of the French national project SMARTEOLE, whose results have been presented in Ahmad et al. (2017) and Duc et al. (2019) for the field campaign #1 and in Simley et al. (2021) for the field campaign #3. The layout of the wind farm is shown on Figure 1 and the long-term wind rose observed at the site presented in Figure 2.

The data set used for this benchmark exercise corresponds to the experiments carried out during field campaign #2, between May 2017 and March 2018. A ground-based lidar Windcube v2 was installed specifically for these field tests, its location close to SMV6 is displayed on Figure 1.

Between August 12th 2017 and October 3rd 2017, SMV6 wind turbine was constantly misaligned for all wind directions. The averaged value of the turbine yaw offset during this 7 weeks period was -13.3° (turbine rotated counter-clockwise when viewed from above), as illustrated in Figure 10 below. This misalignment creates a wake steering that affects the downstream turbines, mostly SMV5 for wind directions between 200° and 215° and more slightly SMV4 to SMV1 for wind directions below 200°.





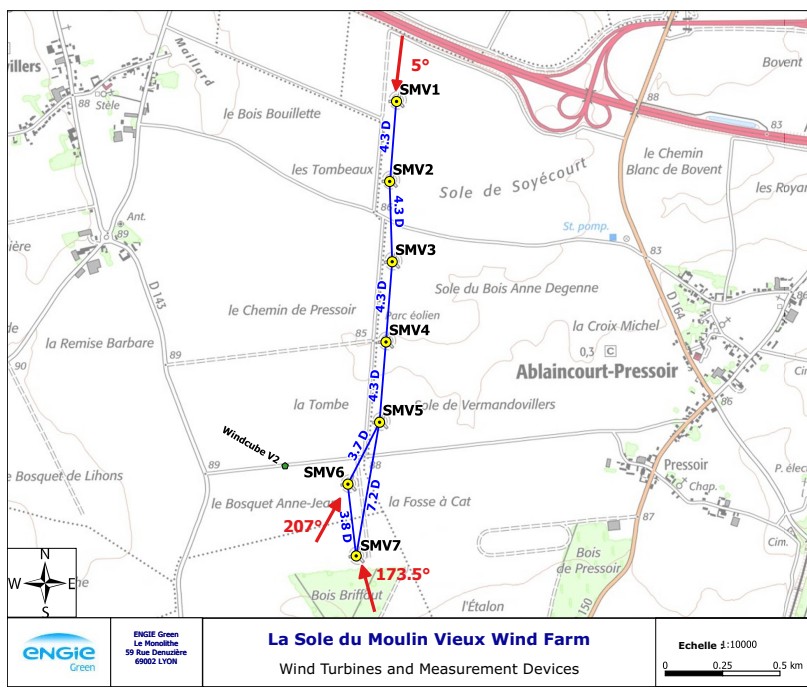

**Figure 1.** Layout of SMV wind farm. Distances between wind turbines (normalised by their rotor diameter $D = 82$ m) and directions related to SMV6 and SMV5 are also shown. The location of the Windcube v2 sensor used for the field campaign #2 is also indicated.

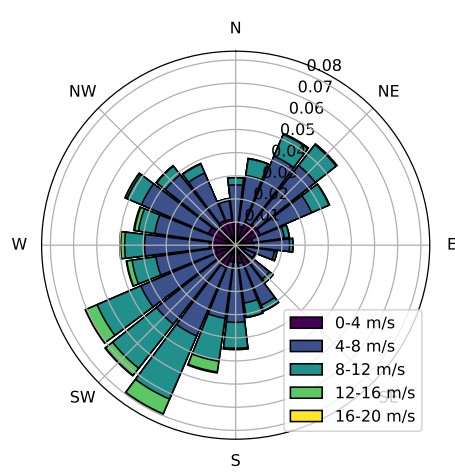

**Figure 2.** Long-term wind rose for the SMV wind plant at hub height (80 m). It was obtained through a correlation process between short-term met-mast measurements on-site and long-term reference wind data (ERA5 reanalysis data). Taken from Simley et al. (2021)

## 2.1 Calibration and Blind Test data sets

Twelve months of normal operation data, from October 1st 2016 to November 30th 2017 (removing the two months for which the wake steering tests were conducted) was provided to the benchmark participants to help them calibrate their wake models. It consists of 10-min statistics (average, minimum, maximum and standard deviation) of ten of the most important variables of supervisory control and data acquisition (SCADA) data, including active power, wind speed and direction, yaw angle, pitch angle, outdoor temperature, and rotor and generator speed.

The data was filtered to keep only timestamps when all wind turbines were operating at the same time. Any timestamps when curtailments were detected in one of the turbines were also removed from the data set. Yaw angle and direction data were corrected for north alignment issues, making sure those signals were consistent over the full period. One turbine experienced a modification of its blade aerodynamic

|  | Single wake blind test | Multiple wake blind test |
|---|---|---|
| Normal operations | $195° - 215°$ | $355° - 15°$ |
| Wake steering | $195° - 215°$ | $180° - 215°$ |

**Table 2.** Summary of wind direction ranges for each of the blind tests.





properties during the period. The effect of this change on turbine performance or anemometer wind speed measurement is unknown and could not be corrected.

The available data set for the wake steering experiment was unfortunately limited and could not be split into calibration and blind test subsets. Consequently the participants could only calibrate their wake deficit and superposition models, and did not 115 have any data to adjust neither the parameters of their wake deflection models nor the yaw-loss function of the misaligned turbine.

The blind data set was prepared using the same procedure as for the calibration data set. The only difference is that the active power signal was removed for all turbines and the wind speed signal was removed for all but the upstream turbines (SMV6 for south, south-westerly and SMV1 for northerly directions, indicated in Table 2). The Windcube data provided to the participants 120 cover the full wake steering period and part of the calibration data, starting on May 31st 2017 and ending on January 23rd 2018. All heights of measurements, ranging from 40 to 200 m were kept in the data set. The participants were left free to use these data to calibrate any onsite atmospheric parameters, such as the turbulence intensity or the wind shear.

## 2.2 Participating Models

Within the FarmConners benchmark, in total 5 participants (IDs = P4, P5, P6, P16, P17) have taken part in the SMV wind 125 farm field data blind test. The participating models cover a relatively broad range of assumptions, approximations and parameters representing the flow behind a steered turbine. Here in this section, these participating models are briefly described and implemented parameters are listed when relevant. Table 3 summarises the prominent characteristics of the participating models.

| | P4 | P5 | P6 | P16 | P17 |
|---|---|---|---|---|---|
| **Wind Farm Model** | FLORIS – non-homogeneous flow field | FLORIS – non-homogeneous flow field | – | – | – |
| **Wake Model** | Gauss-Legacy[a, h] – calibrated | Gauss-Legacy[a, h] – calibrated | Modified Ainslie Eddy viscosity[b] | Gaussian[h] – calibrated | Gaussian-IQ[c] |
| **Added Turbulence Model** | Crespo-Hernandez[d] | Crespo-Hernandez[d] | Quarton-Ainslie[e] | Frandsen[f] | Gaussian-IQ[c] |
| **Wake Superposition** | SOSFS[g] | SOSFS[g] | Sum of deficits | Quadratic | RLSOD[k] |
| **Wake Deflection** | Gaussian[h] | Gaussian[h] | Gaussian[h], Jimenez[i] | Gaussian[h] | Gaussian-IQ[j] |
| **Time-Series** | Yes | Yes | Yes | Yes | No |

**Table 3.** Overview of the participating WFFC-oriented models, Wind Farm Field Data Blind Test.
[a](Bastankhah and Porté-Agel, 2014), [b](Ainslie, 1988), [c](Ishihara and Qian, 2018), [d](Crespo and Hernández, 1996), [e](Quarton and Ainslie, 1990) – with sum of difference, [f](Frandsen, 2007) – IEC 2019 standard, [g]SOSFS – Sum of Squares Freestream Superposition, [h](Bastankhah and Porté-Agel, 2016), [i](Jiménez et al., 2010), [j](Qian and Ishihara, 2018), [k]RLSOD – Rotor-based Linear Sum of Deficits.

### 2.2.1 P4

**Wind speed and turbulence intensity**: The wind speed was defined using nacelle anemometers at each wind turbine. The turbulence intensity was calculated using the mean and standard deviation of the anemometer data for each wind turbine.





**Wind Direction**: It has been assumed that the average of the directions indicated by turbines' wind vanes was a good estimate of the free wind. The multiple wake simulations used the average of all wind turbines measurements, and the single wake simulations averaged SMV5 and SMV6 direction data.

| Turbine ID | $X_{UTM}$ | $Y_{UTM}$ | $Vg$ | $TIg$ |
|---|---|---|---|---|
| SMV1 | 633519 | 2539349 | 1.000 | 1.000 |
| SMV2 | 633489 | 2539000 | 0.997 | 0.938 |
| SMV3 | 633500 | 2538650 | 0.987 | 1.000 |
| SMV4 | 633473 | 2538300 | 0.979 | 1.026 |
| SMV5 | 633445 | 2537950 | 0.985 | 0.997 |
| SMV6 | 633307 | 2537680 | 0.971 | 0.984 |
| SMV7 | 633343 | 2537367 | 0.953 | 1.025 |

**Table 4.** P4: Non-homogeneous factors for multiple wake case

| Turbine ID | $X_{UTM}$ | $Y_{UTM}$ | $Vg$ | $TIg$ |
|---|---|---|---|---|
| SMV5 | 633445 | 2537950 | 1.015 | 1.014 |
| SMV6 | 633307 | 2537680 | 1.000 | 1.000 |

**Table 5.** P4: Non-homogeneous factors for single wake case

**Heterogeneous flow**: The non-homogeneous flow field in the steady simulation was obtained through speed factors between turbines in free sectors ($V_g$ – velocity factor, $TI_g$ – turbulence intensity factor). The factors were obtained using the wind turbines anemometer measurements, and they were applied regardless of the wind direction. These factors were defined with respect to the reference free wind turbine (SMV1 in multiple wake case – Table 4 and SMV6 in single wake case – Table 5).

**Free-stream wind speed and turbulence**: The free-stream turbine conditions for the blind test simulations were determined by the free-stream turbines (SMV1 in multiple wake case and SMV6 in single wake case). For calibration, the free-stream wind speed and turbulence intensity were obtained averaging the free-stream turbines depending on the orientation. The heterogeneous factors were taken into account in order to refer the magnitudes to SMV1 (reference wind turbine for the multiple wake).

**Wake Model**: The calibration performed by P4 was based on FLORIS model, using Gaussian velocity deficit model by Bastankhah and Porté-Agel (2014), and wake added turbulence was modelled with the Crespo-Hernández (Crespo and Hernández, 1996). The combination model selected was "SOSFS", which uses sum of squares freestream superposition to combine the wake velocity deficits to the base flow field (Katic et al., 1986). The yaw steering is represented by the deflection model Bastankhah and Porté-Agel (2016).

**Calibration**: The calibration process was performed using normal operation with multiple wake data. The data were discretized with respect to the assumed wind direction (average of all wind turbines, bin size = 5°) and free wind speed bins (bin size = 2 m/s). From these data, a calibration matrix was created, by extracting mean values for velocity, turbulence intensity and power in each wind turbine. Only cases where there is a wake effect were included in the calibration matrix. The calibration was performed using a genetic algorithm to obtain wake velocity and wake turbulence parameters. The parameters obtained in this process are presented in Table 6.




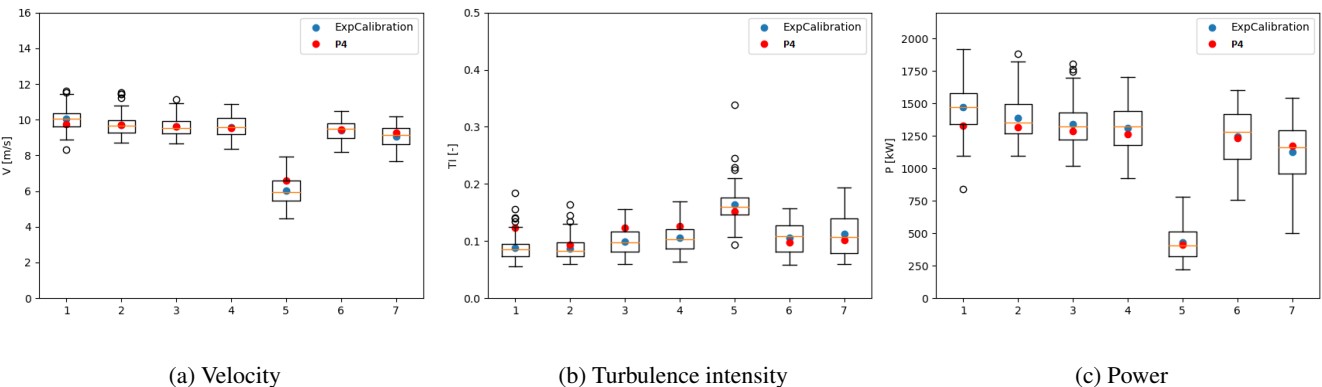

(a) Velocity       (b) Turbulence intensity       (c) Power

**Figure 3.** P4: Comparison per wind turbine (x-axis) between experimental data (boxplot), average values used for experimental calibration (ExpCalibration) and simulation with calibrated parameters (P4). Bin direction = 210º, Bin wind speed = 10 m/s.

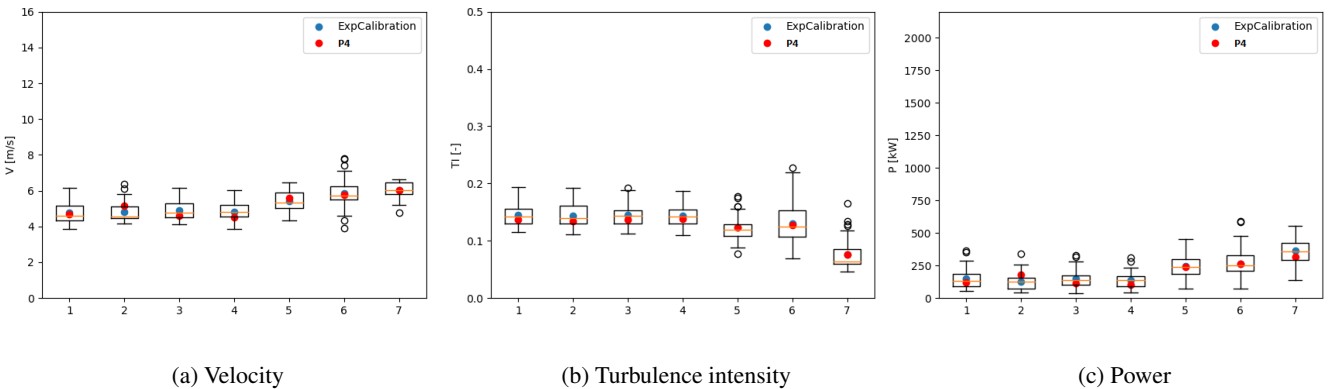

(a) Velocity       (b) Turbulence intensity       (c) Power

**Figure 4.** P4: Comparison per wind turbine (x-axis) between experimental data (boxplot), average values used for experimental calibration (ExpCalibration) and simulation with calibrated parameters (P4). Bin direction = 185º, Bin wind speed = 6 m/s.

The comparison between the original experimental data, the average values used for calibration and the simulation with the
final calibrated parameters can be observed for specific wind conditions in Figure 3 and Figure 4, as illustrative examples. The
original experimental data is shown as a boxplot, representing the median of the magnitudes and the dispersion of the data.
The calibration values and the final simulation are represented by points for each wind turbine. Figure 3 presents the results for
the bin centred in direction = 210º and wind speed = 10 m/s, where the wake is evident for wind turbine SMV5 (single wake
sector). The agreement in the values in terms of velocity, turbulence intensity and power is evident for wind turbine SMV5.
Figure 4 presents results for the bin centred in direction = 185º and wind speed = 6 m/s, where wind turbine SMV7 shows
higher power and wind velocity than the rest of the turbines. The agreement in terms of velocity, turbulence intensity and power
is clear.



FLORIS deflection parameters were not calibrated: the default parameters were used during calibration and subsequent simulation of the results. On the steered turbine, the power loss due to misalignment was modelled via yaw loss function in equation 1, where $\Psi$ is the steering control setting and $n$ is the yaw loss exponent, as stated in Table 6.

$$P = 0.5\rho A C_P u^3 (cos\Psi)^n \tag{1}$$

The yaw loss exponent could not be calibrated, and the value for the simulations was set as $n = 3$. In the blind test simulations, the yaw misalignment time-series data were used instead of using the intervals or average values.

It should be noted that the potential effects of atmospheric stability, shear and veer were not taken into account in the P4 simulations, and tilt angle of the wind turbines were neglected. The parameters listed in Table 6 follow the abbreviation convention in the FLORIS repository (NREL, 2021).

### 2.2.2 P5

The baseline engineering flow model FLORIS (NREL, 2021) was adapted to the site by introducing parametric correction terms, which are learnt from the available training SCADA data.

**Preparation of the calibration data set:** The tuning of the engineering model was performed using the provided calibration data set. As FLORIS is a steady state model, some of the SCADA data were discarded by looking at the nacelle position measurements, with the aim of only using data characterised by fairly steady and uniform inflow and operating conditions. Specifically, a 10-min SCADA data point was discarded if:

- The variation of the nacelle orientation of any turbine between two consecutive timestamps exceeds $20°$, so as to discard data measured under strongly-varying wind direction.

- The deviation of the nacelle orientation of any turbine from the average pointing direction of the entire wind farm exceeds $20°$, so as to discard data measured under highly non-homogeneous wind direction.

- The standard deviation of the nacelle orientation of any waked turbine was not null, so as to discard data recorded while one of the waked turbines was yawing during the 10-min period.

**Determination of the ambient wind conditions:** Once the data was prepared, it was possible to estimate the ambient wind conditions to be used as input to the engineering model. First, the ambient wind direction in each timestamp was calculated by computing the average of the wind directions measured with the turbines' wind vanes. Only observations within the southern $175° - 220°$ and northern $350° - 20°$ sector were kept. Overall, 5329 data points ($\approx 15.4\%$ of the calibration subset) for each of the 7 turbines were used for calibrating the flow model.

Free-stream turbines were used to determine inflow wind speed and turbulence. The wind speed was reconstructed from the turbine power, while the turbulence was computed using the mean and standard deviation values of the nacelle anemometer recordings. The determination of whether a turbine operates in free-stream or not was based on the wind farm layout and inflow wind direction, thus following the recommendations given in International Electrotechnical Commission et al. (2005). For wind directions in the sector between $345° - 25°$, measurements from SMV1 were therefore used. Measurements from SMV6 were





instead used for wind directions between $195° - 225°$. For wind directions between $170° - 195°$, corrected measurements from SMV7 were used, since it was expected that its sensed wind speed were affected by the nearby forest. In details, a $3^{rd}$ order polynomial function was best-fit to the ratio between the wind speed measured by the Windcube v2 at 80 m and the wind speed measured by the SMV7 anemometer, while both were operating in free-stream conditions. The resulting correction, scheduled as function of the SMV7 anemometer measurement, was then applied to both the calibration and blind test subsets. As for the

wind shear, a constant value of 0.25 was used, which corresponds to the average of the shears measured by the Windcube v2.

Finally, the complete data set was binned over wind speed and wind direction in bins of 1 m/s and 1°, respectively, so as to further reduce the measurement noise and speed-up the tuning process.

**Tuning parameters and process:** The velocity deficit was modelled with the kinematic Gaussian velocity deficit model by Bastankhah and Porté-Agel (2014) and the root sum of squared deficits superposition model (Katic et al., 1986). Wake

added turbulence was modelled with Crespo and Hernández (1996) turbulence model and deflection through Bastankhah and Porté-Agel (2016) deflection model. The model calibration parameters that describe the wake velocity and wake turbulence models were further corrected in the tuning process, whereas for the wake deflection model the default FLORIS values were used. In addition to that, the exponent $n$ of the cosine law (equation 1) used to model the power losses of yawed turbines was set to 1.88, *i.e* the value adopted by Gebraad et al. (2016).

As the incoming wind is permanently affected by the local orography and vegetation, it is necessary to account for the long-term, spatial variability of both wind speed and wind direction. To achieve that, a heterogeneous flow field is parameterised in terms of shape functions and associated speed-up ($\Delta$WS) and wind direction ($\Delta$WD) unknown nodal values (Schreiber et al., 2020). The nodes were placed at the coordinates of the turbines and the resulting mesh was further discretized for 5 different inflow wind directions. This resulted in 7 nodes for each of the considered inflow wind directions (WD-20°, WD-175°, WD-

195°, WD-220° and WD-350°) leading to a total of 35 speed-up and 35 wind direction nodes. The desired flow correction is then obtained by mapping the nodal values, with associated linear shape functions, to the locations of interest. The distribution of the nodes in terms of location and direction can be seen in Figure 5(a), which also depicts the resulting background flow field for an inflow speed of 7.56 m/s and a wind direction equal to 214.7°.

The intrinsic parameters of the adopted wake and turbulence sub-models were identified together with the heterogeneous

flow nodal quantities, resulting in a site-specific coupled simultaneous correction and tuning of the model. This ill-conditioned optimisation problem was solved by mapping the unknown parameters into an orthogonal space via the singular value decomposition (SVD), solving the identification by a maximum likelihood estimation in the reduced space, and then mapping back the solution to the physical space (Schreiber et al., 2020). The identified, tuned parameters for the wake model are shown in Table 6, while Figure 5(b) and (c) show the identified speed-up and wind direction nodal values.





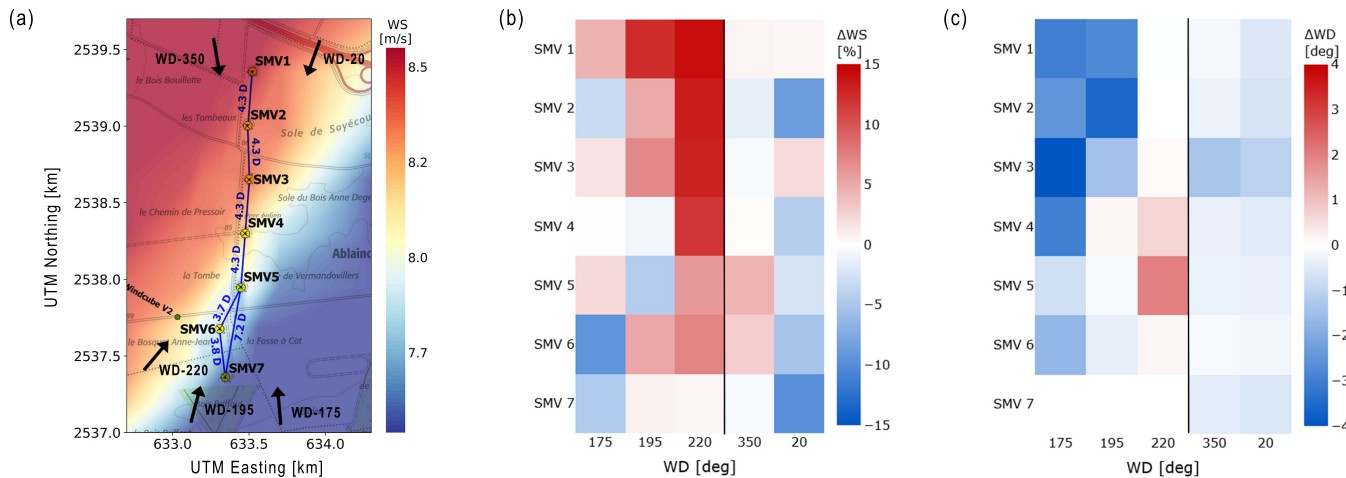

**Figure 5.** (a) Speed-up ($\Delta$WS) and wind direction ($\Delta$WD) nodes locations within the cluster. (b) Identified speed-up nodal values. (c) Identified wind direction nodal values.

| | $\alpha$ | $\beta$ | $k_a$ | $k_b$ | $TI,\text{constant}$ | $TI,ai$ | $TI,\text{initial}$ | $TI,\text{downstream}$ | $n$ |
|---|---|---|---|---|---|---|---|---|---|
| Initial value | 0.58 | 0.077 | 0.38 | 0.004 | 0.9 | 0.75 | 0.5 | -0.325 | |
| P4 | 0.352 | 0.108 | 0.576 | 0.00064 | 0.242 | 0.112 | 0.103 | -0.279 | 3 |
| P5 | 0.76 | 0.084 | 0.18 | 0.0035 | 0.74 | 0.88 | 0.05 | -0.316 | 1.88 |
| Simley et al. (2021)[*] | 0.58 | 0.077 | 0.38 | 0.004 | 0.5 | 0.8 | 0.1 | -0.32 | < 2.3 |

**Table 6.** Calibrated parameters of the used velocity and turbulence sub-models of the relevant models used by participants P4 and P5, where $\alpha$ & $\beta$ & $k_a$ & $k_b$ & $TI,\text{constant}$ & $TI,ai$ & $TI,\text{initial}$ & $TI,\text{downstream}$ are model parameters in FLORIS (NREL, 2021); and $n$ is the yaw loss exponent in equation 1.

[*]In Simley et al. (2021), the wake model fitting was realised by tuning the ambient turbulence intensity value rather than updating the model parameters. Within FLORIS (NREL, 2021), the Gauss-Velocity model (Bastankhah and Porté-Agel, 2014, 2016; Blondel and Cathelain, 2020; King et al., 2021), was used rather than the Gauss-Legacy (Bastankhah and Porté-Agel, 2014, 2016) as for P4 and P5.

### 2.2.3 P6

**Atmospheric conditions at the site:** The provided SCADA data covering the calibration period were used to model the site's characteristics and perform preliminary comparisons.

*Wind direction* - The wind rose obtained from the calibration period data is similar to the one shown in Figure 2, and contains a large portion of data for which no yaw offset is reported (about 65%), whereas the remaining 12046 data points report a yaw offset. The ranges of wind direction occurred during the provided blind test SCADA data are reported in Table 2. The yaw offsets during the calibration phase (calculated as the difference between the Windcube v2 LiDAR directional data at hub height and turbine SMV6's nacelle corrected heading) has a mean value of $-13.3°$, with values ranging from $-26°$ to $+30°$.





*Wind speed* - The only available source of wind speed for both the calibration data set and the blind test data set are from the nacelle-mounted anemometers at each turbine, if the Windcube v2 LiDAR is excluded. Preliminary correlations between

the LiDAR's and turbine SMV6's wind speed signals showed a non-linear relationship and significant amount of scattering, suggesting the underestimation of wind speeds by the nacelle-mounted anemometers, especially for wind speeds lower than 10 m/s (as measured by the LiDAR). As explained in the next section, transfer functions have been used to convert wind speeds from anemometers to free-stream wind speeds using the measured active power and the provided warranted power curve. Additionally, directional correlations between different turbines have been used to obtain a table of wind speed corrections that

represent the variation of wind speed at the site (also known as speed-ups). It is also stressed that any estimation of the wind farm blockage effect has not been attempted.

*Atmospheric Turbulence Intensity* - The variation of turbulence intensity with wind speed is shown in Figure 6 as measured from both the LiDAR and Turbine SMV6. Both the mean and the P90 turbulence levels in the two plots show some similarities especially for wind speeds above approximately 7.5 m/s. Although the correlation between the wind speed standard deviation

at the LiDAR and the SMV6's nacelle-mounted anemometers show significant scattering (not shown here), using the 10-min averaged wind standard deviation from the SCADA data is considered to be broadly suitable in this case for the purpose of wake modelling, as also pointed out in Duc et al. (2019). It is noted that the SCADA wind speed standard deviation data has not been used to obtain any turbine-specific turbulence intensity correction across the site, as was done for the wind speeds.

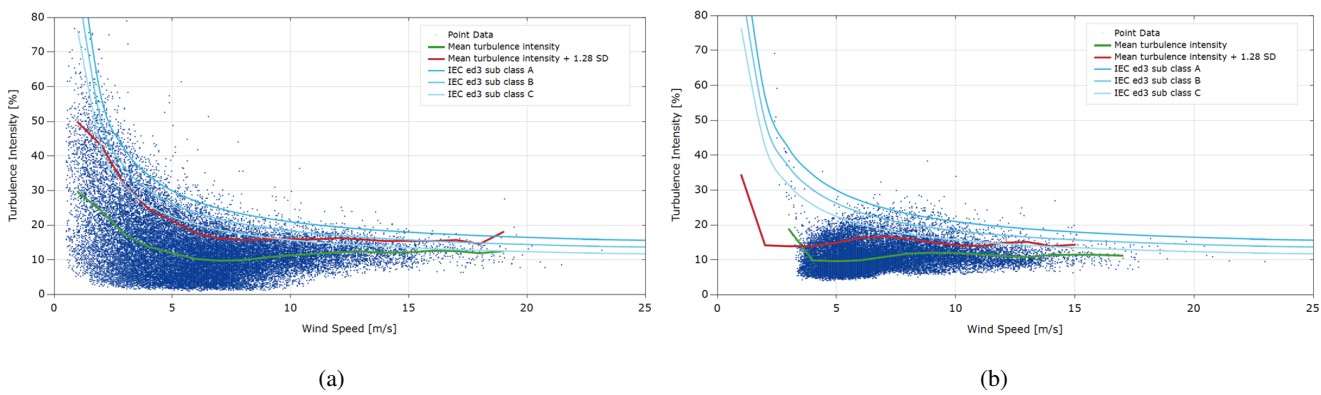

(a)                                                                                       (b)

**Figure 6.** Turbulence intensity measured by (a) Windcube v2 and 80m and (b) Turbine SMV6.

*Air density* - Historical atmospheric data have been obtained from ERA5 and NEWA reanalysis data sets at the closest

available nodes to the site. The long-term averaged air density at the site was found to be 1.23 kg/m$^3$, with values ranging between 1.13 and 1.35 kg/m$^3$. These values have been used to correct the power curve in the simulations (see below).

*Wind shear* - The LiDAR measurements were provided at different heights and for a period of about 8 months. The two measurements closest to the low tip and high tip of the rotor (40 and 120 m, respectively) have been used to estimate the average shear profile across the turbine rotor. Assuming the directional wind shear estimations at the LiDAR location are representative

for all turbine locations (except for the waked directions), power law exponents 0.178 and 0.268, respectively correspondent to wind direction bins centred at 0° and 210° have been used for the blind test simulations based on each case's wind direction





range (see Table 2).

**Calibration of turbine characteristics:** The first step for the calibration process was to focus on data representative of normal
operations only by filtering out any data recorded when yaw misalignment was present. Since a complete description of the
terrain roughness and orography near the site was not available, the variation of wind speeds across the turbine locations was
pragmatically estimated from correlations between the turbines' SCADA data, only from the two main directions of interest
for the blind tests, in Table 2. In order to exclude waked directions and have enough data points, the SCADA wind speed
signals at each turbine were filtered for broader directional sectors around the waked directions. By using SMV1 and SMV6
as reference turbines for the north and the south wind direction cases respectively, correlations between each turbine and each
reference turbine were performed for the filtered wind directions, in order to find speed-up factors describing the local variation
of free wind speeds for the directions of interest. The obtained values were used to define these effects for the excluded waked
directions, by averaging the valid directional values. An example of this approach is given in Figure 7.

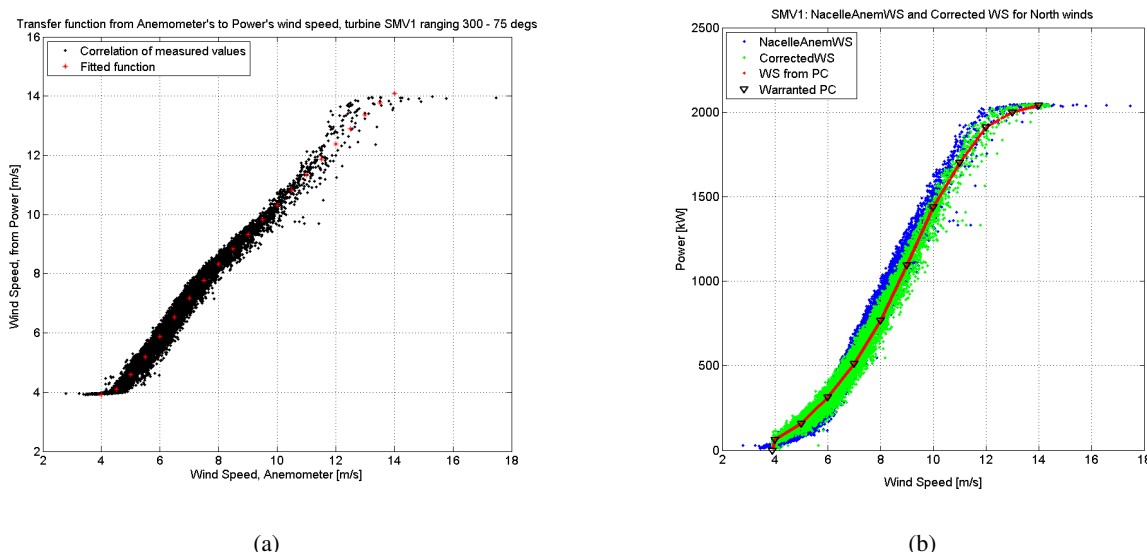

|        (a)        |        (b)        |

**Figure 7.** (a) Fitting between anemometer and power-based wind speeds at Turbine SMV1 for northern wind speeds under rated power. (b)
Comparison at Turbine SMV1 between warranted power curve (PC), power based on anemometer wind speeds, and corrected power curve
using the transfer function as described.

The turbine's wind speed standard deviations values were assumed to be representative of the free atmospheric conditions,
as described in the previous section, and have therefore been used for the definition of turbulent intensity.

Plotting the active power against the wind speeds from the nacelle-mounted anemometers for each turbine, large differences
emerge when compared to the provided warranted power curve. The wind speed could be back-calculated from the active
power SCADA signal; however, this signal is only provided for the calibration data set and not as part of the blind test package.
Since at least one turbine's SCADA wind speed signal was made available for each blind test scenario, the available calibration
data set was used to create a transfer function linking nacelle-anemometers' measurements and wind speeds back-calculated
from active power, exclusively for the wind directions of interest. This was achieved, for each turbine, by using a $6^{th}$ order





polynomial fitting process for SCADA wind speeds ranging between cut-in and rated wind speeds, hence covering the whole wind speed range in all the blind test scenarios analysed.

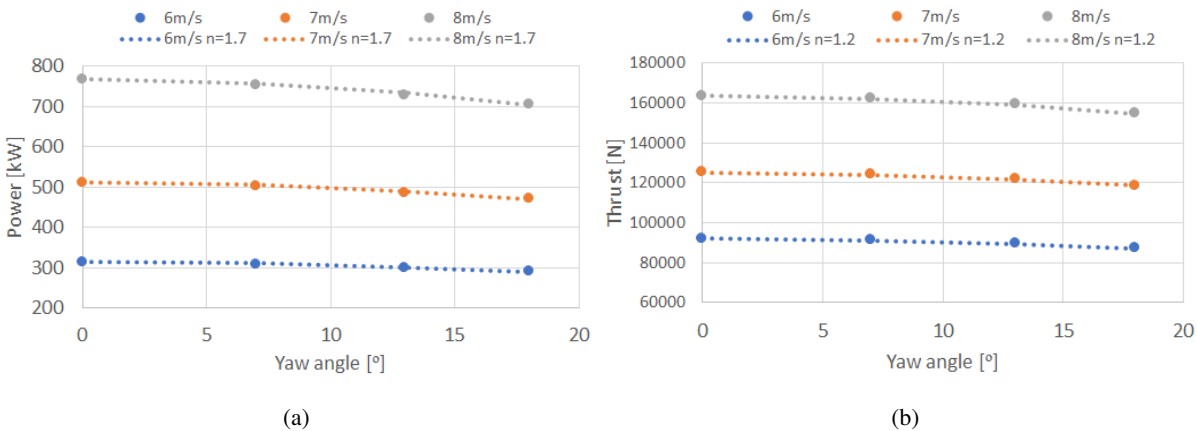

(a)                                                              (b)

**Figure 8.** (a) Power and (b) thrust values obtained from data calibration at three wind speed values and at four yaw-misalignment settings, compared to a $cos(\Psi)^n$ function.

In order to calibrate the power curves from the available data recorded when yaw misalignment was present, the data was binned for different yaw-angles ranges. Due to the limited amount of data recorded during the yaw misalignment tests, it was decided to focus specifically on three misalignment angles, $-7°$, $-13.3°$ and $-18°$, deemed to be representative of the distribution of measured yaw misalignment angles during the blind tests. A power curve for each of these angles was obtained by fitting the data clustered in these three bins, and the thrust curves were also adapted to the yawed cases by applying the

same wind speed shift found for the power curves. It is noted that the commonly used approach of modifying the power and thrust values by multiplying these by a factor such as $cos(\Psi)^n$ (where $n$ is a positive real number and $\Psi$ is the turbine yaw angle, as shown in equation 1) has not been used in this instance, also due to the large uncertainty given by the large range of exponent values found in literature. For completeness, a comparison between the calibrated power and thrust curves and the best-fitting cosine exponents (respectively being found to be 1.7 and 1.2) is shown in Figure 8 for wind speeds of 6, 7 and 8 m/s.


**Wake modelling:** Different wake modelling approaches have been tested for this specific site. The chosen steady-state model utilised, in a time-series fashion, for producing the results presented in this report is based on the Ainslie model (Ainslie, 1988), including modifications suggested by Anderson (2009) and Ruisi and Bossanyi (2019). The wake-added turbulence was modelled using Quarton and Ainslie (1990) model, and for the superposition of wake effects the sum of deficits method is

used for the velocity deficits, and sum of variances is used for turbulence superposition. The time-series of air densities has been used to correct the power curve during simulations, as prescribed by the IEC standards. The rotor-averaged quantities have been calculated by taking into account the directional wind shear across the rotor, as explained in the previous section. Furthermore, it is noted that the effects of wind farm blockage, atmospheric stability and veer have not been taken into account for the purpose of wake modelling at this site.





The model used for this simulation is the one by Bastankhah (Bastankhah and Porté-Agel, 2016), where the four numerical parameters used in the model were not calibrated (hence were kept the same as reported in the original article), however an additional factor of 2 is added to the formula predicting the skew angle (as implemented also in FLORIS (NREL, 2021)). This flavour of the model was preferred to the same model with the exclusion of this additional skew factor, or the model by Jimenez (Jiménez et al., 2010) for which the characteristic parameter $k$ was set to 0.05 (as opposed to 0.15, as suggested in the original

article), based on previous experience with other calibration data sets.

### 2.2.4  P16

P16 results are obtained using an in-house wake modelling code. The calculations are based on single-wake superposition with (Bastankhah and Porté-Agel, 2016) wake model, capable of modelling yaw deflec-

tion. The model parameters are optimised based on the provided calibration data set under SMV wind farm normal operation, using the SGA optimiser of the "pygmo" library (Biscani and Izzo, 2020). For the optimisation run, the data was filtered for wind directions coming from the north, reflecting the multiple wake case. Influences of orog-

raphy and forest, resulting in an in-homogeneous wind field, were not taken into account. Wind speed, wind direction and TI values from the upstream turbine (SMV1) were directly used for the definition of the ambient wind field. The parameters of the wake model were obtained by minimising the sum of the average quadratic differences between

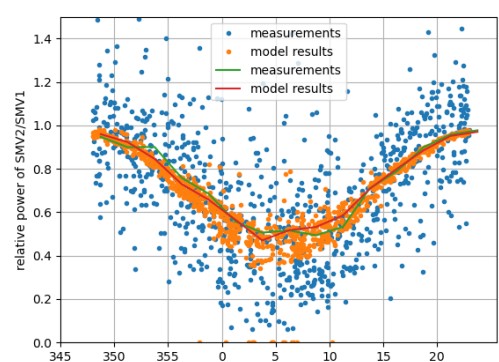

**Figure 9.** Comparison of measurements and P16 model outputs for calibration period, SMV2 compared to SMV1. x-axis indicates the incoming wind direction in [°]

power measurements and simulated power at every turbine of the wind farm. Following parameters were results of the optimisation and used for further modelling: $k_a = 0.234$, $k_b = 0.0037$ and $\alpha_d = 4.967$, $\beta_d = 0.0015$. Measurement uncertainties were not taken into account for the model calibration. We expect the high value of $\alpha_d$, compared to literature studies, to be a reflection of the general uncertainty especially of the wind direction measurement. As a large $\alpha_d$ leads to quite short near-wake lengths, which are essential for the magnitude of wake deflection in the deflection model, we expect the wake deflection with

these parameters to be quite inefficient. For the yawed turbine, the power loss is modelled via equation 1 with $n = 1.88$ and the change of $C_T$ with yaw is modelled analogous with a exponent of $n = 1$. This assumption was made due to the unknown behaviour of the turbine in yaw and based on Fleming et al. (2014).

Since there is no general agreement in literature, we expect the modelling of the turbine performance in yaw to be a large uncertainty factor. The multiple wakes are superposed via the quadratic sum of their deficits and the rotor effective wind speed

is calculated as the weighted sum at 19 points over the rotor. The wake added turbulence intensity is modelled with Frandsen (2007) model. Figure 9 shows the relative power of SMV2 compared to SMV1 for the data which was used for calibration of the model. In general, the mean wake loses show a good agreement with the measured data considering that the spread of the measured data is naturally higher due to measurement uncertainties.



### 2.2.5 P17

**Wake model description:** A modified Gaussian wake model named Gaussian-IQ, which provides three-dimensional wake
characteristics including wake width, velocity deficit, added turbulence (Ishihara and Qian, 2018), as well as wake deflection
caused by yaw offset (Qian and Ishihara, 2018), is utilised by P17. Parameters that govern the evolution of wake in Gaussian-IQ
model are determined as the function of thrust coefficient and local hub-height turbulence intensity. No additional calibration
of model parameters has been added and the default values were utilised. To combine velocity deficits of multiple wakes,
the Rotor-based Linear Sum (RLS) is employed. For turbulence intensity in the multiple wakes, it is formulated based on the
principle of Linear Sum of Square (LSS) with an additional correction term to consider the effects of wake interaction (Qian
and Ishihara, 2021).

**Turbine model description:** The theoretical power and thrust curves of Senvion MM82 at the site is used to determine the
turbine performance under normal operating conditions for an air density of 1.225 kg/m$^3$. For the steered turbine, to model the
power loss and thrust force change due to the yaw misalignment, an effective wind speed is introduced to power and thrust look-
up tables, following the approach recommended by Ruisi and Bossanyi (2019), feeding into equation 1 as $u_{eff} = u \cdot (cos\Psi)^{n/3}$
where, $\Psi$ is the yaw offset angle and $n$ is the yaw loss exponent set to 1.88 which is the value suggested by Gebraad et al.
(2016).

**Simulation process:** The wind farm simulation is performed in steady-state and results are provided via a data set binned
over wind speed and wind direction (WD) in bins of 1 m/s and 5°, respectively. For each wind speed, the binned values
with respect to, for example WD = 5° ± 2.5°, are obtained by taking the average of the results from 10 simulations of WD
= 2.5° : 0.5° : 7.5°. The wind speed, wind direction and turbulence intensity (TI) in the axial direction are assumed to be
homogeneously distributed in the wind farm. A wind shear profile following the power law with a constant exponent of 0.15
is applied to the inflow wind speed. As shown in Figure 6, the variation of turbulence intensity with wind speed measured by
the free-stream turbine SMV6 is quite stable for wind speeds above approximately 4 m/s. Thus, the ambient turbulence level
at hub height is set to be constant with the mean value of TI = 0.11.

### 2.3 Validation Data pre-processing

The wind farm field data blind tests are prepared as Single and Multiple wakes, where the upstream turbine SMV6 was steered
with -13.3° mean with respect to the incoming south-westerly wind. The results are evaluated in terms of balanced energy
gains, as defined in Fleming et al. (2019). Since the experiments were not realised in the form of a toggle test, the baseline case
is taken by considering two 2-months periods before and after the field tests, as shown in Figure 10.

The wind speed and direction reference signals for computing the energy ratios come from the Windcube v2. Due to the
proximity of this sensor with the controlled turbine, the atmospheric conditions measured should be very similar to the ones
faced by SMV6. The reference power signal is issued from SMV7 wind turbine, which is the only remaining upstream turbine
for the southerly wind sector. However, SMV7 is located very close to the forest and therefore experiences a much disturbed
wind compared to SMV6. Consequently the SMV7 active power signal must be corrected to be representative of SMV6 in



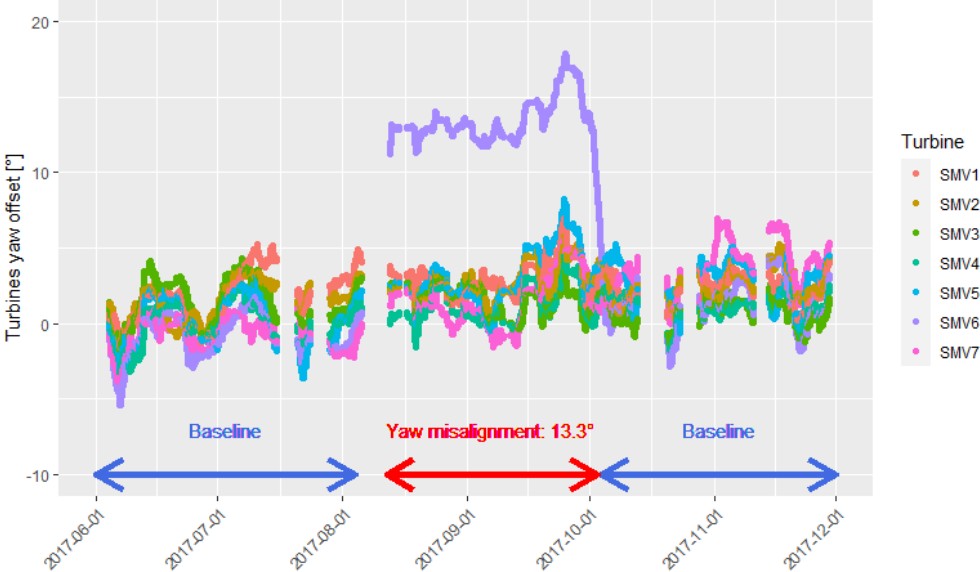

**Figure 10.** Evolution of the yaw offset of all turbines in the farm during the period of the analysis. The reference wind direction is taken from the Windcube lidar. To smooth out the time-series, a moving average of 3 days was applied. The periods corresponding to the baseline case and the wake steering case are marked by the arrows. The averaged misalignment angle on SMV6 turbine is indicated.

baseline situation. This is done following the same procedure as explained in Simley et al. (2021); namely under normal operation, SMV6 and SMV7 active power signals are binned against wind speed (1 m/s bins) and wind direction (calculated every degree on overlapping 10° bins). Then, a transfer function is estimated by dividing the SMV6 averaged power by SMV7 averaged power in each bin. Finally this transfer function is applied on SMV7 power time-series to generate a reference power signal used in both the baseline and the wake steering cases for the computation of the energy ratios.

### 2.4 Single Wake Results

For the single wake case, the estimated and observed performance of the SMV6–SMV5 turbine pair is investigated within a narrow wind sector of 200°–215° (*i.e.* ±7.5° around the perpendicular direction), where the upstream turbine SMV6 is misaligned for 13°–15° counter-clockwise (negative misalignment with mean ≈ -13.3°). The filtered data set for wake steering consists of 216 10-min data points, while the normal operation data set used to calculate the baseline wake effect is made of 1120 10-min data points (484 recorded in June and July 2017, 616 recorded in October and November 2017).

#### 2.4.1 Time-series comparison

For the submitted time-series results from P4, P5, P6 and P16, the predicted and observed power values at the upstream and downstream turbines are compared in Figure 11. The root mean square errors are normalised (NRMSE) by 2050 kW rated





power of the turbines. For the upstream turbine, SMV6, with wake steering control (or yaw misalignment) of -13.3° mean counter-clockwise, P4 and P16 are seen to underestimate the power production for lower wind speeds. On the other hand, P5 seems to overestimate the SMV6 turbine power for higher wind speeds, around the transition between Region II and III. P6, however, is seen to have a very good agreement with the observed power at the controlled SMV6 turbine upstream, though

potentially with a slight under-estimation.

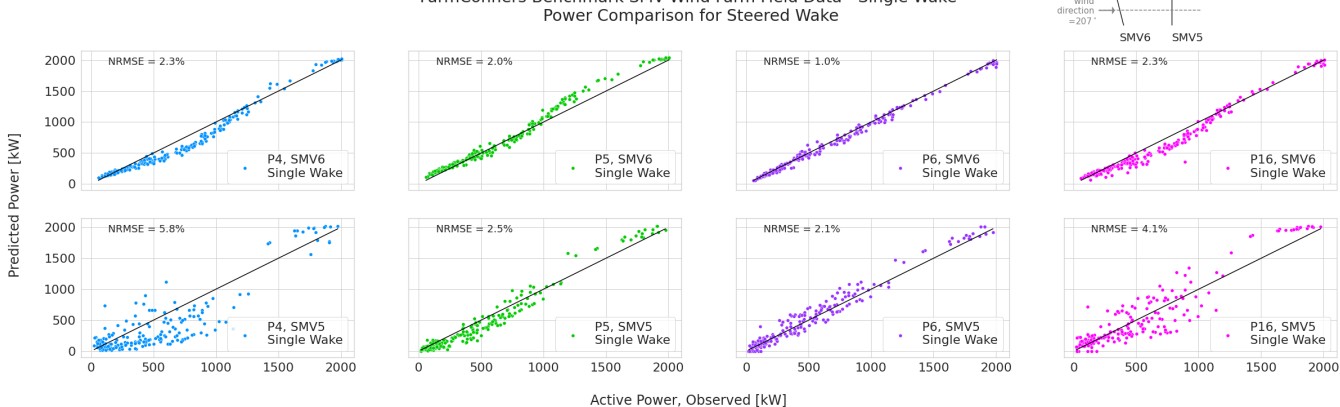

**Figure 11.** SMV Wind Farm (WF) Field Data, Single Wake under Wake Steering Power comparison per participant. (Top Row) Upstream turbine power under wake steering control of -13.3° misalignment, (Bottom Row) Downstream turbine power under the steered wake – field observations vs. participating model predictions. Representative layout with corresponding yaw control set-point is illustrated at the upper right corner.

At the downstream turbine under a steered wake, SMV5, the variance around the power predictions are notably higher for the steady-state models, potentially driven by the wake added turbulence. The underestimation trend by participants P4 and P16 continues at SMV5 power comparisons as well, although a lesser discrepancy is observed for P16 results.

As stated earlier, the calibration data set the participants were provided for the wind farm field blind tests is limited to

normal operation conditions. The limited calibration data surely affects the performance of all the participating models. Its impact is arguably the most visible for P4 and P5, where the same WFFC-oriented platform is utilised. Between P4 and P5, the difference in the comparison for both the upstream and downstream power predictions are expected to be driven by the prior calibration and the final selection of the parameters for the controlled periods. Specifically, the implemented yaw loss exponents $n = 3$ for P4 and $n = 1.88$ for P5 (see Table 6) are argued to be the main factor for the difference observed in the

upstream power predictions; especially compared with the recent field calibration at the same site under WFFC, discussed in (Simley et al., 2021) where wind speed (or indirectly $C_T$) dependent values of $n \approx 2.2 - 2.3$ for wind speeds between $4 - 8$ m/s, $n \approx 1.3 - 1.35$ for $8 - 12$ m/s and $n \approx 0.36$ for $12 - 14$ m/s are reported. On that regard, Figure 11 highlights the sensitivity of the widely adopted WFC-oriented models to the employed parameters and the importance of a comprehensive calibration data/process. It also shows the significance of clear parameter descriptions for overall reproducibility of the results.



### 2.4.2 Binned Quantities of Interest: Energy Ratio & Power Gain

To analyse the effect of wake steering on the 2-turbine configuration wind farm (SMV6 as upstream, SMV5 as downstream) a similar methodology to that described in (Fleming et al., 2019) is followed. Accordingly, both the observed power and the estimations by the participating models are distributed over $\pm 0.5$ m/s wind speed and $\pm 2.5°$ wind direction bins. Per bin, the energy ratio, $R_{\text{Energy}}$ is calculated via equation 2 with weighted summation; where $N$ is the total number of wind speed bins per sector $(i)$, $\omega_i$ are the weights per bin, $P_i^{\text{Test}}$ is the mean of the total power per bin (either the normal operation/baseline power or power under wake steering WFFC, summation of SMV6 and SMV5 turbines for the single wake case); and $P_i^{\text{Ref}}$ is the gross production averaged per bin, *i.e.* the power of the wind farm without the wake losses ($P_i^{\text{Ref}} = 2 \cdot P_{\text{SMV6}}$ for single wake case).

$$R_{\text{Energy}} = \frac{\Sigma_{i=1}^{N} \omega_i P_i^{\text{Test}}}{\Sigma_{i=1}^{N} \omega_i P_i^{\text{Ref}}} \qquad (2)$$

The weights per wind speed bin, $\omega_i$, aim to compensate for the non-equivalent number of samples within 'Test' (numerator) and 'Ref' (denominator) periods in energy ratio estimation. Accordingly, they are assigned via the relative density of the samples within the respective bins. As an indication of uncertainty around the energy ratios, the standard deviation of the total power per wind speed bins are propagated, assuming the numerator and denominator of equation 2 is uncorrelated. Detailed description of the weighting strategy and the energy ratio calculation as well as the simplified uncertainty propagation can be found at the post-processing notebook published at the open-access FarmConners benchmark repository (Göçmen et al., 2021). It should be underlined that the resulting distribution of the model results consider only the variance in the time-series samples within the bin, therefore simplistic and potentially conservative. A comprehensive analysis including input and model (parameter) uncertainties and their propagation is left as future work.

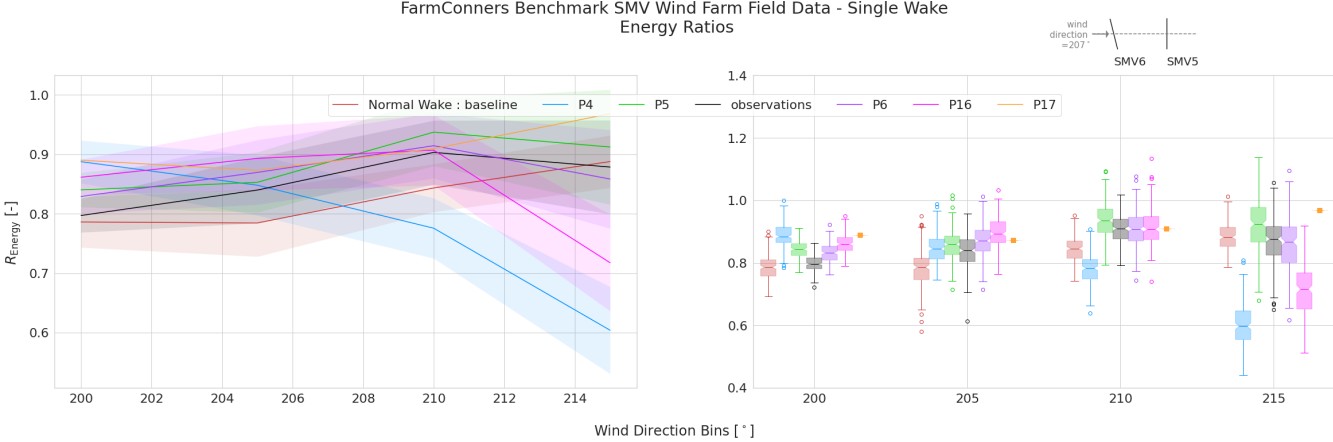

**Figure 12.** SMV WF Field Data, Single Wake under Wake Steering - Energy Ratio comparison under wake steering control with -13.3° upstream misalignment. Representative layout with corresponding yaw control set-point is illustrated at the upper right corner.





Figure 12 shows the energy ratio, $R_{\text{Energy}}$ during the normal operation (or baseline) as well as under wake steering WFFC as observed on the field or estimated by the participating models (P4, P5, P6, P16, P17), per wind direction bin. Especially in the close to perpendicular wind sector, where wind direction $\in 207° \pm 5°$, it can be seen that the steered wake is observed and estimated to be more energetic (*i.e.* higher $R_{\text{Energy}}$); indicating a positive energy gain. The behaviour and the scale of the observations are in line with the recent field test results from the same SMV wind farm (Simley et al., 2021). For the participating models, the agreements are significantly better in the close sector. However, the variations notably increase at the wake border around $215°$. For the first wind sector centred around $200°$, all the models are seen to overestimate the energy ratios compared to the observations. The overall agreement becomes much better closer to the wake centre at $205°$ bin, where P16 has a slight over-estimation. In line with the power scatter plots in Figure 11, P4 notably underestimates the energy ratio for the remaining two-sectors; where P5 and P6 have similar and overall very good agreement with the observations, P16 has mostly good agreement except of the significant under-estimation around the wake border at $215°$. Note that for P17, where only the pre-binned quantities of interest were submitted, Figure 12 includes only the mean $R_{\text{Energy}}$. Based on the mean quantities, P17 is observed to slightly overestimate the energy ratios for almost all the sectors analysed.

The power gain observed at the field and estimated per participant in the blind test is then calculated following equation 3, where the energy ratio computed in equation 2 during normal operation, $R_{\text{Energy}}^{\text{Test = Normal Operation}}$ is subtracted from the energy ratio under wake steering flow control $R_{\text{Energy}}^{\text{Test = WFFC}}$. The uncertainty around the power gain is quantified via propagating the uncertainties of $R_{\text{Energy}}^{\text{WFFC}}$ and $R_{\text{Energy}}^{\text{Normal Operation}}$ estimated in equation 2.

$$P_{\text{GAIN}} = \left( R_{\text{Energy}}^{\text{WFFC}} - R_{\text{Energy}}^{\text{Normal Operation}} \right) \cdot 100 \tag{3}$$

Figure 13 compares the power gain under wake steering WFFC observed at the SMV wind farm and estimated by the participating models in the blind test. The boxplots show that for the close wake sector, *i.e.* wind direction $\in 207° \pm 5°$, a positive power gain with $13° - 14°$ yaw control at the upstream turbine has above 75% likelihood. In fact, the likelihood of more than 5% gain in power exceeds 50% in the the same wind sector. However, around the borders of the wake, for the bins centred at $200°$ and $215°$ loss of power is equally as likely as a potential gain; indicating the importance of uncertainties for the risk assessment of WFFC implementation, also as underlined in Hulsman et al. (2020). Observably, the participating model behaviours for the power gain estimations follow the discussions on $R_{\text{Energy}}$ for Figure 12. However it should be noted that, P17 power gain now has a variation around its estimation, driven from the standard deviation of the energy ratio for the normal operation, $R_{\text{Energy}}^{\text{Normal Operation}}$ in equation 2. With that, the over-estimation trend is down-scaled and a better agreement can be argued for larger sectors $> 200°$.





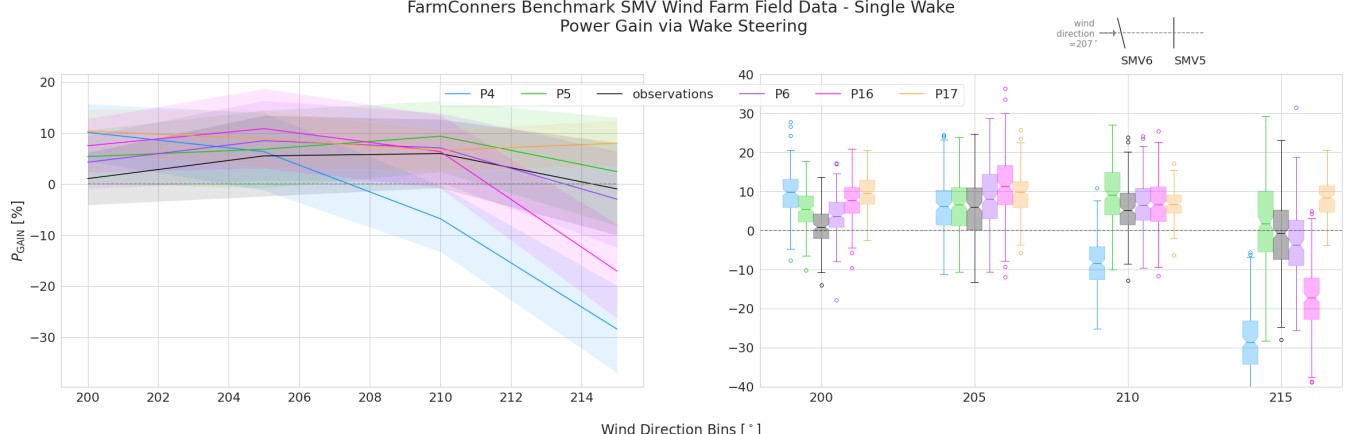

**Figure 13.** SMV WF Field Data, Single Wake under Wake Steering - Power Gain comparison under wake steering control with -13.3°
upstream misalignment. Representative layout with corresponding yaw control setting is illustrated at the upper right corner.

## 2.5 Multiple Wakes Results

For the Multiple wake case, the estimated and observed performance of the SMV6–SMV1 downstream turbines is investigated
within a larger wind sector starting from 180° up to 215°, so that effects of the wake steering at SMV6 can also be observed
at the turbines further downstream of SMV5. Due to the imperfect alignment of the farm layout, it must be noted that this

does not correspond to a multiple full wake effect, instead it is most probably a combination of overlapping partial wakes.
Furthermore, for wind directions close to 180°, the misaligned turbine SMV6 is in the wake of SMV7. Given the uncertainty
in the evaluation data set, mainly driven by the layout of the wind farm, the multiple wake results of the wind farm field data
blind tests are presented in Appendix A.

## 2.6 Summary of the Wind Farm Field Data Blind Test

Although it is the key priority for advancing wind farm control technology (van Wingerden et al., 2020), the field test and
validation for WFFC-oriented models are significantly challenging due to the stochasticity, non-stationarity, high variability
and overall uncertainty. Specifically for the FarmConners benchmark, the highlights of the participating model performance
for the wind farm field data blind test can be summarised as below.

**Similar models, disparate behaviour** In this blind test, several participants implemented similar models to resolve the
wake behaviour behind a steered turbine (as listed in Table 3). However, even for the same framework utilised by
  P4 and P5, the results are seen to notably differ. It indicates high model sensitivity to the employed parameters and
  emphasises the importance of the calibration procedure, as also analysed for another wind farm by van Beek et al.
  (2021). It also underlines the significance of clear methodology description and parameter listing for reproducible and
  credible estimations of the potential benefits of the technology.



**Importance of the calibration data set** In the wind farm field data blind test, the calibration data was confined to the normal operation periods. It was mainly due to the limited availability of the controlled operation, which is typically the case for the majority of the operating wind farms. However, the blind test results show how crucial the information regarding the power loss at the controlled turbine(s) and basic downstream behaviour is to be able to customise the low-cost, control oriented models under low observability of inflow conditions and turbine response.

**A better wake steering implementation at the field** Given the high sensitivity of the parameters on the prediction of the gains observed, it can be concluded that the wake steering solution in practice would/should not be designed solely based on normal operation data. It would/should rather follow an iterative process in which *a priori* strategy can be developed and implemented based on normal operation tuning or standard/recommended/off-the-shelf parameter values. After a certain period of data collected (*e.g.* a few months), the model parameters could be updated and *a posteriori* strategy (with a new set of optimal yaw control set-points) can then be defined. Such a process could continue until a satisfactory agreement is reached between the model results and the observations.

## 3 BLIND TEST #2: CL-Windcon Wind Tunnel

The data set used for this benchmark exercise has been obtained through wind tunnel testing campaigns performed during the EU CL-Windcon project. Specifically, this paper focuses on the experiments conducted with up to 3 scaled wind turbines operating both aligned and misaligned with respect to the wind tunnel inflow.

### 3.1 Experimental Setup

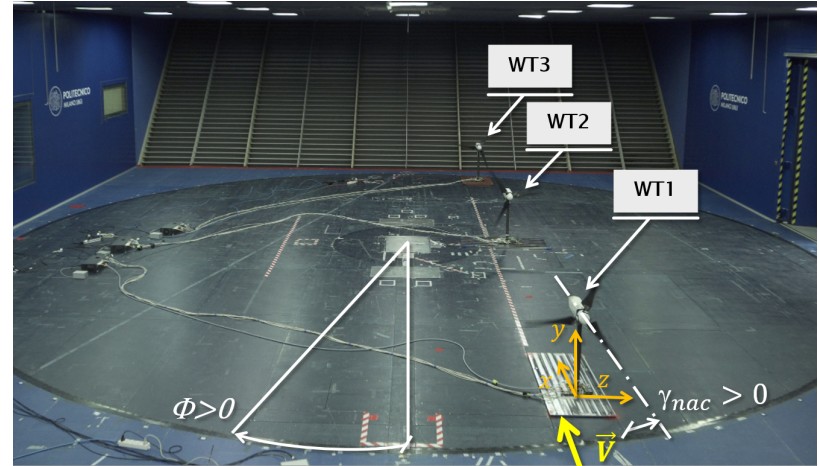

**Figure 14.** Experimental setup, showing the three model turbines mounted on the wind tunnel turntable. The x–y–z frame is fixed with respect to the tunnel and does not rotate with the turntable (Campagnolo et al., 2020).

The scaled cluster is the one described in Campagnolo et al. (2020), and is depicted in Figure 14. It is composed of up to three fully sensorized scaled G1 wind turbines installed on the 13 m diameter turntable of the atmospheric test section of the wind tunnel of the Politecnico di Milano. The pitch and generator torque of the G1s are regulated by a standard power controller, as well as the machines can operate misaligned with respect to the incoming flow direction of an angle $\gamma$, defined positive for a counterclockwise misalignment viewed from the top. Each G1 is equipped with a rotor whose diameter $D$ is equal to 1.1 m, while the hub height is 0.825 m. Further features of the G1s are de-





scribed in Campagnolo et al. (2020) and Bottasso and Campagnolo (2021), while Wang et al. (2021) discusses about the similarity of the wake shed by a G1 with respect to the one shed by a multi-MW wind turbine.

WT1 (upstream), WT2 (center), and WT3 (downstream) turbines are longitudinally spaced 5D and located 1.5D aside of
the center-line of the turntable. This last can be rotated by the angle Φ to simulate different wind directions; Φ is zero when the turbine row is parallel to the wind tunnel center-line, while it is positive for a clockwise rotation of the turntable, if viewed from the top as shown in Figure 14.

During the experiment, the ambient wind speed $U_{\mathrm{Pitot}}$ was measured by a pitot tube located outside the turntable, 3D upwind of WT1 and at hub height. Two boundary layers, characterised by moderate (mod-TI) and high (high-TI) turbulent inflows,
were instead simulated by placing roughness element on the floor and turbulence generators at the chamber inlet. The resulting inflow was measured with three-component constant-temperature hot-wire probes (CTA), scanning a vertical plane 4D upwind of WT1. The corresponding speed mappings, normalised with respect to the inflow speed measured at the pitot tube location, are depicted in Figure 15(a) and (b). The vertical profiles of the normalised wind speed measured at the pitot location are instead shown in Figure 15(c) on the left, together with their best-fitting power-laws. Finally, the vertical profiles of the turbulence
intensity are shown in Figure 15(c) on the right.

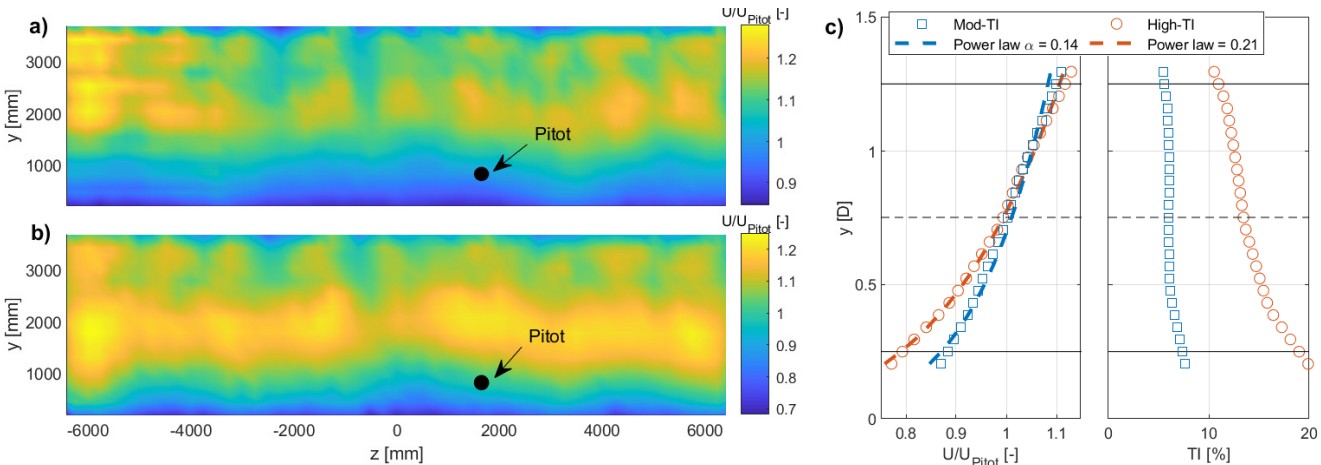

**Figure 15.** Characteristics of the inflows simulated within the wind tunnel and measured by the CTA probes: normalised wind velocity (a,b); vertical profiles of the normalised wind speed and their best-fitting power-laws (c, left), and vertical profiles of the turbulence intensity (c, right); within (c), the black dashed line indicates the hub height, while the two solid black lines limit the rotor disk.

### 3.2 Calibration data set

The data set provided for the calibration consists of:

– Time-series of 3 wind speed components measured with the CTA probes and within the wakes behind one or two G1s,

– Time-series of the air density and wind speed measured by the pitot tube placed 3D upstream of WT1,



– Time-series of the G1s collective pitch angle, rotor speed and azimuth position, nacelle orientation, main shaft torque and two out-of-plane bending moments measured on the shaft and at tower bases,

     – The inflow mappings shown in Figure 15,

     – A FASTv8 (Jonkman and Jonkman, 2016) model of the G1 featuring ad-hoc tuned airfoil polars (Wang et al., 2020), and the $C_P$ and $C_T$ versus wind speed relationship for non-yawed (normal) operations.

A detailed description of the calibration data set provided to the participants for single and multiple wake cases is presented below.

**Single full-wake** Measurements of the wake shed by a single G1 operated aligned ($\gamma = 0°$) and misaligned ($\gamma = \pm20°$) with respect to the wind tunnel inflow. The wake flow was sampled at 5D, 7.5D and 10D downstream positions along 1) a horizontal line located at hub height, and 2) a vertical line passing through the wake center. Among the shared G1 signals are the data recorded during the wake measurements, as well as the data recorded on board of the two aligned G1s ($\Phi = 0°$) operated at null yaw misalignment angles, representing normal operation.

**Single partial-wake** Physical quantities measured on board two G1s laterally spaced 0.5D. During the experiments, WT2 was kept aligned with respect to the wind tunnel inflow, while WT1 was operated both aligned ($\gamma = 0°$) and misaligned ($\gamma = 30°$).

**Multiple wakes** Measurements of the wake shed by two aligned and 0.5D laterally spaced G1s. In both cases, data were recorded for two different combinations of WT1 and WT2 misalignment. The wake flow was sampled at 7.5D and 10D downstream of WT2 and along three horizontal lines; 1) located at hub height, 2) 0.36D above and 3) 0.36D below hub height. Among the shared G1 signals are the data recorded during the wake measurements, as well as the data recorded on board of the three G1s operated at $\gamma = 0°$ and with turntable angle $\Phi$ equal to $0°$ and $-6.9°$ (in this last case, two nearby G1s resulted being laterally spaced of 0.6D).

### 3.3   Blind Test data set

The data set provided for the blind tests consists of:

     – Time-series of the air density and of the wind speed measured by the pitot tube placed 3D upstream of WT1,

     – Time-series of the nacelle orientation, collective pitch and rotor speed for all the involved wind turbines.

Each participant was then required to provide the values of the expected average wind speed and turbulence intensity of the longitudinal flow at several locations within the wake, as well as the expected average power outputs for all the involved wind turbines. A detailed description of the required predictions, for each of the three considered cases, is presented below.

**Single full-wake** The expected average wind speed and turbulence intensity of the longitudinal flow within the wake shed by a single G1, which was operated at six different misalignment angles ($\gamma = \pm10°, \pm30°, \pm40°$). Furthermore, participants



were required to provide the expected power production for two aligned G1s operated at 50 different combinations of
       WT1 and WT2 yaw misalignment.

   **Single partial-wake** The expected power production for two G1s laterally spaced 0.6D (achieved by rotating the turntable of
       $-6.9°$), and operated at 25 different combinations of WT1 and WT2 yaw misalignment.

   **Multiple wakes** The expected longitudinal flow speed within the wake shed by two aligned and 0.5D laterally spaced G1s. In
both cases, participants were required to predict the flow field for two different combinations of WT1 and WT2 misalign-
       ment. Furthermore, it was also required to provide the expected power production for three aligned ($\Phi = 0°$) G1s, with
       WT3 operated with null yaw misalignment ($\gamma = 0°$), and with WT1 and WT2 operated at 50 different combinations of
       yaw misalignment. Similarly, it was required to provide the estimations of the produced power when testing with three
       G1s laterally spaced 0.6D one from each other (achieved by rotating the turntable of $-6.9°$). Also in this case, WT3 was
kept aligned to the wind tunnel inflow, while the nacelle orientations of WT1 and WT2 were set equal to 25 different
       combinations.

### 3.4   Participating Models

The overview of the participating models is given in Table 7. Two participants (P3 and P18) have submitted results for CL-
Windcon wind tunnel blind tests where P3 has participated with essentially 4 different models. For each model, further details
are given in the following subsections.

| | P3, "Gauss" | P3, "Super-Gauss" | P3, "Gauss" Inflow Map. | P3, "Super-Gauss" Inflow Map. | P18 |
|---|---|---|---|---|---|
| **Wind Farm Model** | FarmShadow™ | FarmShadow™ | FarmShadow™ | FarmShadow™ | Qian-Ishihara[c] |
| **Wake Model** | Gaussian[a] | super-Gaussian[b] | Gaussian[a] | super-Gaussian[b] | Qian-Ishihara[c] |
| **Added Turbulence Model** | Qian-Ishihara[c] | Qian-Ishihara[c] | Qian-Ishihara[c] | Qian-Ishihara[c] | Qian-Ishihara[c] |
| **Wake Superposition** | Local-Linear-Sum[d] | Local-Linear-Sum[d] | Local-Linear-Sum[d] | Local-Linear-Sum[d] | SOSFS[g] |
| **Wake Deflection** | Gaussian[e] | super-Gaussian[f] | Gaussian[e] | super-Gaussian[f] | Qian-Ishihara[c] |
| **Wake Predictions** | Yes | Yes | Yes | Yes | No |

**Table 7.** Overview of the participating WFFC-oriented models for the CL-Windcon Wind Tunnel Blind Test. [a](Bastankhah and Porté-Agel, 2014), [b](Blondel and Cathelain, 2020), [c](Qian and Ishihara, 2018), [d](Niayifar and Porté-Agel, 2015), [e](Bastankhah and Porté-Agel, 2016), [f](Blondel et al., 2020), [g](Katic et al., 1986)

#### 3.4.1   P3

Participant P3 used two sets of models, implemented in a solver called FarmShadow™, where the first one is based on a
Gaussian wake model, as described in Bastankhah and Porté-Agel (2014), together with the yaw model from Bastankhah
and Porté-Agel (2016). For the velocity deficit, the calibration proposed in Niayifar and Porté-Agel (2015) is used. Default



parameters proposed in Bastankhah and Porté-Agel (2016) are used for the yaw model. The second set is based on the super-Gaussian model for the velocity deficit (Blondel and Cathelain (2020)), using the calibration proposed in Cathelain et al. (2020). Regarding the wake deflection, the super-Gaussian-based approach introduced in Blondel et al. (2020) is applied, using again the calibration proposed in Cathelain et al. (2020). For the two sets, the wake added turbulence model proposed by Qian and Ishihara (2018) is applied, without modifications. A so-called "Local-Linear-Sum" superposition algorithm is employed

to accumulate the velocity deficits, while a so-called "Maximum-Value" algorithm is employed for the wake added turbulence accumulation. Down-regulated wind turbine operating conditions have been disregarded, and the provided $C_P$ and $C_T$ versus wind speed look-up table have been used directly. To account for power (or thrust) losses due to a wind turbine misalignment, the classical cosine-squared low is used, *i.e.* the power of a yawed turbine is given by: $P_\gamma = cos(\gamma)^2 P_{\gamma=0}$, with $\gamma$ the yaw angle. The model implementation is steady-state and three-dimensional, rotors being treated as actuator-disks, discretized

using a $9 \times 9$ polar grid.

Two different inflows are considered. First, the mean velocity measured by the pitot tube, together with the turbulence intensity provided within the benchmark are imposed everywhere in the domain (*i.e.* Uniform Inflow approach). Second, the provided steady-state wind tunnel inflow maps are used as an input (*i.e.* Inflow Map. approach). In this case, the inflow map velocities are rescaled using the time averaged velocity measured using the pitot tube, based on the pitot tube location on the

map. Two different maps are used, one for the so-called mod-TI cases, the other for the so-called high-TI cases. The turbulence intensity is kept constant over the whole domain.

### 3.4.2 P18

P18 used an in-house code, in which the wake deficit and wake deflection are modelled by a Gaussian-based wake model for yawed turbine developed in Qian and Ishihara (2018). The default values recommended in the original study (Qian and

Ishihara, 2018) are used for the model parameters. More description of model can be found in 2.2.5. To combine the effects of multiple wakes, a wake superposition method called "SOSFS" Katic et al. (1986) is implemented. Based on the provided rotor speed and blade pitch measurement, as well as on the model-estimated wind speed, the $C_P$ and $C_T$ values are calculated by interpolating within a Look-Up Table (LUT) containing $C_P$ and $C_T$ scheduled as function of wind speed, tip speed ratio and pitch angle. This LUT was obtained through simulations performed with the provided FASTv8 model. Power production of

yawed turbine is instead computed using equation 1, with yaw loss exponent set as $n = 1.88$, thus equal to the value adopted in Gebraad et al. (2016).

### 3.5 Single Full-Wake Results

At first we compared the main features of the measured and predicted wake profiles at hub height for a single G1 operating at 6 different yaw misalignment. In this regard, both the experimentally measured and predicted wake profiles were best-fitted to

equation 4 below;

$$U_{\text{wake}}(z) = U_0 - U_1 z - \delta_U e^{-\left(\frac{z-\delta_{\text{wc}}}{\sigma_w}\right)^2}, \tag{4}$$





which superimposes a Gaussian-shaped wake to a linearly varying inflow. From the best-fitted parameters, the following wake main features can be derived:

- Wake deficit at hub height, defined as $(U_0 - \delta_U)/U_0$.

- Wake center position at hub height, *i.e.* $\delta_{\mathrm{wc}}$.

- Wake width at hub height, computed without considering inflow non-uniformity and defined as the distance between the points within the wake whose speed is equal to $0.99U_0$, *i.e.* $w_{\mathrm{width}} = 2\sigma_w \{\log \delta_U - \log [U_0 (1 - 0.99)]\}^{0.5}$.

Figure B1 within Appendix B1 shows the measured wake profiles at hub height together with the corresponding best-fitted curves and the predictions obtained using the following four models implemented by P3:

- The Gauss (P3, "Gauss") and Super-Gauss (P3, "Super-Gauss") models, considering laterally and vertically uniform inflow;

- The Gauss (P3, "Gauss Inflow Map") and Super-Gauss (P3, "Super-Gauss Inflow Map") models, considering the laterally and vertically non-uniform inflow shown in Figure 15(a) and (b).

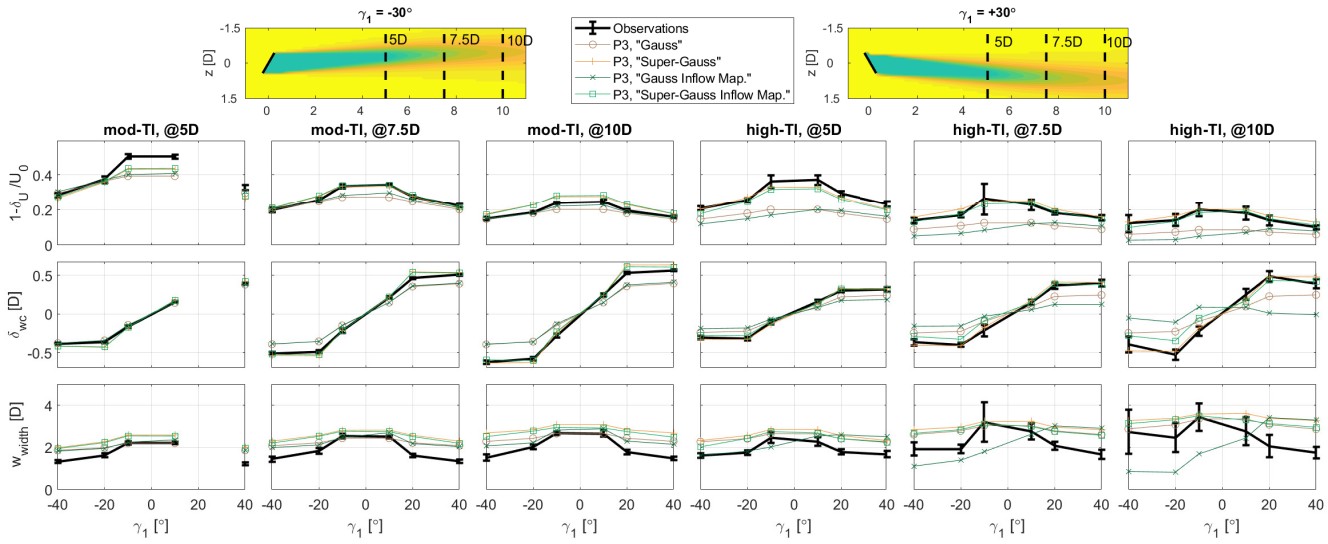

**Figure 16.** Single full-wake results: comparison between the mean features of the measured ('Observations') and model predicted wake profiles by P3 at hub height.

Figure 16 compares the measured and predicted wake main features. The three left columns show the comparisons for the
mod-TI inflow, while the three right column the ones for the high-TI inflow. The error bars of the "Observations" plots are the 95% confidence interval of the corresponding fitted parameters. Generally, the confidence interval is higher as one looks at wake profiles recorded with high-TI inflow and within the far wake. This is quite expected: in such cases the wake deficit





is indeed quite weak, and so the intrinsic measurement noise leads to uncertain estimations of the wake features. In order to retain good readability of the graphs, the plots do not report the 95% confidence interval of the parameters fitted to the
numerical predictions. Finally, the contour plots at the top of Figure 16 were obtained with a Matlab implementation of the FLORIS model developed by Doekemeijer et al. (2018), and only serve to facilitate the interpretation of the figure and clarify the misalignment sign convention for the CL-Windcon wind tunnel blind test.

From the comparison in Figure 16, the following observations can be derived.

– **Wake deficit**: It is generally well-captured by the Super-Gaussian models. For null yaw misalignment, the wake deficit is
slightly under-predicted when looking at 5D, while it is over-predicted at 10D, especially for mod-TI inflow. This seems to indicate that the observed wake recovery rate is higher than the modelled one, which would therefore require a further tuning. For yawed operations, instead, there is overall a better agreement.

– **Wake center position**: The yaw-induced wake deflection is generally very well-captured by both Super-Gaussian models, also within the far wake and for high yaw misalignment. Some mismatch is observed when looking at the far wake
and high-TI inflow. As mentioned before, in such conditions the wake deficit is quite modest, and the uncertain estimation of the wake features could be the base of the observed mismatch.

– **Wake width**: It is generally sufficiently-captured by all models, despite they all tend to slightly overestimate it, especially for mod-TI inflow and at the closest distances from the rotor. Specifically, the overestimation is quite significant for yawed conditions. The reason behind such observation can be traced back to the Gaussian wake shape assumed by the
models. There is indeed experimental (Howland et al., 2016) and numerical (Vollmer et al., 2016) evidence that when a wind turbine is yawed, the wake is not only deflected in a direction opposite to the yaw angle, but its shape also changes from Gaussian-like to elliptic (Martínez-Tossas et al., 2019).

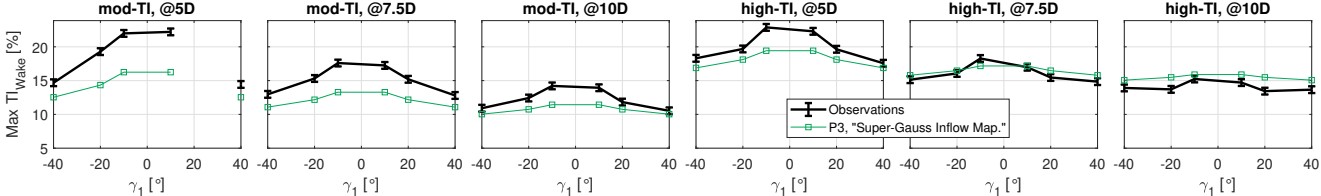

**Figure 17.** Single full-wake results: comparison between the measured and predicted maximum turbulence intensity within the wake at hub height.

Figure B2 within Appendix B1, instead, shows the measured profiles of the longitudinal turbulence intensity at hub height, together with the corresponding predictions. The data reported therein have been used to determine the maximum turbulence
intensity measured within the wake. Figure 17 shows the comparison between such measured quantities and the corresponding values predicted by the sole "Super-Gauss Inflow Map" model, as all models make use of the identical wake added turbulence model. Overall, the models well predict the observed trends, *i.e.* the decrease of the wake added turbulence as the wake recovers



and for misaligned operations. However, the used models tend to under-predict the wake added turbulence, especially for mod-TI inflow and closer to the wind turbine rotor. A proper tuning of the model parameters is likely to improve the match between measurements and predictions.

## 3.6 Multiple Wake Results

### 3.6.1 Wake Profiles Comparison

After comparing measurements and predictions for single wakes, we moved on to comparing measurements and predictions of the multiple wakes shed by two G1s under various inflow and operating conditions. Figures 18 and 19 show the comparison between the measured and predicted wake profiles after the second G1, with aligned and 0.5D laterally shifted 2-turbine configurations, respectively. In both cases, the wakes were traversed 7.5D and 10D downstream of WT1 and along three horizontal lines located at hub height $y_H$, and 0.36D above and below $y_H$. Finally, the y-axes in Figures 18 and 19 show the normalised wind speed in the wake $\bar{U}_{\text{wake}}$, obtained by dividing the measured or predicted data by the pitot wind speed.

The plots within the three left columns of Figure 18 report comparisons related to an operating condition in which both turbines are operated with 30 degrees misalignment ($\gamma_1 = \gamma_2 = 30°$). The plots on the three right columns, instead, pertain to the condition in which the upstream turbine is operated misaligned of 30 degrees ($\gamma_1 = 30°$), while the downstream turbine is kept aligned to the upstream inflow ($\gamma_2 = 0°$). The vertical dashed black line indicates the rotor apex location of both G1s, while the contour plots at the figure top, again obtained with the Matlab implementation of FLORIS, only serve to facilitate the interpretation of the provided results. The following consideration can be drawn from the analysis of the plots.

- $\gamma_1 = \gamma_2 = 30°$: There is generally a good match between the measured wake profiles and the corresponding predictions provided by the Super-Gaussian model, especially when accounting for inflow non-uniformity. On the other hand, looking at the predictions given by the models which assume a uniform inflow, it can be seen that the velocity in the wake is overestimated or underestimated when respectively looking at the wake profiles measured above or below the rotor.

- $\gamma_1, \gamma_2 = [30°, 0°]$: There is generally a relatively poor match between the predictions provided by all models and the measured wake profiles. Once more, the Super-Gaussian model accounting for the inflow non-uniformity seems to provide the closest predictions. Looking at the comparison between the measured and predicted wake profiles at hub height, it appears that all the models fail to predict the wake center position, which is indeed laterally shifted from the rotor center position of WT2, despite the downstream turbine being aligned to the upstream inflow. Specifically, the wake appears to be shifted along a direction equal to the one towards which the wake emitted by WT1 is deflected as a result of its misaligned operation. This observation can partially be traced back to what is referred as "secondary steering" (Fleming et al., 2018). It has been indeed shown in many publications (Wang et al., 2018; Fleming et al., 2018; Howland and Dabiri, 2021; Zong and Porté-Agel, 2021) that the wake shed by a yawed wind turbine induces a deflection of the wake shed by the downstream turbine, even when the downstream turbine itself is not yawed. It can also be attributed to the non-uniform local inflow effect at the downstream turbine. Under the partial wake, the downstream





turbine itself is effectively misaligned, causing a steering of the second wake as observed under the upstream normal
operation by Schepers et al. (2012).

Similarly, the plots within the three left and right columns of Figure 19 respectively report comparisons related to the
following operating condition: $\gamma_1, \gamma_2 = [0°, 30°]$ and $\gamma_1, \gamma_2 = [30°, 0°]$, while vertical dashed black and red lines indicate the
rotor apex locations for the upstream and downstream G1, respectively.

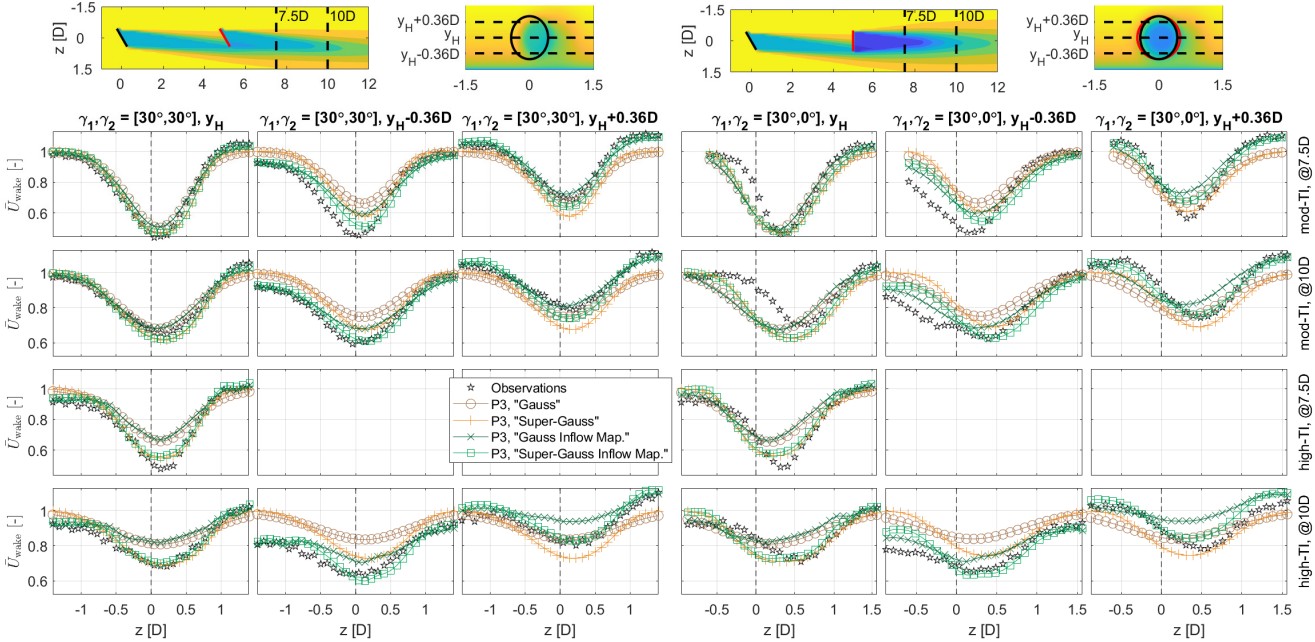

**Figure 18.** Multiple Wake results: comparison between measured and predicted wake profiles with aligned wind turbines.

The considerations that can be drawn from the depicted comparison are quite similar to those previously mentioned. When
only the second turbine is yawed, (*i.e.* with $\gamma_1, \gamma_2 = [0°, 30°]$), there is generally quite a good match between the predic-
tions provided by the Super-Gaussian model and the measured wake profiles, especially when accounting for the inflow
non-uniformity. When instead only the first turbine is yawed (*i.e.* with $\gamma_1, \gamma_2 = [30°, 0°]$), the match between measured and
predicted wake profiles is again quite poor. Moreover, all models tend to overestimate the wind speed in the wake (thus under-
estimating the wake deficit) shed by the lower half of the rotor, *i.e.* for $y_H$ and $y_H - 0.36D$.



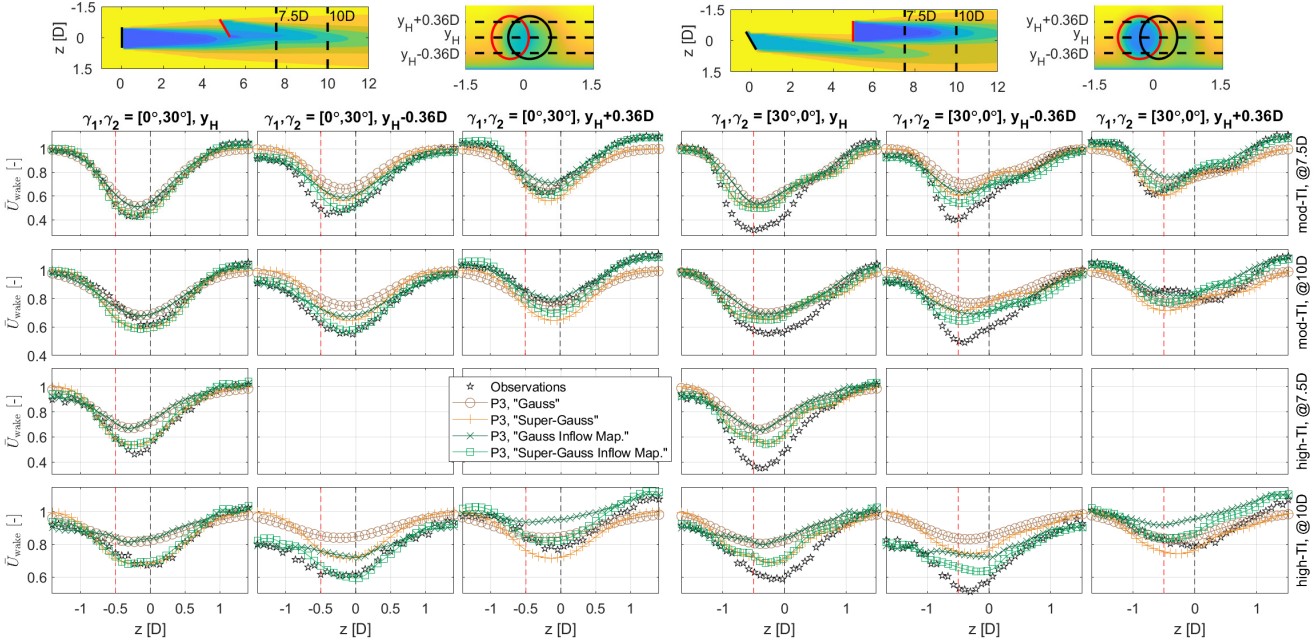

**Figure 19.** Multiple Wake results: Comparison between measured and predicted wake profiles with 0.5D laterally shifted wind turbines.

In order to quantify the mismatch between measured and predicted wakes, Figure 20 shows the distribution of the average error $\epsilon_{\text{wake}}$. For each investigated combination of inflow and wind turbine operating conditions, as well as for each of the two downstream locations at which data were recorded, the average error is computed as the normalised mean of the vector difference between all predicted and measured speed within the wake. Positive values of $\epsilon_{\text{wake}}$ therefore roughly indicate that

the corresponding model under-predicts the average wind speed within the wake, while the wind speed measured by the pitot tube 3D upstream of WT1 was used for the normalisation. Overall, it can again be noticed that the Super-Gauss model provides the closest estimation of the average wind speed within the wake, and that slightly better predictions are sometimes obtained when also accounting for the inflow non-uniformity. Focusing on the error observed with the aligned turbines (Figure 20(a)) and at 10D, *i.e.* at the position of the third turbine within the tested cluster (see Section 3.1), it is possible to derive the following

observations.

– **Mod-TI**: All models under-predict the average wind speed within the wake, with significant disparities when only the upstream turbine operates misaligned. In such a situation, therefore, the models underestimate the beneficial effect due to secondary steering in terms of possible power gain obtainable by misaligning the upstream turbine only.

– **High-TI**: All models slightly over-predict the average wind speed within the wake, and apparently the errors are quite

similar whether only the downstream or both of the turbines are yawed.





Similar conclusions can be drawn with 0.5D laterally shifted turbines (Figure 20b), as higher errors are observed when yawing solely the upstream G1 rather than the downstream, which can again be traced back to the under-representation of the secondary steering effects.

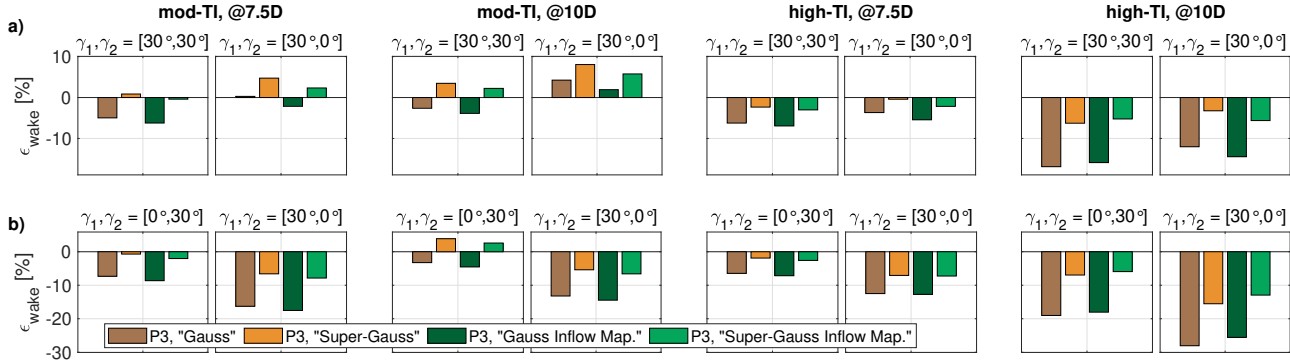

**Figure 20.** Multiple Wake results: Non-dimensional mean error $\epsilon_{wake}$ between measured and predicted wake data with aligned (a) and 0.5D laterally shifted (b) wind turbines.

Figures B3 and B4 within Appendix B2, instead, show the measured profiles of the longitudinal turbulence intensity, together
with the corresponding predictions. Similarly to the single wake cases, the used models tend to under-predict the wake added turbulence, specially for mod-TI inflow and closer to the wind turbine rotor.

### 3.6.2  Power Productions Comparison

The next step was to compare predicted and measured power during tests conducted with three G1s, and with the two upstream machines operated at a wide combination of yaw misalignment. For each of the considered five models described in Section 3.4,
Figure 21 reports the comparison between measured data ($P^{\mathrm{Meas}}$, reported on the x-axis) and corresponding predicted data ($P^{\mathrm{Model}}$, reported on the y-axis). Comparisons for each of the three turbines are reported in the three upper rows, while the lower one depicts the comparisons for the overall cluster (WF) power production. Each subplot contains the corresponding root mean square error normalised with the rated power, equal to 46 W and 138 W respectively for the isolated wind turbine and the considered cluster. Moreover, different symbols are used for different inflow conditions (*i.e.* inflow turbulence and
wind direction). Finally, the contour plots at the figure top depict the flow within the cluster predicted by the adopted Matlab implementation of FLORIS, only to serve as a representation of the investigated layouts.





**Figure 21.** Multiple Wake results: Comparison between measured and predicted power for each of the considered five models.

The following conclusions can be drawn from the analysis of Figure 21.

- **WT1**: Overall, all models tend to under-predict the produced power. However, a better agreement is seen when the inflow is considered non-uniform. The improvement is particularly significant for $\Phi = -6.9°$. A rotation of the turntable causes a lateral displacement of the machines located at a greater distance from the center of rotation, *i.e.* WT1 and WT3, which are therefore laterally shifted with respect to the position of the pitot placed upstream of the cluster. Due to the non-uniformity of the inflow, the wind speed measured by the pitot differs from the average rotor speed at WT1 and WT3. By considering the non-uniformity of the inflow, the power predictions for the upstream machine can thus become more accurate. The remaining mismatch could still be due to the discrepancies between the modelled and true inflow, as well as the improper modelling of the yaw-induced power losses.



– **WT2**: As a whole, all models tend to overestimate the produced power. The best predictions are observed when using the Super-Gauss model, in agreement with the findings discussed in Section 3.5. The observed mismatch can be related to potential deficiencies in modelling the main features of the wake shed by WT1 (see Figure 16), and/or inability to capture the variation of the yaw-induced power losses with the rotor effective wind speed, as suggested by Simley et al. (2021).
The results reported in Figure 21, indeed, comprise cases in which also WT2 is strongly misaligned while operating at a different rotor effective wind speed than WT1.

– **WT3**: Generally, it can be seen that the predictions made by all the models differ substantially from the observations, and in many cases the produced power is underestimated. Specifically, the biggest difference is found when the inflow is moderately turbulent (mod-TI) and when the downstream machines are fully immersed in the wake emitted by the
upstream ones ($\Phi = 0°$). The match is considerably better if one looks at the data measured when the downstream machines are partially impinged by a wake ($\Phi = -6.9°$). In this latter case, indeed, the average rotor speed is only partially affected by the wake presence. An erroneous modelling of the wake has, therefore, a smaller impact on the estimation of the power produced by the downstream turbine compared to full wake conditions with $\Phi = 0°$.

– **WF**: Notwithstanding the differences observed between measurements and predictions at the single turbine level, it
is very interesting to notice that the discrepancies at the cluster level are on the whole much less. This means that the overestimated power produced by WT2 is generally compensated by the underestimation of the power produced by WT3 or WT1. Finally, in agreement with the previous findings, the Super-Gauss model provides the most accurate predictions, especially if the non-uniformity of the inflow is accounted for.

### 3.6.3    Quantities of Interest: Power Gain

At last, we compared the power gain obtainable, at wind farm level, by applying wake steering by yawing to the first two machines of the tested 3-turbines cluster. Specifically, for the four considered inflow conditions all the provided numerical and experimental data sets were searched for the combination of yaw misalignments that maximises the overall cluster power, thus using equation 5;

$$[\gamma_1^*, \gamma_2^*] = \arg\max_{\gamma_1, \gamma_2} \bar{P}_{\text{WF}}, \tag{5}$$

with $\gamma_1^*$ and $\gamma_2^*$ the optimal yaw misalignment for WT1 and WT2, respectively, while $\bar{P}_{\text{WF}} = P_{\text{WF}} / \left(1/2\rho A U_{\text{Pitot}}^3\right)$ is the cluster power normalised by the available free-stream wind power, *i.e.* half of the product of the air density $\rho$, the rotor disk area $A$ and the cube of the wind speed measured by the pitot tube.

The center and right plots of Figure 22 compare the experimental and numerically-predicted optimal yaw misalignment control settings, while the left plot depicts the comparison between the corresponding power gains computed via equation 6;

$$P_{\text{GAIN}} = \left( \frac{\bar{P}_{\text{WF}}^{\text{WFFC}}}{\bar{P}_{\text{WF}}^{\text{Normal Operation}}} - 1 \right) \cdot 100, \tag{6}$$



with $\bar{P}_{\text{WF}}^{\text{WFFC}}$ and $\bar{P}_{\text{WF}}^{\text{Normal Operation}}$ the normalised cluster powers with optimal yaw control settings and null yaw misalignment, respectively. Error bars, computed by adding fractional uncertainties in quadrature, are associated to the measured gains; to this aim, the measured cluster power was assumed to be affected by an uncertainty equal to 0.98% of the available free-stream wind power (Campagnolo et al., 2020).

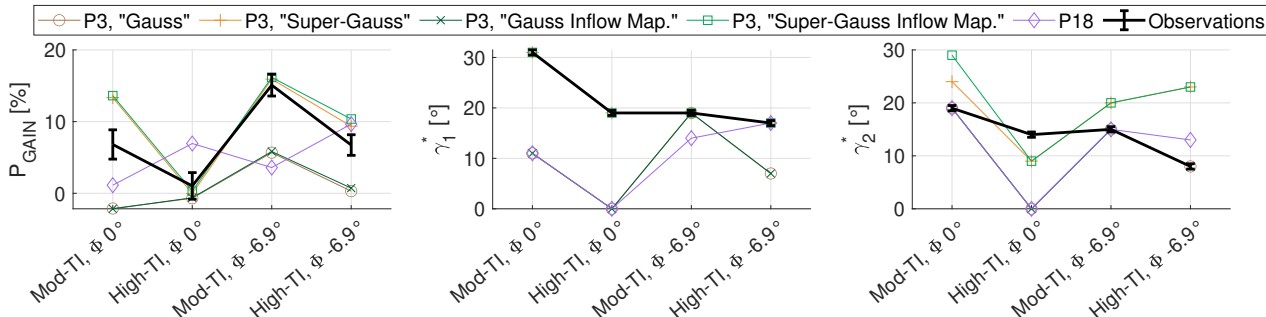

**Figure 22.** Multiple Wake results: comparison between measured and predicted power gain of wake steering wrt normal operation (left); comparison between the corresponding measured and predicted optimal yaw misalignment for WT1 (center) and WT2 (right).

The following conclusions can be drawn from the analysis of the figure plots:

– **Power gain**: As expected, the highest gains are observed when the machines operate in partial wake conditions ($\Phi = -6.9°$) and when the inflow is moderately turbulent (mod-TI). Moreover, as previously noted, the Super-Gauss model provides the most accurate predictions in terms of expected gains, with marginal differences depending on whether or not non-uniformity of the inflow is accounted for. A significant difference between measured and predicted gains is observed
only for moderately turbulent inflow and when the downstream machines are completely immersed in the wake shed by the upstream turbines, *i.e.* for the scenario characterised by the greatest mismatch between numerical and experimental results (see Figure 21).

– **Optimal yaw misalignment**: WT2 optimal yaw control set-points are lower than WT1 ones, similar to the observations in other recent wind tunnel campaigns (Bastankhah and Porté-Agel, 2019). Interestingly, the Super-Gauss model predicts
very well the optimal yaw misalignment of WT1 for all the considered inflow conditions, but tends to overestimate the optimal yaw settings of the second wind turbine. The reason behind this observation could be that the used Super-Gauss model does not account for the additional wake deflection after WT2 induced by secondary steering, as observed by King et al. (2021).

### 3.7 Summary of the CL-Windcon Wind Tunnel Blind Test

The results previously discussed can be summarised as follows:

**Reasonable predictions despite none of the models were ad-hoc calibrated** None of the participants used the calibration set to tune the parameters of the corresponding models. Despite this, the numerical predictions of the velocity profiles of





both single and multiple wakes, as well as the estimates of the power produced by the various machines in the cluster, are reasonably close to the corresponding experimental values. A significant difference is instead observed between the
predictions and the measurements of the wake turbulence profiles. These evidences on one hand confirm the robustness of the sub-models used to predict the wind speed in the wake, while on the other hand they highlight the need for a better tuning of the wake-added turbulence sub-models. As also pointed out in Section 2.6 for the SMV WF field data blind test, a dedicated site-specific calibration of the model parameters could significantly improve the numerical-experimental agreement of the results.

**Wake modelling can be improved** Overall, the Super-Gauss model, *i.e.* the best performing model among those used by the participants in the benchmark, reproduces with good accuracy the velocity deficit and the position of the wake shed by a single turbine operating at different misalignment angles. In the case of a strongly misaligned turbine, however, the Gaussian shape of the wake assumed by the model leads to an overestimation of the actual wake width. This shortcoming could be addressed by an appropriate modelling of the wake that captures its characteristic curled shape, as performed
by *e.g.* Bastankhah et al. (2022). Regarding the modelling of the multiple wake shed by two upstream turbines, it can be seen that the models that take into account the non-uniformity of the inflow generally provide more accurate predictions. However, large differences between experimental and numerical results can be detected when only the upstream machine is operated misaligned with respect to the upstream flow, while the second turbine is kept aligned. None of the models used by the participants, indeed, account for the effects caused by secondary steering, which induces a deflection of the
wake after the second machine, although it operates aligned to the upstream flow.

**Power production at cluster level and power gains are generally accurately predicted** Similar to the observations of the SMV WF field data blind tests in Section 2, there is generally a good agreement between experimental and numerical results concerning the power produced by the first two machines of the cluster. However, significant discrepancies are observed at the power produced by the third turbine. Possible explanations could be the lack of modelling of the sec-
ondary steering effects and/or an inappropriate modelling of the wakes superposition. However, at the cluster level the agreement between numerical and experimental results is very good, especially when the non-uniformity of the inflow is accounted for. Consequently, the gains obtained by the investigated wake steering WFFC settings are also predicted with sufficient accuracy, at least for the majority of the considered inflow conditions.

## 4 BLIND TEST #3: CL-Windcon LES

This blind test builds upon a set of high-fidelity simulations performed in the context of the European R&D project CL-Windcon (CL-Windcon, 2016). They were performed with NREL's open-source SOWFA simulation package NREL (2012), which is a set of computational fluid dynamics (CFD) solvers based on OpenFOAM (2013), boundary conditions, and turbine models represented through OpenFAST simulation tool (Jonkman et al., 2017). The scenarios consist of 2 different layouts (see Figure 23), for the 10 MW INNWIND.EU reference turbine (INNWIND.EU, 2013): *i*) 3 turbines with 3×1 configuration



(3WT), 5D distance among turbines where the third turbine has 0.5 diameters (D) lateral displacement with respect to the flow and, *ii*) 9 turbines with $3\times3$ configuration (9WT), 7D distance among the turbines. Further details on the simulations setup can be found in (Gomez-Iradi et al., 2019; Doekemeijer et al., 2019).

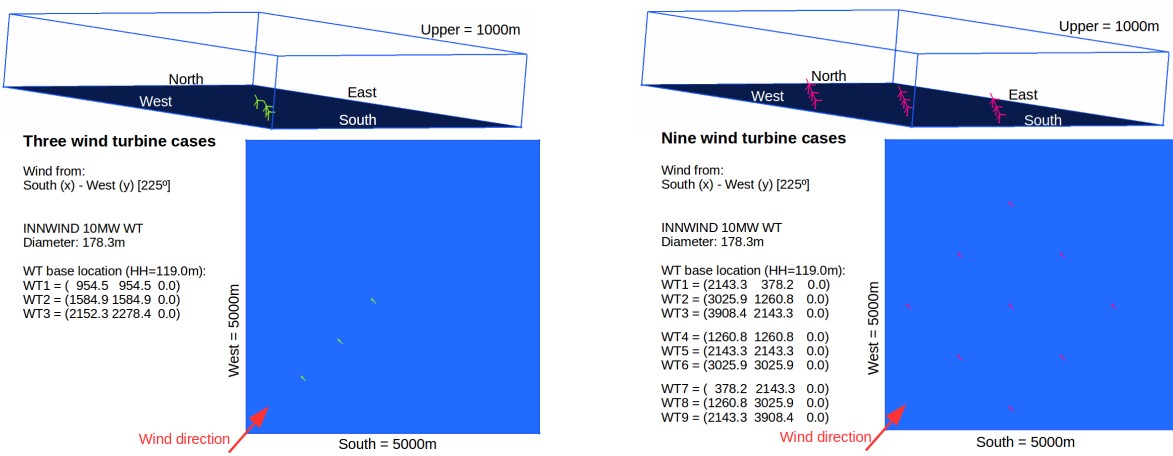

**Figure 23.** CL-Windcon LES layouts **Left:** 3 WT scenario. **Right:** 9 WT scenario.

For the whole data set, combinations of 3 wind speeds, turbulence intensities and roughness lengths were applied together with varied wind turbine control settings. However, for the specific blind test in FarmConners benchmark, an inflow corre-
sponding to an average wind speed of 7.7 m/s, turbulence intensity of 5.39%, and roughness length of 0.001 m was selected (referred as A4 in the original database (CL-Windcon, 2019)). CL-Windcon LES blind test is among the most diverse and comprehensive in terms of the wake steering control strategies applied within the FarmConners benchmark. The yaw misalignment was varied in the range of $\pm30°$ for the first and/or the second row of turbines, see Table 8 for particular control settings among 3WT and 9WT scenarios. The direction for the yaw convention, as well as the distribution of the control set-points among the
turbines are illustrated per investigated case in the presentation of the results later in Sections 4.2 and 4.3.

Targeted test cases explore single full wake, as well as multiple wake under wake steering WFFC. Available data covers power production, hub-height and/or rotor effective wind speeds at the turbines, as well as several structural loading variables namely flap-wise root bending moment, total shaft bending moment and total tower bottom bending moment. However, the load channels are excluded from the analysis in this study and the participating models are evaluated based on the reported
power production with and without WFFC.





| 3WT Scenario | | | | | | | | | | | |
|---|---|---|---|---|---|---|---|---|---|---|---|
| CALIBRATION | Y000 | Y000_Y-20 | Y020 | | Y-20 | | Y-20_Y-10 | | Y-20_Y-20 | | Y-20_Y-30 |
| BLIND | Y000_Y-10 | Y000_Y-30 | Y010 | Y030 | Y-10 | Y-30 | Y-10_Y-10 | Y-30_Y-10 | Y-10_Y-20 | Y-30_Y-20 | Y-10_Y-30 | Y-30_Y-30 |

| 9WT Scenario | | | |
|---|---|---|---|
| CALIBRATION | Y000 | | Y-20_Y123 |
| BLIND | Y-123 | Y-10_Y123 | Y-30_Y123 |

**Table 8.** Yaw misalignment control settings for the calibration and blind test for CL-Windcon LES cases. Yxxx: the downstream turbines are not misaligned where xxx[°] indicates the yaw misalignment of the upstream turbine(s) (WT1 for 3WT scenario; WT1, WT4, WT7 for 9WT scenario). Yxxx_Yyyy: upstream and the first downstream turbines are misaligned, where xxx[°] and and yyy[°] indicates the yaw misalignment of the upstream and downstream turbines, respectively. For 9WT scenario, xxx or yyy=-123 indicates -10°, -20° and -30° yaw setting at turbines WT2, WT5 and WT8, respectively.

## 4.1 Participating Models

Within the FarmConners benchmark, CL-Windcon LES blind test has had 4 participating models in total (IDs = P11, P12, P16, P19). Two steady-state and two (quasi-)dynamic models have been utilised for the exercise, covering a wide range of model fidelity. Table 9 lists their main characteristics for an easy comparison, especially for the lower fidelity models. Further details of the implemented models are presented in the following sections.

| | **P11** | **P12** | **P16** | **P19** |
|---|---|---|---|---|
| **Wind Farm Model** | EllipSys3D-RANS-AD[a] | FLORIDyn – revised[b] | – | FRED[h, i] |
| **Flow Model** | RANS | Gauss-Legacy[c, d] | Gaussian[c] | Discretized Navier-Stokes[i] |
| **Added Turbulence Model** | – | Crespo-Hernandez[e] | Frandsen[g] | – |
| **Wake Superposition** | – | Geometric Sum[f] | Quadratic | – |
| **Wake Deflection** | – | Gaussian[c] | Gaussian[c] | – |
| **Time-Series** | No | Yes | No | Yes |

**Table 9.** Overview of the participating WFFC-oriented models, CL-Windcon LES Blind Test. [a](Sørensen, 1995), [b](Becker, 2020; Becker et al., 2022), [c](Bastankhah and Porté-Agel, 2016), [d](Bastankhah and Porté-Agel, 2014), [e](Crespo and Hernández, 1996), [f](Shao et al., 2019), [g](Frandsen, 2007) – IEC 2019 standard, [h](van den Brooek, 2021), [i](Van Den Broek and van Wingerden, 2020)

### 4.1.1 P11

P11 uses PyWakeEllipSys (DTU Wind Energy, 2021), which consists of the elliptic RANS solver, EllipSys3D (Sørensen, 1995). The numerical setup is similar to the one in Larsen et al. (2020), where the wake deflection was also studied. The turbines are modelled by Joukowsky AD's (no nacelle nor tower are modelled) and the disk averaged normal velocity is



used to control each turbine using 1D momentum theory. The wind turbine data needed for the AD model, *i.e.* $C_T(U_{H,\infty})$, $C_P(U_{H,\infty})$ and $TSR(U_{H,\infty})$, are taken from the DTU 10 MW report by Bak et al. (2013).

Turbulence is modelled with the $k$-$\varepsilon$-$f_P$ closure of van der Laan et al. (2015) and the inflow follows neutral atmospheric surface layer profiles. These profiles are prescribed to match the freestream velocity and total turbulence intensity at hub height: $U_{H,\infty} = 7.7$ m/s and $I_{H,\infty} = 5.4\%$ (the "A4" wind case of the CL-Windcon campaign used in the FarmConners benchmark).

### 4.1.2 P12

The parametric WFFC model used for this benchmark study by P12, also detailed in Becker (2020); Becker et al. (2022), will be referred to as FLORIDyn (FLOw Redirection and Induction Dynamics model). The central idea of FLORIDyn is to approximate the dynamic wake behaviour of wind turbines in a wind farm with low computational cost by piece-wise updating the steady state flow field with a new steady state description which fits the new states. This update from the precursor FLORIS

model, see Doekemeijer et al. (2020), is driven by Observation Points (OPs), which are created and updated at the rotor plane for each time step. They represent the influence of the turbine state travelling downstream. Within FLORIDyn, the steady state wake is modelled via FLORIS (NREL, 2021) which is then propagated through the wind farm, instantly affecting turbines down stream. For instance, when the yaw angle of the turbine changes, the new generation of OPs will copy the new angle while old OPs still travel according to the previous angle. In the case of overlapping wakes, an OP travels into the wake of

another turbine. It locates the closest up and downstream OPs from the foreign wake and interpolates their reduction factor at its location. The calculation of $C_T$ and $C_P$ is based on the lookup table generated via SOWFA (NREL, 2012) high fidelity simulations for the reference turbine in the blind test. For the current benchmark study, the statistical properties of the wind field are matched (mean wind speed and turbulence intensity), however, no specific calibration to match to the CL-Windcon LES wake data was performed. Calibration using uncertainty quantification is intended to form part of a future publication.

### 4.1.3 P16


For the P16 results, the same in-house wake modelling tool as described in 2.2.4 are used. It consists of a Gaussian-based wake model (Bastankhah and Porté-Agel, 2016) with a quadratic single-wake superposition model. Based on the calibration data provided, the wake model parameters $k_a, k_b, \alpha_d$ and $\beta_d$ are also optimised with the SGA optimiser of the "pygmo" library (Biscani and Izzo, 2020) and the same loss function, *i.e.* the sum of the average quadratic differences between the calibration

states power and simulated power at every turbine of the wind farm. However, the optimisation procedure is different compared to the SMV wind farm field data blind test. Instead of using a single global wake model parameter for all turbines, here each turbine has independent wake model parameters which are determined by the optimiser. The power loss and the dependency of $C_T$ on the yaw angle are modelled by factors $(\cos\Psi)^n$, with the same values for $n = 1.88$ as described in Section 2.2.4 and equation 1. For the power and thrust curves a lookup table that was derived from blade element momentum (BEM) model of

the 10 MW DTU reference turbine is used.



### 4.1.4 P19

The dynamic WFFC model used for this benchmark study by P19, currently under development (van den Brooek, 2021), will be referred to as FRED (Framework for wind farm flow Regulation and Estimation with Dynamics). For the sake of brevity, a concise description is presented here, where interested readers may refer to Van Den Broek and van Wingerden

(2020). The model is based on Navier-Stokes equations that are discretized in time and finite-element method that is used for spatial discretisation with the Taylor-Hood element (Wieners, 2003). Dirichlet boundary conditions (Givoli and Keller, 1989) prescribe the inflow velocity. The inflow boundaries are dynamically chosen based on the wind direction. For example, for the current blind benchmark case where the wind flows from south-west direction, the south and west boundaries are marked as inflow. The other boundaries are given Neumann conditions (Givoli and Keller, 1989), by default. The model is initialised with

a uniform flow-field given by initial velocity and a constant pressure. A generalised mixing length (Morgan et al., 1977) model is used to model the sub-grid-scale eddy viscosity. The wind turbine forcing on the flow is approximated using an actuator disk model. For the current benchmark study, the statistical properties of the wind field are matched (mean wind speed and turbulence intensity), however, no specific calibration to match to the CL-Windcon LES data was performed. Calibration is intended to form part of our future works.

## 4.2 Single Wake Results

To have a generic comparison for both the single and multiple wake results for 2 time-series and 2 steady-state results submitted, exclusively the mean quantities of interests estimated by the participating models are presented in the CL-Windcon LES blind test results. Accordingly, the mean power per turbine, $P_{Ti}$, submitted by the participants under WFFC and Normal Operation are used to calculate the power difference per turbine, $\Delta P_{Ti}$, and power gain, $P_{\text{GAIN}}$, by equation 7 and 8, respectively;

$$\Delta P_{Ti} = P_{Ti}^{\text{WFFC}} - P_{Ti}^{\text{Normal Operation}} \tag{7}$$

where $P$ is the power and $Ti$ is the turbine ID within the wind farm configuration (T0 for upstream, T1 for downstream in the single wake case).

$$P_{\text{GAIN}} = \frac{\left(\Sigma P_{Ti}\right)^{\text{WFFC}}}{\left(\Sigma P_{Ti}\right)^{\text{Normal Operation}}} - 1 \tag{8}$$



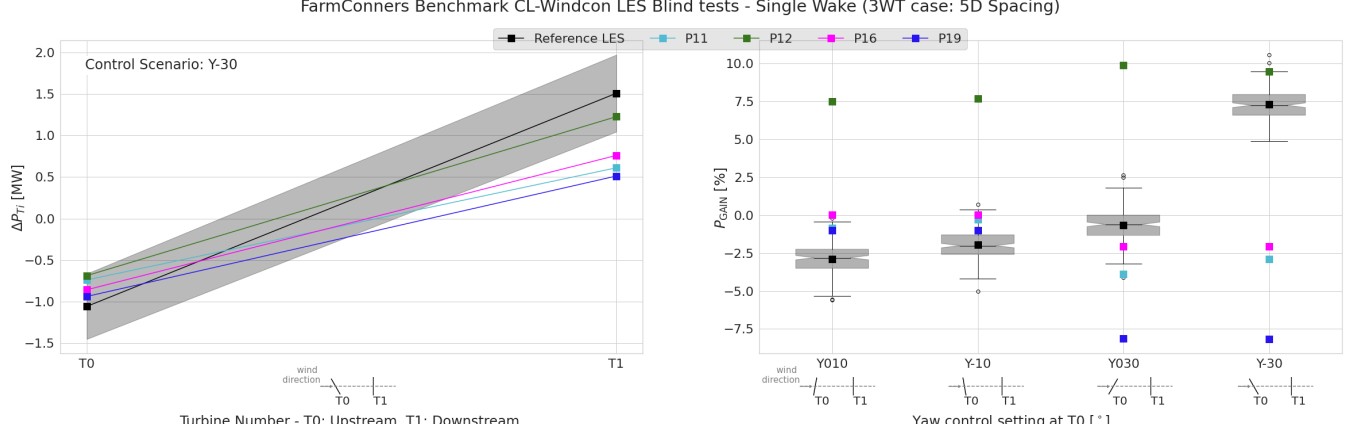

**Figure 24.** CL-Windcon LES blind tests, Single Wake under Wake Steering for 3-turbine configuration (3WT) with 5D spacing. **Left:** Power difference (mean) in absolute values [MW] per upstream (T0) and downstream (T1) turbines, when T0 is -30° misaligned (Y-30). **Right:** Power gain (mean) for the 2-turbine subset of the wind farm (T0 + T1). Boxplots represent the distribution of the reference CL-Windcon LES time-series per control setting where T0 is steered +10° (Y010), -10° (Y-10), +30° (Y030) and -30° (Y-30). Representative layouts with corresponding yaw control settings are illustrated along the x-axes.

Figure 24 shows $\Delta P_{Ti}$ and $P_{\text{GAIN}}$ for a single wake with 5D spacing where the upstream turbine, T0, is misaligned +10°
(Y010), -10° (Y-10), +30° (Y030) and -30° (Y-30). For $P_{\text{GAIN}}$ in Figure 24, the boxplots represent the distribution of the reference high-fidelity simulations at each control setting, where the mean $P_{\text{GAIN}}$ estimated by the participating models are illustrated on top. For ±10° misalignment at T0, CL-Windcon LES shows that the likelihood of power loss (up to more than 5%) is higher than the gain for the investigated 2-turbine configuration. This is relatively well captured by P11, P16 and P19, where P12 overestimates the potential of the control strategy. However, for larger upstream misalignment, the agreement
between the participating models and the validation data set generally declines. For +30° steering at T0 (clock-wise rotation), the low likelihood of the power gain is closely estimated by P16 where P11 and P19 indicate over-estimation of the wake losses and the general trend of high gain predictions from P12 continues. The asymmetry in the wake behaviour is significant when investigating -30° steering set-point, which is likely to be caused by wake rotation and/or wind veer combined with the upstream misalignment. This is, although less pronounced, also observable for ±10° and not represented by the participating
models for either of the control settings. For the -30° upstream steering the potential of the power gain exceeds 5% in CL-Windcon LES results which is relatively well captured by P12 and significantly underestimated by the other models. These rather *pessimistic* gains reported at -30° steering can be further analysed by $\Delta P_{Ti}$ in Figure 24. There, it is seen that the upstream power loss is represented fairly well by all the participating models (within ±1 standard deviation bounds), but the wake losses behind a misaligned turbine is overestimated by the majority of the models, except of P12. This can potentially
be explained by the less wake deflection produced by the models, as *e.g.* studied previously for P11 (Larsen et al., 2020). The same trend is visible in the other high-fidelity simulations within the benchmark as well, as can be seen in TotalControl LES blind test in Section 5.2, for high degrees of upstream misalignment in P11 and P16 results.



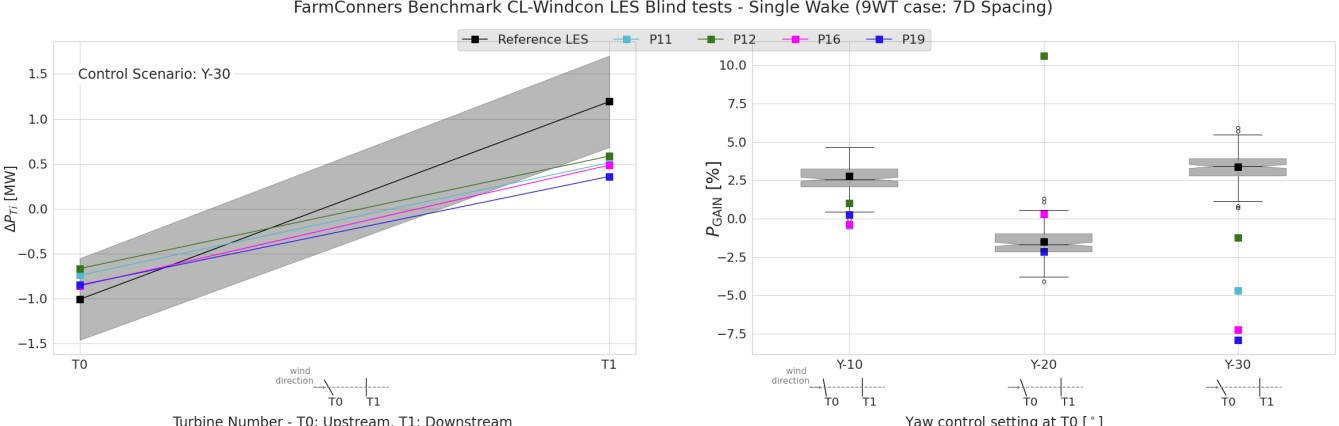

**Figure 25.** CL-Windcon LES blind tests, Single Wake under Wake Steering for 9-turbine configuration (9WT) with 7D spacing. **Left:** Power difference (mean) in absolute values [MW] per upstream (T0) and downstream (T1) turbines, when T0 is -30° misaligned (Y-30). **Right:** Power gain (mean) for the 2-turbine subset of the wind farm (T0 + T1). Boxplots represent the distribution of the reference CL-Windcon LES time-series per control settings where T0 is misaligned -10° (Y-10), -20° (Y-20) and -30° (Y-30). Representative layouts with corresponding yaw control settings are illustrated along the x-axes.

Figure 25 investigates $\Delta P_{Ti}$ and $P_{\text{GAIN}}$ for a single wake, this time for 9WT scenario with 7D spacing where the upstream turbine, T0 is misaligned -10° (Y-10), -20° (Y-20) and -30° (Y-30). It should be noted that the control settings presented in here in 9WT single wake cases are the subset of the Y-123 settings under the blind tests reported in Table 8, based on each row of turbines in Figure 23. $P_{\text{GAIN}}$ in Figure 25 shows an interesting trend where -10° and -30° upstream misalignment result in positive power gain but -20° indicates a potential power loss for the investigated 2-turbine configuration in the reference CL-Windcon LES results. Compared to 5D spacing in Figure 24, -10° upstream misalignment produce more power at 7D downstream turbine, hence the positive power gain which is slightly underestimated by the participating models. The agreement for -20°, however is significantly better for the majority of the models which could be due to the same upstream control setting in the calibration data set, though the downstream yaw setting(s) are different (*i.e.* Y-20_Y123 in Table 8).The upstream power loss due to misalignment is more than compensated for -30° yaw setting in CL-Windcon LES results, as also seen in $\Delta P_{Ti}$ behaviour. However the downstream power gain is significantly underestimated by all the models, as also observed and discussed for 5D spacing in Figure 24 as well as under similar upstream control settings applied in TotalControl LES in Section 5.2 where P11 and P16 also participated.

### 4.3 Multiple Wake Results

Similar to the single wake results, the mean power estimated by the participating models are compared with the CL-Windcon LES blind test database for the multiple wake cases. Accordingly, the power difference per turbine and the wind farm level power gain via equations 7 and 8 are presented on Figures 26 and 27. The turbine IDs for multiple wake cases, however, are WT1, WT2, WT3 for 3-turbine configuration and WT1, WT2, ..., WT9 for 9-turbine configuration. The representative layouts





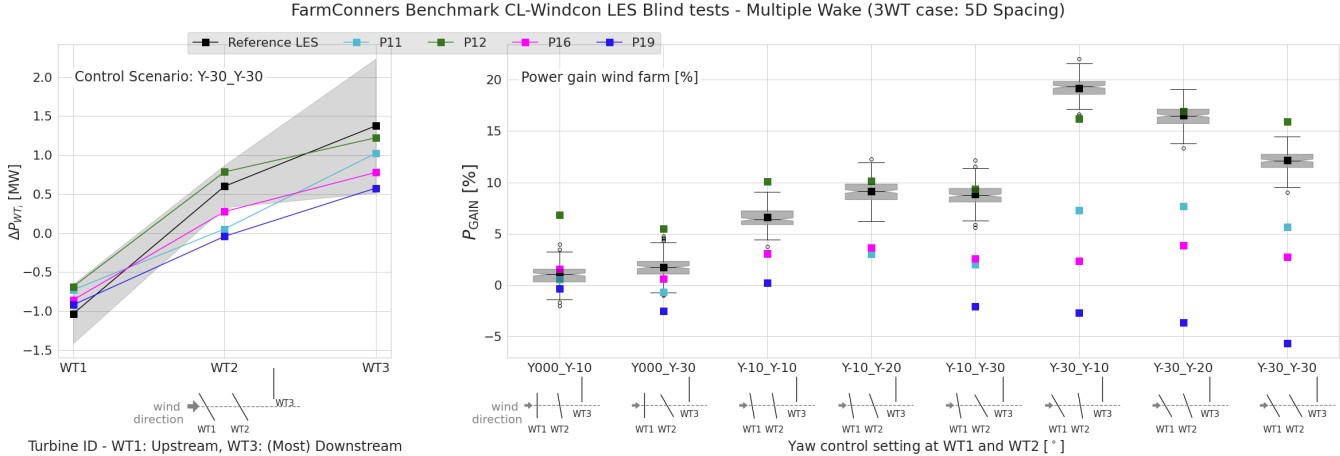

**Figure 26.** CL-Windcon LES blind tests, Multiple Wake under Wake Steering for 3-turbine configuration (3WT) with 5D spacing. **Left:** Power difference (mean) in absolute values [MW] per turbine, when WT1 and WT2 are both $-30°$ misaligned (Y-30_Y-30). **Right:** Power gain (mean) per control setting in 3-turbine wind farm. Boxplots represent the distribution of the reference CL-Windcon LES time-series per control setting listed in Table 8. Representative layouts with corresponding yaw control settings are illustrated along the x-axes.

Figure 26 shows $\Delta P_{Ti}$ and $P_{\text{GAIN}}$ for 3-turbines with 5D spacing (where WT3 is laterally 0.5D apart) for several upstream and (first) downstream yaw control settings for the CL-Windcon blind tests listed in Table 8. For the investigated configuration, Figure 26 indicates that the power gain is mainly driven by the control setting applied at the upstream turbine, WT1, where the misalignment of WT2 has relatively lower impact. This is in line with other LES studies that investigate multi-turbine wake steering (*e.g.* Archer and Vasel-Be-Hagh, 2019). Especially with the lateral spacing of WT3, all the control cases where WT1 is misaligned indicate positive power gain up to more than 20% in CL-Windcon LES runs. For all the participating models except P12, this trend is highly underestimated and the agreement gets gradually worse with higher wake deflection. P12, however, estimates the behaviour relatively well, especially when WT2 is also misaligned. This can potentially be attributed to the high levels of wake deflection embedded in P12 model parameters. $\Delta P_{Ti}$ enables a closer look at the power difference at the individual turbine level for the highest degrees of control settings at WT1 and WT2, -30° each. There, similar to the single wake analyses, it can be seen that the upstream losses observed in reference CL-Windcon LES at WT1 is relatively well represented by the participating models but potential gain at the controlled downstream turbine (WT2) is underestimated by the majority. In addition to the under-representation of the wake deflection compared to the CL-Windcon LES; the difference in the power yaw loss exponent, $n$ in equation 1, for the misaligned turbine in the wake can also play a role here, as demonstrated in (Liew et al., 2020b). For the laterally spaced most downstream (WT3) turbine, the overall behaviour in the power performance is seemingly captured by all the participating models. However, significantly higher fluctuation in the expected power gain at WT3 should also be noted with approximately $\pm 1$ MW.

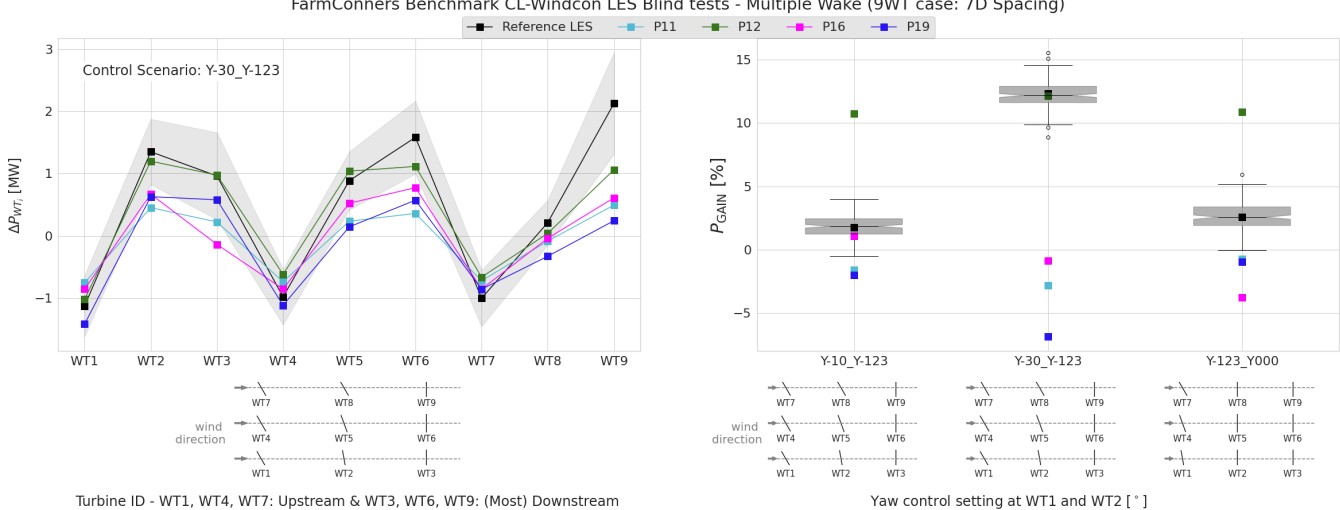

**Figure 27.** CL-Windcon LES blind tests, Multiple Wake under Wake Steering for 9-turbine configuration (9WT) with 7D spacing. **Left:** Power difference (mean) in absolute values [MW] per turbine, for the control case Y-30_Y123 as shown in Table 8. **Right:** Power gain (mean) per control setting in 9-turbine wind farm. Boxplots represent the distribution of the reference CL-Windcon LES time-series per control setting listed in Table 8. Representative layouts with corresponding yaw control settings are illustrated along the x-axes.

Figure 27 illustrates $\Delta P_{Ti}$ and $P_{\mathrm{GAIN}}$ for 9-turbine configuration with 7D spacing in regular layout consisting 3 rows with WT1, WT4 and WT7 as upstream turbines. Similar to the previous results of the CL-Windcon LES blind test, the highest yaw control setting, at the upstream turbines (Y-30_Y123) is observed to produce the highest $P_{\mathrm{GAIN}}$ in the validation data set, up to 15% for the 9-turbine layout. Again, underestimated by all the participating models except of P12; this time at higher discrepancies with up to 20% less in the mean values. $\Delta P_{Ti}$ illustrates the disagreements in the power difference per turbine

for that control setting; where the upstream power loss at -30° steered turbines WT1, WT4 and WT7 are well reproduced. The error in the most downstream power predictions, however, is significantly higher for all except P12 at the turbines WT3, WT6 and WT9. This is also in line with the behaviour previously discussed for the other configurations in the blind test where the upstream yaw setting is -30°. As opposed to P12, the wakes behind less steered turbines (Y-10_Y-123 and Y-123_Y000) are better captured by P11, P16 and P19 with overall less sensitivity to the yaw setting at the second turbines in the rows, WT2,

WT5 and WT8. This also implies an overall stronger wake deflection reported in CL-Windcon LES reference data set than the majority of the participating models.

## 4.4   Summary of the CL-Windcon LES Blind Test

Due to its capability of representing significant features of wind-farm flows, LES is an ideal choice for proof of concept studies regarding WFFC strategies. On that regard, with many control settings included, CL-Windcon LES blind test provides a broad

overview for the initial comparison of participating models. The highlights can be summarised as:



**Higher steering, higher disparity** Although -20° upstream misalignment was in the calibration data set for the CL-Windcon LES blind tests, the overall participating model agreement is seen to be better for -10° upstream steering than -30°. Further analysis indicate the main cause of this disparity to be due to deflected wake modelling rather than the upstream power loss at the controlled turbine(s). As the reference LES suggest higher likelihood of power gain for those con-
trol settings, the need for better deflection models and/or (even) more comprehensive calibration data set with higher deflection angles should be remarked.

**Most upstream misalignment is the main driver for the expected power gains** Within the FarmConners benchmark, CL-Windcon blind test provides a unique opportunity with its diverse control settings, including the misalignment of the downstream turbine(s). For the investigated layouts with 3×1 and 3×3 turbine configurations, however, it is seen that
the misalignment of the second row of turbines has a relatively lower impact on the expected power gains, compared to the most upstream. Similar results are reported in other LES studies with multi-turbine yaw misalignment (*e.g.* Archer and Vasel-Be-Hagh, 2019).

**Take all the conclusions with a grain of salt** As indicated earlier, high-fidelity simulations are very well suited for the proof of concept studies. Although has many advantages, one of the main drawbacks of the reference CL-Windcon LES data
set is its limited simulation period of 10-min. The impact of steering reported here and the overall model comparison (especially for the steady-state models), therefore, should be read as an initial comparison, not a comprehensive validation. Further discussion on high-fidelity simulations based code comparison is provided in the next section with the TotalControl LES blind test results.

## 5 BLIND TEST #4: TotalControl LES


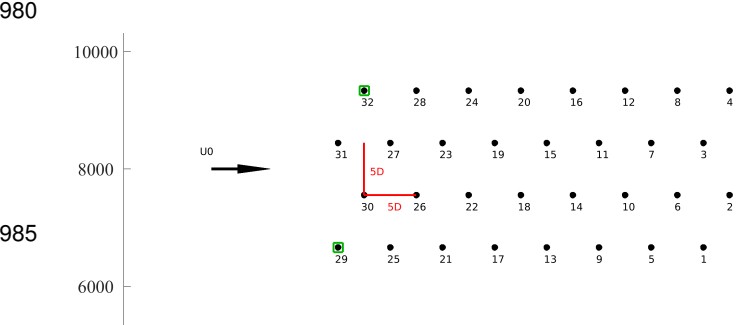

**Figure 28.** Layout of the TotalControl reference wind farm with wind direction 90°. Green squares mark yawed turbines. The turbine IDs are
referred as WT1, WT2, ..., WT32 throughout the rest of the section.

This blind test is based on an extension of high-fidelity simulations performed as part of the Horizon2020 project: TotalControl (TotalControl, 2018). The high-fidelity simulations have been performed using Ellip-Sys3D (Michelsen, 1992, 1994; Sørensen, 1995), which is a finite volume Navier–Stokes solver. The wind turbines are modelled using the Actuator Line method (Sørensen and Shen, 2002; Sørensen et al., 2015), which is fully coupled to the aeroelastic tool, Flex 5 (Øye, 1996).

TotalControl defined a virtual reference wind farm (Andersen et al., 2018), which consists of 32 DTU10MW turbines (Bak et al., 2013) in a $8 \times 4$ with every other row

offset in the streamwise direction. LES were performed for different atmospheric conditions and wind directions, and a subset





of the data is publicly available (Andersen and Troldborg, 2020). The selected case corresponds to a conventionally neutral boundary layer with a geostrophic wind of $G = 12$ m/s, a roughness length of $z_0 = 2 \times 10^{-3}$ m, and a Coriolis parameter

of $f_c = 10^{-4}$ s$^{-1}$, which represents a latitude of $43.43°$, see additional details in (Andersen et al., 2019). The resulting mean velocity at hub height is approximately $10.4$ m/s.

The wind farm layout for the two blind tests are shown in Figure 28, where the turbine spacing is marked in red and the intentionally yawed turbines are marked in green boxes.

An additional simulation was performed for the FarmConners blind test by intentionally yawing turbines WT29 and WT32

under the exact same inflow conditions. All simulations have been run for a total of 4500 seconds, where the initial 900 seconds of transient have been removed to have 3600 seconds of operation for the blind test.

Figure 29 shows a comparison of the power production during normal operation (in black) and steering (in red) for the two turbines in the single wake scenario corresponding to wind direction of $90°$. Clearly, the power production is reduced on the upstream turbine, when steering, while the power production increases for the downstream turbine operating in the steered

wake. It is also evident how the normal operating turbines frequently reach rated power. Hence, this scenario is particularly challenging for a blind test as it is close to rated wind speed of the turbine of $11.4$ m/s, where correct representation of the turbine control is essential.

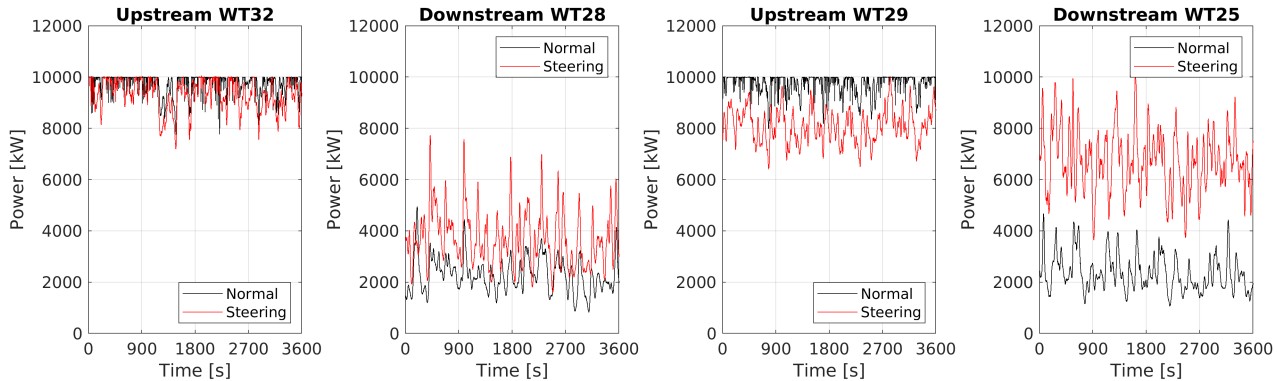

**Figure 29.** Power time-series for single wake scenario in the TotalControl LES blind test for the four turbines: WT32->WT28 and WT29->WT28. Black corresponds to normal operation, while red is power production during wake steering.

## 5.1 Participating Models

Within the FarmConners benchmark, in total 5 models (P8, P10, P11, P16, P20) have participated in TotalControl LES blind

test. The participating models cover a range of fidelities for both the flow modelling and the turbine representation. The participating models and their calibration processes are briefly described in this section, where Table 10 summarises the main characteristics of the models.



| | **P8** | **P10** | **P11** | **P16** | **P20** |
|---|---|---|---|---|---|
| **Wind Farm Model** | HAWC2Farm | StrathFarm | EllipSys3D-RANS-AD[h] | – | SP-Wind[g] |
| **Wake Model** | DWM[a] | Gaussian Wake model[c] with meandering[d] | RANS | Gaussian[c] | LES |
| **Added Turbulence Model** | Windowed TI | None | – | Frandsen[f] | – |
| **Wake Superposition** | Weighted max | Linear velocity deficit[e] | – | Quadratic | – |
| **Wake Deflection** | DWM[b] | Empirical model | – | Gaussian[c] | – |
| **Wake Advection** | DWM[a] | Mean wind speed | – | No | – |
| **Time-Series** | Yes | Yes | No | No | Yes |

**Table 10.** Overview of the participating WFFC-oriented models, TotalControl LES Blind Test

[a](Larsen et al., 2008; Madsen et al., 2010), [b](Larsen et al., 2020), [c](Bastankhah and Porté-Agel, 2016), [d](Poushpas, 2016), [e](Niayifar and Porté-Agel, 2016), [f](Frandsen, 2007) – IEC 2019 standard, [g](Calaf et al., 2010; Allaerts and Meyers, 2015; Munters and Meyers, 2018), [h](Sørensen, 1995)

### 5.1.1 P8

The P8 methodology uses a preliminary version of HAWC2Farm to perform dynamic wind farm simulations on the blind test case. HAWC2Farm leverages the aeroelastic wind turbine software, HAWC2 (Larsen and Hansen, 2007; Madsen et al., 2020), with the dynamic wake meandering (DWM) model (Larsen et al., 2008; Madsen et al., 2010). While HAWC2Farm typically uses HAWC2 as the underlying wind turbine software, this preliminary implementation uses HAWCStab2 data (Verelst et al. (2018)) to synthesise a dynamic turbine model, from which, the dynamic rotor induction is fed into the DWM model to generate the wakes. A flexible version of the DTU10MW is used to calibrate the rotor induction model to ensure the wake deficit is not overestimated, as described by Liew et al. (2020a). The wake passive tracers advect in all spatial directions based on large scale turbulence. Wake deflection is modelled using a modified version of the DWM model, using a Hill's vortex method to simulate the lateral wake deflection. This method has shown good agreement with other wake deflection models as well as full scale experiments (Larsen et al. (2020)). The added wake turbulence model as defined by Larsen et al. (2008) was not activated in this investigation due to the absence of turbine loads, however, the wake dissipation model was adjusted continuously based on a windowed turbulence intensity sensor with a length of 500 seconds. Wake summation is performed by using only the dominant wake (*i.e.* largest wake deficit) at any point in space, followed by a further adjustment which was calibrated using the blind test data;

$$\tilde{U}(x,y,z) = U(x,y,z) - \alpha(N)\min\left(\Delta U_1, \Delta U_2, ..., \Delta U_N\right) \tag{9}$$

where $\tilde{U}(x,y,z)$ and $U(x,y,z)$ is the adjusted and unadjusted longitudinal wind speeds at location $(x,y,z)$ respectively, $\alpha(N)$ is a calibrated factor as a function of the number of upstream turbines, $N$, and $\Delta U_i$ is the wind speed wake deficit of the $ith$ turbine.



### 5.1.2 P10

The P10 model consists of wind turbine models based on the work of Neilson (2010), augmented with an individual blade model based on the work by Gala-Santos (2018). The wind turbine model is validated against DNV Bladed in Gala-Santos (2018). As an input, the wind turbine requires rotor averaged effective wind speeds, which can be generated from input criteria of mean wind speed and turbulence intensity. The requirement for an effective wind speed rather than a point-wise wind speed means that the wind data is not identical as a time-series to that used by the TotalControl LES results. However, the statistical properties of the flow are matched. A lateral wind speed is also generated that informs the meandering of wakes (see Poushpas (2016)). Whilst previous work involving the P10 model has used the Frandsen wake model Frandsen (2007), the wake modelling was recently updated using a the Gaussian wake model of Bastankhah and Porté-Agel (2014, 2016), adapted for effective wind speed modelling (including load impacts) via a similar methodology to the wind shear and tower shadow modelling in Gala-Santos (2018). Further additions are made to account for wake steering effects, though these use an empirical method. It is assumed that the reduction in power from a wake is $\cos^{1.9}(\phi)$ based on Simley et al. (2020) (where $\phi$ is the yaw misalignment. In order to model this change in power via wind speed, and adjustment to the wind speed of $\cos|\phi^k|$ with $k = 1.25$ is applied. Though crude, this method has good agreement with the $cos^{1.9}$ power adjustment up to yaw angles of between 20 and 30 degrees. Beyond ensuring that the statistical properties of the wind field are matched (the same turbulence and mean wind speed), no specific calibration to match to the TotalControl LES data was performed. Note that the new wake model is intended to form part of a future publication.

It should also be noted that the controller used for the wind turbine differs from the DTU controller typically used for this turbine. Compilation issues prevented the DTU controller from being used and so the turbine control strategy and basic controller detailed in Recalde-Camacho et al. (2020) is used instead.

### 5.1.3 P11

The numerical model, code and turbine data are the same as described in Section 4.1.1. Free-stream velocity and total turbulence intensity at hub height used: $U_{H,\infty} = 10.5$ m/s and $I_{H,\infty} = 5.0\%$. These values are based on time and plane averages of the LES data in the region upstream of the wind farm. To save computational time and simplify the simulation, the whole TotalControl reference wind farm was not simulated (32 turbines as seen in Figure 28), but only the relevant row of turbines (WT2, WT3 and WT8, respectively, depending on the case), hence lateral blockage effects are neglected in our simulations.

### 5.1.4 P16

For the P16 results, the calibration procedure is identical to the procedure described in 4.1.3 for the CL-Windcon results. More details on the wake-models can be found in section 2.2.4. The mean value of the time-series wind speed was used as input to our model and the TI was derived from mean and standard deviation.



### 5.1.5 P20

The P20 model is SP-Wind, an in-house Large Eddy Simulation code built on a high-order flow solver developed over the last 15 years at KU Leuven (Calaf et al., 2010; Allaerts and Meyers, 2015; Munters and Meyers, 2018). SP-Wind solves the three-dimensional, unsteady, and spatially filtered Navier-Stokes momentum and temperature equations, with wind turbines contributing to the forcing terms in the equations. Spatial discretization is performed in the horizontal and span-wise directions by using pseudo-spectral schemes while a vertical fourth-order energy-conservative finite differences are used in the vertical direction. The equations are marched in time using a fully explicit fourth-order Runge-Kutta scheme, and grid partitioning is achieved through a scalable pencil decomposition approach. The turbines in the flow domain are paremeterized using the Aeroelastic Actuator Sector Method (AASM) (Vitsas and Meyers, 2016). Subgrid-scale stresses are modeled with a standard Smagorinsky model with Mason and Thomson wall damping (Allaerts and Meyers, 2015).

Wind farm simulations are run for a period of 75 minutes, which includes a 15 minute start-up period for the settling of initial transients. A previously developed inflow database (Munters et al., 2019a, b, c, d) is utilised to provide the inflow conditions for the wind farm through the concurrent precursor method (Stevens et al., 2014; Munters et al., 2016), and the entire wind farm is rotated in the flow domain to simulate different wind directions. The structural and aerodynamic properties of the DTU 10 MW turbine tower and blades, and the DTU Wind energy controller are used to simulate the turbine operation (Bak et al., 2013).

### 5.2 Single Wake Results

For the single wake case within TotalControl LES blind tests, the model performances are compared for 2-turbine subsets of the TotalControl reference wind farm in Figure 28; with WT32-WT28 and WT29-WT25 for $90°$ incoming wind direction. In these 2-turbine subsets, the upstream turbines WT32 and WT29 are misaligned for $20°$ and $30°$ counter-clockwise, respectively. The blind tests include 1-hour simulations where the wake loss reduction, $\Delta u$ in equation 10, and power gain, $P_{\text{GAIN}}$ in equation 11, are compared among the participating models. Note, that both $\Delta u$ and $P_{\text{GAIN}}$ are evaluated per model via the submitted controlled (WFFC) and Normal Operation results of the participating models.

$$\Delta u = \frac{\sum(U_{\text{up}} - U_{\text{down}})_{\substack{\text{Normal} \\ \text{Operation}}} - \sum(U_{\text{up}} - U_{\text{down}})_{\text{WFFC}}}{U_{\text{up}}} \tag{10}$$

where $U$ is either the hub height or rotor effective wind speed that represents the spatially averaged wind speed over the rotor; at the upstream, $U_{\text{up}}$, and downstream turbine(s), $U_{\text{down}}$.

$$P_{\text{ratio}} = \frac{(\sum_{i=1}^{n} P_i)_{\text{WFFC}}}{(\sum_{i=1}^{n} P_i)_{\substack{\text{Normal} \\ \text{Operation}}}} \quad \text{and} \quad P_{\text{GAIN}} = P_{\text{ratio}} - 1 \tag{11}$$

where $P$ is the power and $i$ is the turbine ID in the investigated subset of the layout.




Figure 30 shows the wake loss reduction for both WT32-WT28 and WT29-WT25 turbine pairs with $-20°$ and $-30°$ up-
stream misalignment (counter-clockwise), respectively. The time-series on the left is illustrated at the frequency of the wind
speed signals for all the (quasi-) dynamic models, including the validation data set TotalControl LES. Notably high fluctuations
are seen for both of the control settings in the validation data set, which are very well captured by P8 with 5-sec resolution. P20
is also in relatively good agreement, where the observed dynamics in the validation data set is closely followed. For P10, the
rotor effective wind speed is used throughout the model, resulting in much lower fluctuations compared to other dynamic mod-
els due to spatial averaging of the point-wise wind speed variation. Figure 30 boxplots indicate that all the models, including
steady-state P11 and P16, suggest a better recovery when the upstream turbine is $-30°$ yawed compared to $-20°$ with up to
40% reduction in median wake losses relative to normal operation. The uncertainty, however, should also be noted with large
fluctuations in the dynamic simulations with WFFC in 2-turbine configuration with 5D spacing.

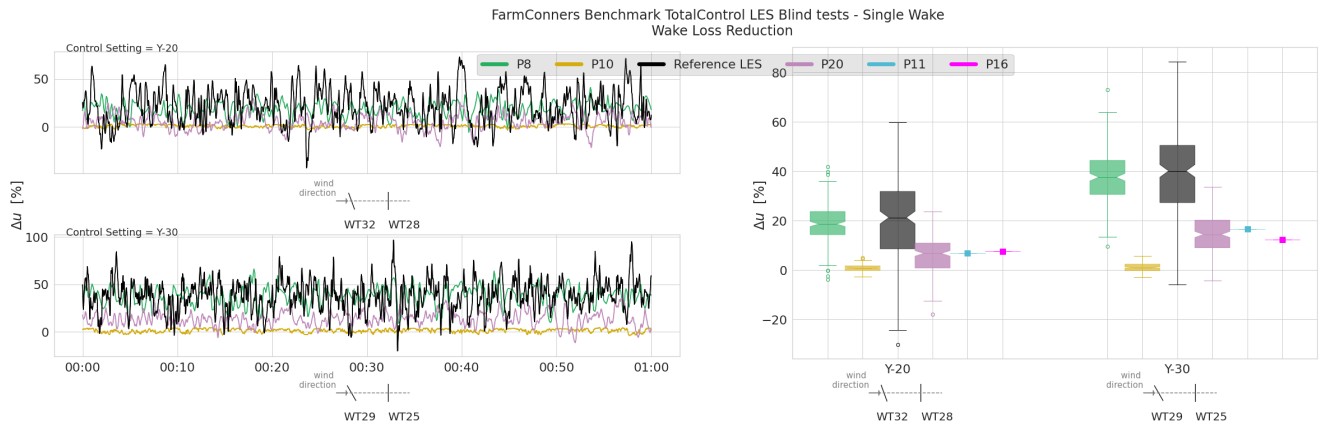

**Figure 30.** TotalControl LES blind tests, wake loss reduction, $\Delta u$, for single wake under wake steering for 2-turbine configurations with 5D
spacing (WT32-WT28 with $20°$ yaw and WT29-WT25 with $30°$ yaw at the upstream turbines). **Left:** Time-series of $\Delta u$, illustrated at the
submitted frequency by the participants, for both of control scenarios. **Right:** The distribution of the $\Delta u$ within the 1-hour blind test period
for both $20°$ and $30°$ counter-clockwise yaw misalignment WFFC scenarios.

The transition from the wake loss reduction to the power gain comparison is highly affected by the turbine representation and
the controller implementation of the participating models. Figure 31 highlights that clearly, especially considering the higher
sensitivity of the power surface to the investigated wind speed interval around the rated region (as illustrated in Figure 29).
Although the wake recovery is captured very well by P8 as seen in Figure 30, the positive reduction in wake losses do not
translate to positive power gains for either of the control settings. Similar to P10, the power loss at the upstream turbine due to
misalignment exceeds the power gain observed at the downstream turbine. The trend is further emphasised for higher degrees
steering and potentially reinforced by under-representation of the power curve under normal operation downstream (shifted to
the right). Here, it should be underlined again that the controlled settings were not included in the calibration data set distributed
to the participants and the mismatch in the upstream power loss can partially be attributed to that. See Appendix C for further
analysis and illustration of the power difference per upstream and downstream turbines.



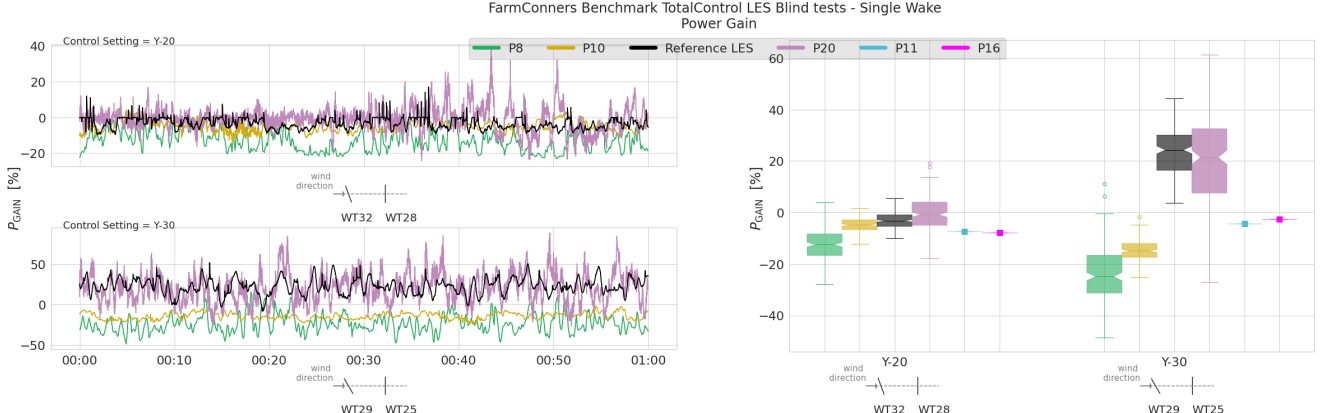

**Figure 31.** TotalControl LES blind tests, Power Gain, $P_{\text{GAIN}}$, estimations for single wake under wake steering for 2-turbine configurations with 5D spacing (WT32-WT28 with $20°$ yaw and WT29-WT25 with $30°$ yaw at the upstream turbines). **Left:** Time-series of $P_{\text{GAIN}}$, illustrated at the submitted frequency by the participants, for both of control scenarios. **Right:** The distribution of the $P_{\text{GAIN}}$ within the 1-hour blind test period for both $20°$ and $30°$ counter-clockwise yaw misalignment WFFC scenarios.

Figure 31 also shows that the only participating LES, P20, predicts similar levels of power gain overall with TotalControl LES, for both of the control scenarios. Although the reduction of the wake losses are underestimated in comparison, it is compensated by lower power losses estimated at the controlled turbine upstream (see Figure C1 in Appendix C for further details). The time-series in Figure 31 also indicate a faster controller response in P20 which results in a much larger spread around the reported $P_{\text{GAIN}}$.

The steady-state results from P11 and P16 is in relatively good agreement for the lower degrees of misalignment at $-20°$, but underestimated the power gain likelihood for higher steering. It is indeed in line with their behaviour for similar control scenarios under CL-Windcon LES blind tests discussed in Section 4.2, where less wake deflection is produced by the models at higher steering as also observed in the wake loss reductions results in Figure 30 above.

### 5.3    Multiple Wake Results

Similar to the single wake cases, $20°$ and $30°$ counter-clockwise upstream yaw misalignment control scenarios for $90°$ incoming wind direction are investigated for the multiple wake results in TotalControl LES blind tests. This time, 8-turbine subsets with 5D spacing within the TotalControl reference wind farm are analysed, namely the rows WT32, WT25, ..., WT1 and WT29, WT28, ..., WT4 in Figure 28. The layouts of these subsets as well as the corresponding control settings are illustrated in the presented results, *i.e.* the x-axes in Figure 32.





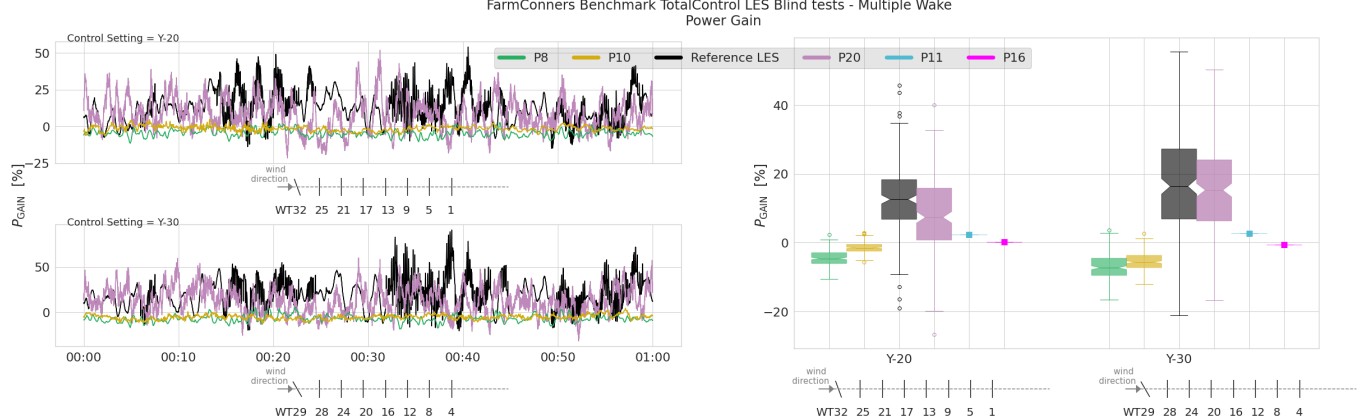

**Figure 32.** TotalControl LES blind tests, Power Gain estimations, $P_{\mathrm{GAIN}}$, for Multiple Wakes under the upstream Wake Steering for 8-turbine configurations with 5D spacing (WT32 – WT4 with 20° yaw and WT29 – WT1 with 30° yaw at the upstream turbines). **Left:** Time-series of $P_{\mathrm{GAIN}}$, illustrated at the submitted frequency by the participants. **Right:** The distribution of the $P_{\mathrm{GAIN}}$ within the 1-hour blind test period, for both 20° and 30° counter-clockwise upstream yaw misalignment WFFC scenarios.

Figure 32 shows the power gain, $P_{\mathrm{GAIN}}$ of the 8-turbine, 5D spacing wind farms under 20° and 30° counter-clockwise upstream yaw misalignment WFFC scenarios. Compared to the single wake results in Figure 31, the fluctuation is seen to be much larger in the validation data set for multiple wakes reaching up to 30%. However, the trend in the median is similar with positive power gains for both of the WFFC scenarios, indicating higher likelihood under −30° upstream misalignment. It is closely captured by the other LES, P20, with highly correlated time-series. Conversely, the other (quasi-)dynamic models

underestimates the power gain with a larger discrepancy for the higher steering. At least for P8, the overestimated power loss at the controlled turbine is argued to be the underlying reason, as also observed in the comparison of the single wake results in Section 5.2 and Appendix C. Similarly for the steady-state models P11 and P16, the under-estimation of the power gain perseveres. However, the difference is less prominent as the effect of under-represented upstream wake deflection fades out with increasing number of turbines along the row.

## 5.4   Summary of the TotalControl LES Blind Test

With additional participants, longer time-series, simpler layouts and focused control scenarios, TotalControl LES blind test provides interesting comparison and supplementary discussion for model performance. Its highlights can be summarised as below.

**Similar trends in the high fidelity models** TotalControl LES blind test hosts a unique comparison of two separate LES
frameworks developed in different institutes. As stated earlier, comparison of lower cost models against such tools (often
        referred as numerical validation) is seen as a pre-requisite for their implementation in the field. Therefore, it is reassuring
        for the further adoption of the WFFC technology to have relatively good agreement with correlated dynamics between



the two methodologies; as was observed in the TotalControl blind test especially for the power gain results with different control scenarios in single and multiple wake analysis.

**Turbine representation and controller implementation** The TotalControl LES blind test showcases the sensitivity of the results to the turbine representation and the implementation of the controller. This was particularly emphasised via the translation of potential wake loss reduction to the power gains under WFFC. A prior study to compare (and calibrate if relevant) the power surfaces and controller operation under uniform flow is recommended for similar blind tests or numerical validation studies in the future.

**Uncertainties and risk** Although longer time-series compared to CL-Windcon LES blind tests, TotalControl LES also replicates conventionally neutral boundary layer. It does not include severe variability that might be observed in the field (*e.g.* Göçmen et al., 2020b), and corresponding uncertainties particularly for the wind direction. For the investigated scenarios, higher likelihood of benefits via wake steering control is estimated by the majority of the models in the blind tests. However, especially the high fidelity models indicate significant risk of inducing additional losses, given the probability

of the power gain based on instantaneous values. Such trends are also discussed in (Kheirabadi and Nagamune, 2019) and their implications to operational risks under WFFC should be further evaluated by the end-users of the technology.

## 6   CONCLUSION

Here in this article we present the results of the FarmConners benchmark for code comparison under controlled operation. The benchmark brought together 4 data sets generated under several European WFFC projects: 1) SMV wind farm field data,

2) CL-Windcon wind tunnel experiments, 3) CL-Windcon LES and 4) TotalControl LES databases. Although the original benchmark included more control strategies (*i.e.* axial induction) and quantities of interest (*i.e.* load channels), the analysis presented here is limited to wind speed and power behaviour under wake steering WFFC.

The results from 13 participants and 16 participating models in total are then presented separately under these 4 blind tests. The highlights of the blind test exercises are summarised individually in their corresponding sections through the article. A

compilation of the observations/reference simulations and participating model trends for the overall benchmark is listed below.

**Customisable WFFC-oriented models** The overwhelming majority of the participating models in FarmConners benchmark are parametric, typically modular frameworks; indicating their popularity within the field. Using similar approaches to resolve the wake behaviour, the main difference among them is the calibration procedure. Accordingly, the importance of variety in terms of control set-points in the calibration dataset is underlined in all the blind tests. Similarly, a clear

description of the calibration procedure with a list of parameters when disseminating the results is crucial for reproducible and credible assessment of the potential gains via WFFC.

**Beyond flow modelling** FarmConners benchmark also highlights the importance of turbine representation and controller implementation in realisable power gains via wake steering WFFC. A separate comparison and calibration of the power





surfaces and controller operation for isolated cases are recommended prior to field implementation, as well as future
blind tests or validation studies.

**Overall a good agreement** Especially for well-calibrated models with a relatively good representation of the dynamics, the
participating model agreement to the observations/reference simulations are seen to be reasonable for all the blind tests.
This is particularly the case for smaller yaw control set-points and lower (temporal and/or spatial) fluctuations in the
inflow. Within the benchmark, two separate LES frameworks developed in different institutes are also compared and
high correlation observed in their results are found reassuring for the TRL of WFFC where high-fidelity simulations are
considered to be the key enabler for further field implementation.

Although an already extensive analysis, the presented FarmConners benchmark results are limited in the applied control
strategy and the investigated quantities of interest. It should therefore be read as the first step where other benchmarks can be
built on. With increasing availability of field tests, wind tunnel experiments and reference high-fidelity simulation databases,
future work should include larger wind farms, different control strategies and other control objectives such as potential load
alleviation and profit maximisation.

*Code availability.*   The notebooks for the blind tests results, including data snippets, can be achieved via the public repository of FarmConners
benchmark (Göçmen et al., 2021).

*Data availability.*   All the data used in FarmConners benchmark blind tests can potentially be made available for non-commercial purposes.
Please contact us (per blind test data set) using the details provided under the FarmConners benchmark wiki page https://farmconners.
readthedocs.io/en/latest/contact_us.html.

## Appendix A: Wind Farm Field Data Blind Test - Multiple Wake Results

As discussed in Section 2.5, the multiple wake cases for the SMV wind farm blind test results are deemed to be inconclusive.
The main difficulty is the wind farm layout orientation and several partial wake scenarios included in the wider sector behind
the controlled turbine. Nevertheless, the analysis is presented here for the interested parties, where the main outcomes in terms
of the participating model performances are in line with the single wake analysis of the same blind test presented in Section 2.4.

Due to its wider wind direction sector, the filtered data set for wake steering consists of 579 10-min data points (including
the 216 points already used for the single wake case), while the normal operation data set used to calculate the baseline wake
effect is made of 1849 10-min data points (841 recorded in June and July 2017, 1008 recorded in October and November 2017).





## A1 Time-series comparison

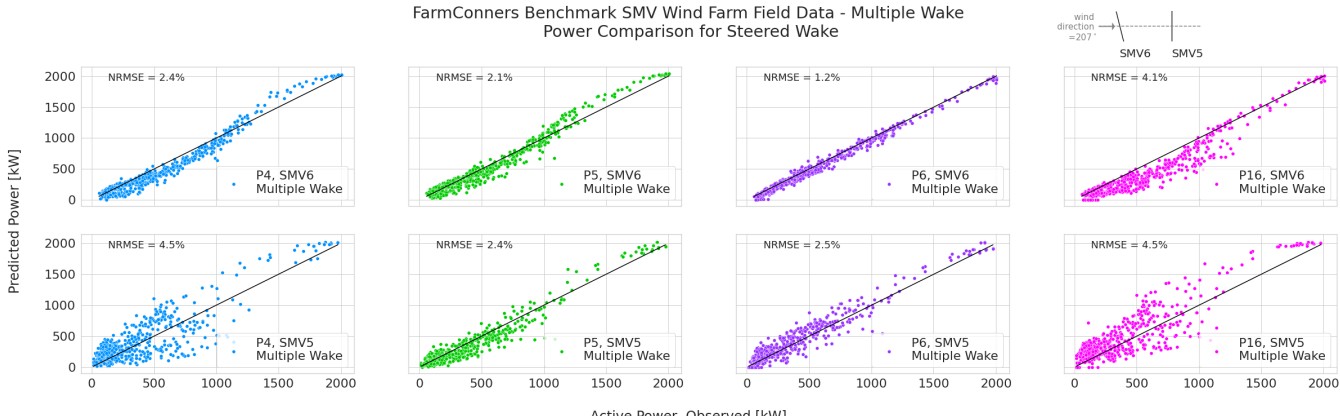

**Figure A1.** SMV WF Field Data, Multiple Wake under Wake Steering WFFC with -13.3° upstream misalignment - Power comparison. Representative layout with corresponding yaw control setting is illustrated at the upper right corner.

## A2 Binned Quantities of Interest: Energy Ratio & Power Gain

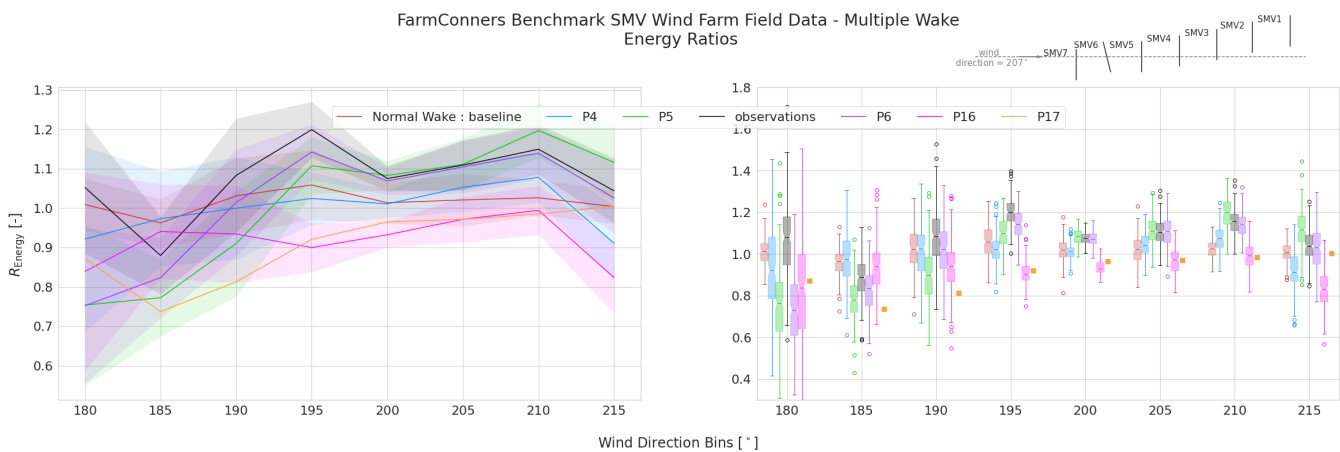

**Figure A2.** SMV WF Field Data, Multiple Wake under Wake Steering WFFC with -13.3° upstream misalignment - Energy Ratio comparison. Representative layout with corresponding yaw control setting is illustrated at the upper right corner.

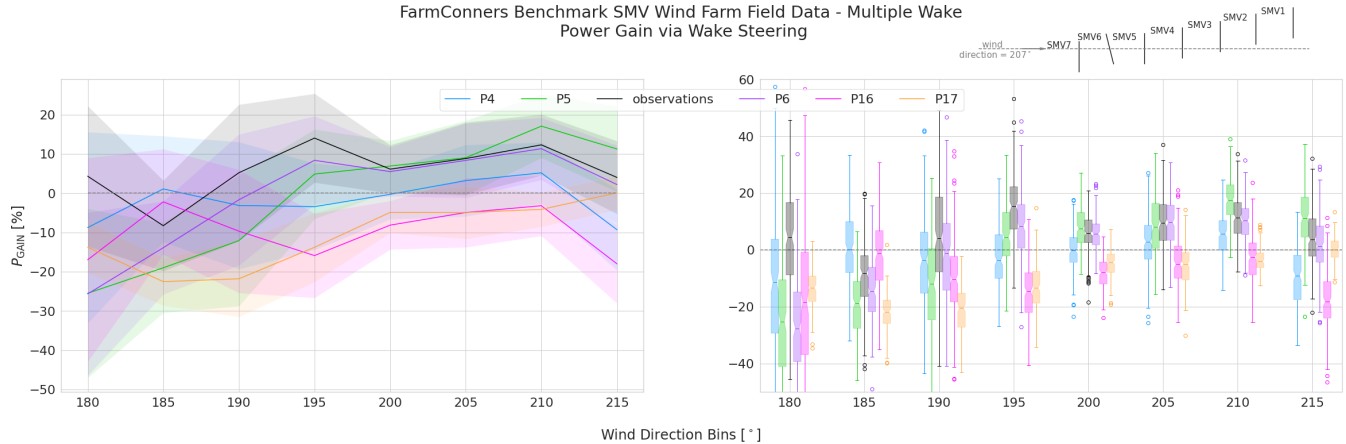

**Figure A3.** SMV WF Field Data, Multiple Wake under Wake Steering WFFC with -13.3° upstream misalignment - Power Gain comparison. Representative layout with corresponding yaw control setting is illustrated at the upper right corner.

## Appendix B:  CL-Windcon Wind Tunnel: Further Results and Wake Measurements

The Appendix includes further graphs providing quantities of possible interest to the readers.

### B1   Single Full-Wake Results

Figure B1 reports the normalized measurement of the longitudinal flow speed at hub height $\bar{U}_{\text{wake}}$ within the wake shed by a single G1, its best-fitted Gaussian curve (c.f. Eq. 4, as well as the corresponding models' predictions. The wind speed measured by the pitot tube placed 3D upstream of the turbine was used for the normalisation.





**Figure B1.** Single full-wake results: comparison between measured and predicted wake profiles at hub height.

Figure B2, instead, compares the longitudinal turbulence intensity, measured at hub height and within the wake shed by a single G1, to its corresponding predicted values,



**Figure B2.** Single full-wake results: comparison between measured and predicted turbulence intensities profiles at hub height.

## B2 Multiple Wake Results

Finally, Figures B3 and B4, compare the longitudinal turbulence intensity, measured at three different distances from the gournd and within the wake shed by two G1s, to its corresponding predicted values,





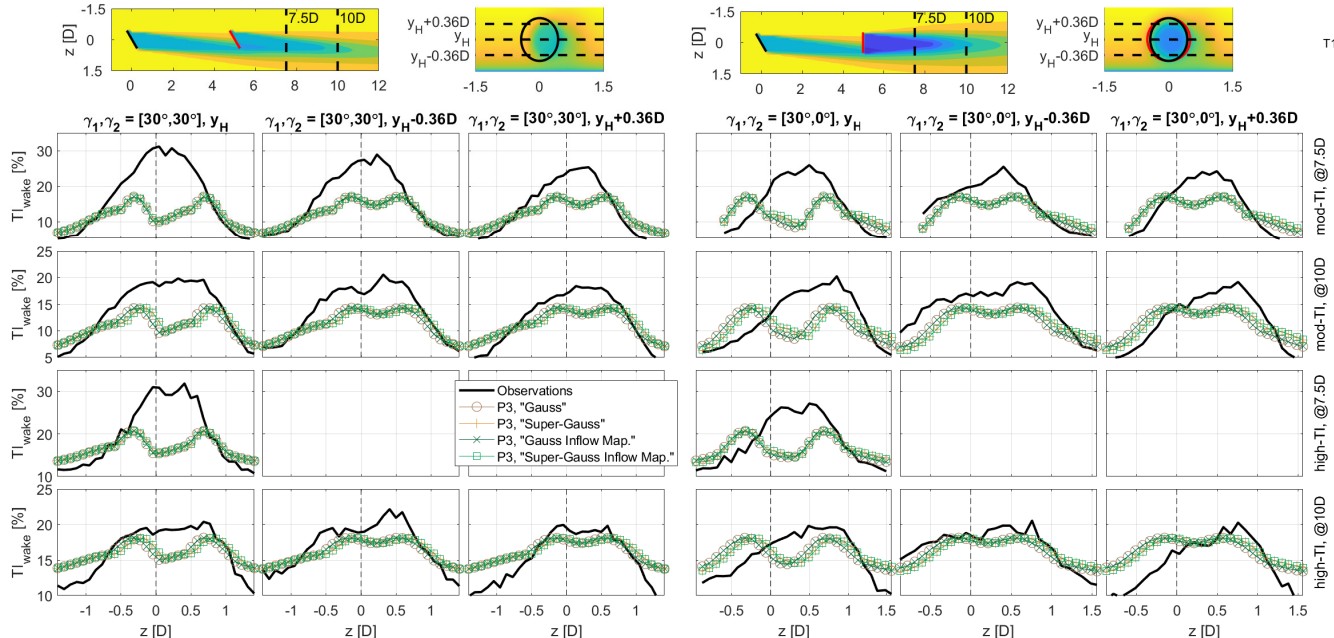

**Figure B3.** Multiple Wake results: comparison between measured and predicted longitudinal turbulence profiles within the wake shed by two aligned G1s.

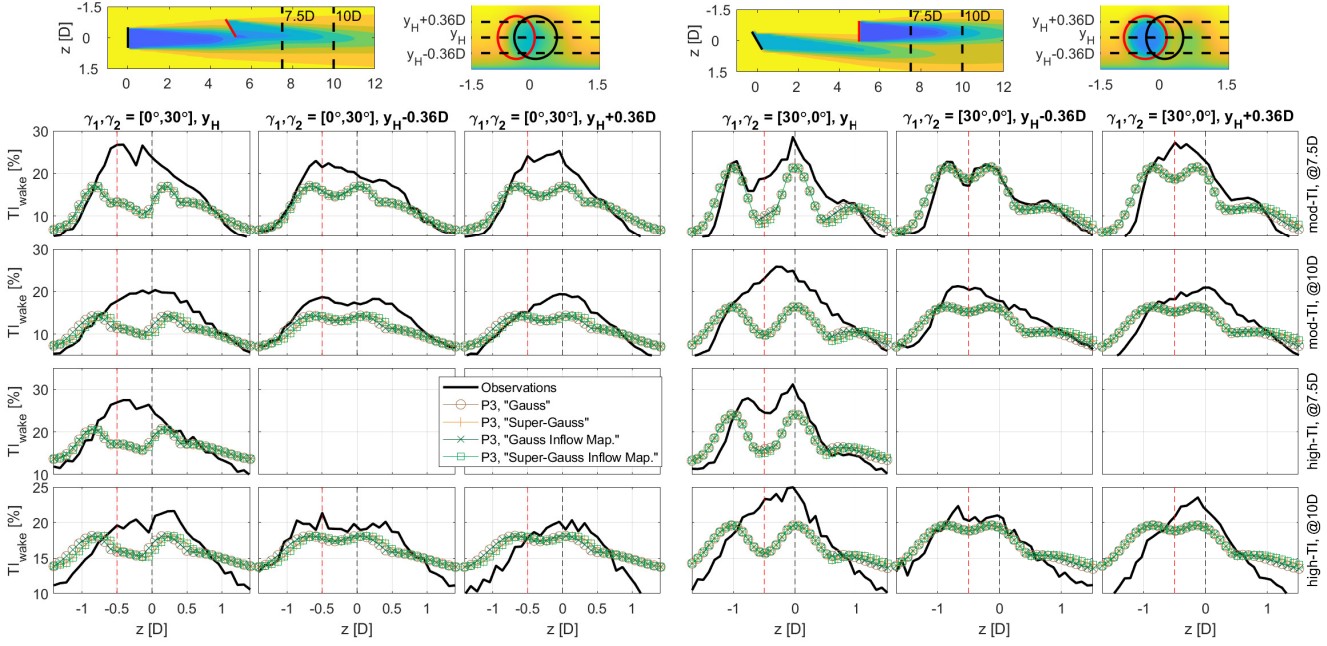

**Figure B4.** Multiple Wake results: comparison between measured and predicted longitudinal turbulence profiles within the wake shed by two 0.5D laterally-shifted G1s.





### Appendix C: TotalControl LES Blind Test - Power difference per turbine for single wake cases

In order to analyse the participating model behaviours further, power difference, $\Delta P$ in equation 7, per turbine is compared for
upstream (WT32 and WT29) and downstream (WT28 and WT25) turbines is illustrated in Figures C1 and C2 below, respectively. They show the differences in the representation of the controlled and normal operation turbine power surface, where the former was not included in the calibration data set. This analyses supports the discussions carried out under Section 5.2 and aims to distinguish the underlying reasons of the model behaviours in terms of power gain particularly in Figure 31. It further highlights the differences in the controller implementation and turbine representation, given the sensitivity of the results to the
blind test. It should also be noted that P10 submitted the time-series for $P_{\text{ratio}}$ directly, therefore excluded in the illustrations below.

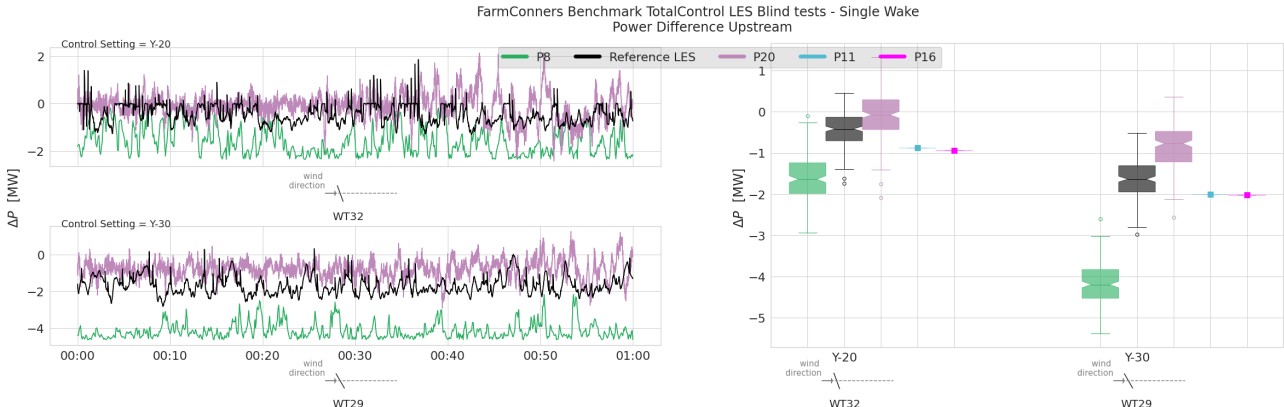

**Figure C1.** TotalControl LES, Single Wake under Wake Steering power difference, $\Delta P$ comparison for the upstream turbines WT32 ($-20°$ yaw) and WT29 ($-30°$ yaw). **Left:** Time-series of $\Delta P$ at the upstream turbines illustrated at the submitted frequency by the participants. **Right:** The distribution of $\Delta P$ within the 1-hour blind test period. For both $20°$ and $30°$ counter-clockwise yaw misalignment WFFC scenarios.

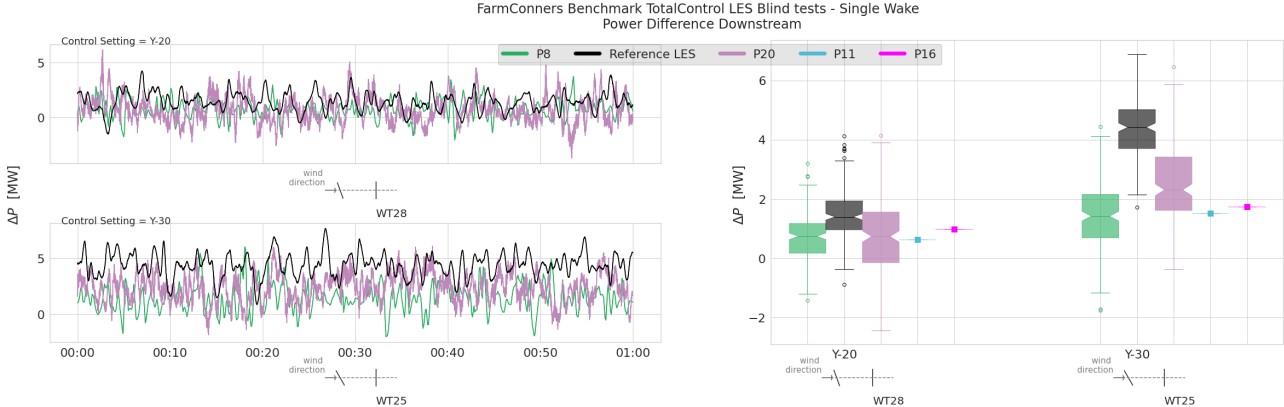

**Figure C2.** TotalControl LES, Single Wake under Wake Steering power difference, $\Delta P$ comparison for the downstream turbines WT28 (behind $-20°$ yawed turbine) and WT25 (behind $-30°$ yawed turbine). **Left:** Time-series of $\Delta P$ at the downstream turbines illustrated at the submitted frequency by the participants. **Right:** The distribution of $\Delta P$ within the 1-hour blind test period. For both $20°$ and $30°$ counter-clockwise yaw misalignment WFFC scenarios.

*Author contributions.* FarmConners benchmark has been a comprehensive collaborative effort within the WFFC community world-wide. Tuhfe Göçmen is the main coordinator of the FarmConners benchmark and the lead author of this article. She has led Sections 2, 4 and 5. Filippo Campagnolo has contributed significantly to the FarmConners benchmark organisation and led CL-Windcon wind tunnel blind tests in Section 3. Thomas Duc, Irene Eguinoa and Søren Juhl Andersen have also contributed significantly to the FarmConners benchmark organisation and aided with the data preparation, description and analysis of the results for SMV wind farm field blind test in Section 2, CL-Windcon LES blind test in Section 4 and TotalControl LES blind test in Section 5, respectively. Vlaho Petrović has also been an active member of the FarmConners benchmark organisation team and, together with all the other members listed above, has co-written the introductions and helped disseminate the benchmark further to achieve high(er) number of participants. The rest of the authors, namely Lejla Imširović, Robert Braunbehrens, Ju Feng, Jaime Liew, Mads Baungaard, Maarten Paul van der Laan, Guowei Qian, Maria Aparicio-Sanchez, Rubén González-Lope, Vinit V. Dighe, Marcus Becker, Maarten van den Broek, Jan-Willem van Wingerdan, Adam Stock, Matthew Cole, Renzo Ruisi, Ervin Bossanyi, Niklas Requate, Simon Strnad, Jonas Schmidt, Lukas Vollmer, Frédéric Blondel, Ishaan Sood and Johan Meyers, are the benchmark participants who have provided model runs for the blind test(s) they have registered for, descriptions in the corresponding sections for their participating models and extensive reviews of the analysis. They are listed in no particular order to preserve anonymity as much as possible.

*Competing interests.* At least one of the (co-)authors is a member of the editorial board of Wind Energy Science.





*Acknowledgements.* The FarmConners benchmark is organised and conducted under the FarmConners project, funded by the European Union's Horizon 2020 research and innovation programme with grant agreement No 857844.





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
