# Peer review of "FarmConners Wind Farm Flow Control Benchmark: Blind Test Results Part I"

_Wind Energy Science, 2022_

## Referee Comment (RC2)

**Reviewer comments**

This is a very interesting paper presenting blind tests results for different wake models estimating the power gains through wind farm wake control (WFFC). The paper focuses on wake control by yawing the upstream rotors. Four different data sources are used for the blind comparisons: open field measurements form an onshore wind farm, wind tunnel measurements using model rotors, and two different high-fidelity simulation tools based on Large Eddy Simulation (LES).

The paper is in general very well written. Use of the English language is very good and only a few typos are present. The presented analysis is very comprehensive and the contribution to new knowledge is significant. However, it is believed that the paper is far too long to be published as a single journal paper. It is recommended to split the script in two separate papers, for example one dealing specifically on the modelling of large-scale rotors (Blind tests 1, 3 and 4), while the other focusing only on the blind test using the wind tunnel data.

The following is a summary of the main comments:

**Major Comments**

1. Page 2, Introduction: It is recommended to highlight in more detail literature sources documenting previous validation work of the wake models used in this blind comparison.

2. Page 9: Equation 1 – the paper should elaborate in more detail the limitations of this model for the range of time-averaged CT considered in the blind tests. References to relevant past works dealing with yawed rotor aerodynamics should be included. Consideration has to be given to the following: (1) n does depend on the tip speed ratio (& hence CT) acting on the rotor; (2) rotor yaw causes CP (and CT) to become a function of time and (3) the time-averaged CP does decreases with yaw angle.

3. Page 10, first paragraph: the statistics of the polynomial fits should be included (R squared, standard error).

4. Page 10: an important consideration where considering open field measurements is the influence of shear. Wind shear causes the wake to deflect upwards, thus also theoretically contributing to the wake losses reduction. Thus, apart from considering the impact of rotor yawing, the effects of shear should also be assessed. Have the effects of wind shear without rotor yawing been examined? This analysis is important inorder to properly quantify the real contribution of rotor yawing alone.

5. Page 13: Including data about the site topography is important for the reader to understand better the operating environment of the wind farm. Such data can be obtained through satellite data.

6. Page 13, line 5: the term "waked directions" is inappropriate.

7. Page 14, line 285: It is understood that, following the binning process, the average wind direction was computed - was a vector average used?

8. Page 15, first para: It is helpful to present the relation for estimating the wake skew angle here. How does the formula correct for the variation of the wake skew angle with CT? Past works have shown that this follows a quasi-linear relationship.

9. Page 15, line 326: Explain what these parameters stand for.

10. Page 17, line 373: To what extent has this transfer function been validated?

11. Page 17, line 380: The numerical technique applied for filtering has to be stated.

12. Page 22, Section 3.1: Details about the operating conditions of the rotor are lacking (e.g. tip speed ratio)

13. Page 22, Section 3.1: the Reynolds number matching is difficult to achieve in the wind tunnel tests given that the scale ratio is far too small. The paper should elaborate about this limitation.

14. Page 26, line 578: in yawed rotors, the disk-averaged induction at the rotor becomes a function of time. This violates the assumption for the simplistic wake losses models that are based on the linear momentum equation applied for stead flow conditions. A remark on this matter should be included.

15. Page 37: The paper should explain briefly the numerical verification work undertaken to ensure that numerical errors are negligible. It should be ensured that the uncertainty arising from numerical spatial and temporal discretization does not mask the differences in power estimated with and without WFFC.

**Minor Comments**

1. Page 1, line 11: "The majority of …"

2. Page 2, line 1: "In addition to the flow modelling, the sensitivity….."

3. Page 3, line 1: Do not start new paragraph with "Also"

4. Page 3, line 60: "….power gain at the wind farm level by applying wake steering control strategy."

5. Page 3, line 65: "induction control that were included in the majority"

6. Page 4, Line 1: "The results of the benchmark are classified based on the technology readiness level…"

7. Figure 10: The quality of this figure has to be improved.

8. Figure 11: enlarge font size of text in the figure (this applies to similar figures in the paper)

9. Figures 12 and 13: the figures are too small to make the colour scheme easy to distinguish between the different plots.

---

## Author Comment (AC1)

**Response to Anonymous Referee 1 comments of Manuscript ID WES-2022-5 entitled "FarmConners Wind Farm Flow Control Benchmark: Blind Test Results Part I"**

Thank you for taking the time to review our article. We would, however, like to state that we find the excessive use of exclamation points (!) quite confusing in your comments and hope that you would still find them adequately addressed. Please also note that we have decided to divide the article into two, where Part I (the current form) consists of Blind Tests #1, #3 and #4 (field tests and LES comparisons); and Part II focuses on Blind Test #2 with the wind tunnel experiments.

1. The labels of the following figures is too small!!! Figures 3, 4, 6, 7, 9, 11, 12, 13, 16, 18, 19, 27, 30, 31, 32, A1, A2, A3, B1, B2, B3, B4, C1, C2

    - Figures 3, 4, 6, 7, 9 (regarding 'methodology'): The axis labels are explicitly added to the captions to increase clarity.

    - Figures 11, 12, 13 and 27, 30, 31, 32, as well as A1, A2, A3, C1, C2 (regarding 'results': The font size for the axis labels are increased directly.

    - Figures 16, 18, 19 as well as B1, B2, B3, B4: Are now excluded from this article (Part I of the benchmark results), and transferred to the subsequent paper (Part II of the benchmark results).

2. Table 1: Instead of "x" put "name of the research institute / name of Code" Table 1 is put in the introduction to give a generic overview of how many participants have participated in the benchmark, and in which blind test. Indication of the name of the institute per participants in Table 1 would violate the anonymity, so it is avoided. For the indication of the models, the benchmark results show more details are needed. For example, many participants have used the same framework for the models (e.g., FLORIS) with different sub-modules and calibration procedures. This is highlighted in detail in Tables (now) 3, 8 and 9, where such information is provided per blind test.

3. Table 3: where is the difference between P4 and P5? Maybe write in the "Wake Model" line "Calibration1" and "Calibration2"

    The difference between P4 and P5 are indeed in the calibration procedure (not just the wake model, but several other modules within the modelling framework, including the yaw loss parameters as highlighted later in the section). To further underline such differences, a few sentences are added before Table 3 reads as "However, it should be noted that seemingly identical model applied by the participants is likely to be calibrated differently, resulting in different performance in their predictions. This is further discussed in the detailed model descriptions per participants, and highlighted in the blind test results later in the section."

4. Figure 3: Include SMV5 in the caption and Figure 4: Include SMV7 in the caption

    Turbine numbering SMV1 – SM7 are all in x-axis of Figures 3 and 4. Now explicitly added to the caption.

5. Figure 3, Figure 4: what do the empty black circles represent?

    They represent outliers of the boxplot - now added explicitly to the caption.

6. Figure 5a: diagram "WS", caption "delta_WS": correct one of them.

    The caption is corrected.

7. Table 6: Describe the parameters "alpha", "beta", ...,"n". Best, if also in the text!

    It is out of scope to describe the parameterization of the participating models in detail. Rather, the benchmark focuses on the inference part of the existing (widely-used) models, including the calibration processes. However, before Table 6, the interested readers are now encouraged to "For further details on the description of model parameters, see (NREL, 2021). For further discussion on the significance of such parameterization, see e.g., (van Beek et al., 2021)."

8. Figure 9: Distinguish blue dot from green line (not both measurement)! And orange dot from red line!

    They are indeed both. To clarify, now added to the caption 'Solid lines are the binned average of the measurements and model results indicated as scatters.'

9. Figure 10: Lines too thick, not clearly visible! Make lines (symbols) smaller

    Figure 10 is enlarged now.

10. Line 408: Why are the wind direction bins in the text +-2.5° and in Figure 12 the labels are by 2°?

    Figure 12 x-axis labels (left figure) are updated now.

11. Figure 12 left: Colours not always easy to distinguish, put symbols

    Figure 12 left is updated with Participant IDs as markers now.

12. Figure 12: Why line plot (left) and bar diagram (red)? No added value! Same for Figure 13

    For Figures 12 and 13, the line plot on the left is a 'classical' representation of the energy ratio and power gain as seen in literature. The box-plots on the right is argued to be a better representation of the distribution of the same quantities, including uncertainty levels to be expected.

13. Rewrite Lines 405 - 422 because concept is quite complex and not described enough. Especially rename term "_i_Test" because the naming is confusing! Give table with weights and explain concept much more detailed.

    The description of the weights under Section 2.4.2 is now extended and $P_i^{Test}$ is renamed as $P_i^{WF}$ in equation 2.

14. Line 1000: give equivalent number of revolutions for 3600s simulation time

    It is not clear how that is relevant to the rest of the blind test presented in the article. Accordingly the paragraph (now starts at line 677) is left unchanged. More information on the reference database for the TotalControl LES blind test, as well as the data itself, can be accessed via `https://data.dtu.dk/articles/dataset/FarmConners_cnblz02e3m_rot90_WakeSteering/13414922`.

15. Table 10: line for partner description

    (Now Table 9) As stated earlier, the participant IDs are kept anonymous throughout the study. That was a pre-condition many participants have requested from the initial launch of the FarmConners benchmark.

16. Line 1089: make clear what is meant by subset! WT32,WT28 and WT29, WT25 or different? All turbines in the parc?

    (Now line 767) Clarification added at the end of the sentence as "(2-turbines for single and 8-turbines for multiple wake analysis behind the controlled turbines WT29 and WT32 in Figure 19)".

17. Line 1123,1124: 8-turbine subset have to be horizontal lines! WT32, WT25,..., WT1 are not in a row!

    (Now line 800) The IDs of the first turbines in the rows are corrected.

---

## Author Comment (AC2)

**Response to Anonymous Referee #2 comments of Manuscript ID WES-2022-5 entitled "FarmConners Wind Farm Flow Control Benchmark: Blind Test Results Part I"**

Thank you very much for taking the time to review our article and your comments. We're glad to hear you find the results also interesting. Following your suggestion, and other feedback we got from the community, we have indeed decided to divide the article into two, where Part I (the current form) consists of Blind Tests #1, #3 and #4 (field tests and LES comparisons); and Part II focuses on Blind Test #2 with the wind tunnel experiments.

**Major Comments**

1. Page 2, Introduction: It is recommended to highlight in more detail literature sources documenting previous validation work of the wake models used in this blind comparison.

   The most recent (and the most in-depth) discussion of the WFFC validation overall is presented in Meyers et al. 2022 - now added to the Introduction (at the end of the first paragraph).

2. Page 9: Equation 1 – the paper should elaborate in more detail the limitations of this model for the range of time-averaged CT considered in the blind tests. References to relevant past works dealing with yawed rotor aerodynamics should be included. Consideration has to be given to the following: (1) n does depend on the tip speed ratio (& hence CT) acting on the rotor; (2) rotor yaw causes CP (and CT) to become a function of time and (3) the time-averaged CP does decreases with yaw angle.

   There are indeed several studies in literature indicating the sensitivity of the yaw loss exponent (or equation 1 in general) to the wind turbine model/type and experimental settings. To clarify further, a few sentences are added before equation 1, including the majority of these studies (around line 170) that read as: "There are several values proposed for n depending on the experimental setup and the turbine type; where n = 3 is typically considered based on blade element momentum (BEM) theory as well as numerous wind tunnel experiments (Krogstad and Adaramola, 2012; Bartl et al., 2018b, a), n = 1.8 is proposed by other wind tunnel experiments (Schreiber et al., 2017) and LES (Draper et al.,175 2018), n = 1.88 was considered in (Gebraad et al., 2016), and n = 1.4 was used in (Fleming et al., 2014). Further discussion on varying n within the wind farm (based on upstream and downstream turbine configurations) can be found in (Liew et al., 2020b)."

3. Page 10, first paragraph: the statistics of the polynomial fits should be included (R squared, standard error).

   Now a new figure (Figure 5) is added, in which the statistics of the fit is also described.

4. Page 10: an important consideration where considering open field measurements is the influence of shear. Wind shear causes the wake to deflect upwards, thus also theoretically contributing to the wake losses reduction. Thus, apart from considering the impact of rotor yawing, the effects of shear should also be assessed. Have the effects of wind shear without rotor yawing been examined? This analysis is important in order to properly quantify the real contribution of rotor yawing alone.

   The authors agree to the importance of shear (as well as veer) in the wake behaviour change, with or without misalignment. In order to isolate the effects as much as possible, the normalised quantities of interests, namely the energy and power gain in equations 2 and 3, are discussed to evaluate the contribution of the rotor yawing alone. It should also be noted that the main objective of the article is to evaluate and discuss the participating model performances under flow control, rather than analysing the field test results which is presented in Simley et al. 2021 as cited in the article, studying the same wind farm as in Blind Test #1 in the FarmConners benchmark.

5. Page 13: Including data about the site topography is important for the reader to understand better the operating environment of the wind farm. Such data can be obtained through satellite data.

   The site description provided in the beginning of Section 2, as well as the references therein that studies the same wind farm, are indeed adequate to understand the operating environment of the SMV wind farm. On the other hand, the participants were not provided detailed terrain maps or satellite data within Blind test #1. On page 13, participant P6 instead used the provided time series under the calibration data set to infer local differences on the turbine productions within the wind farm, which is due to combination of factors (terrain + spatial variability of the flow / turbulent structures, etc.). Now the statement is slightly rephrased to avoid confusion.

6. Page 13, line 5: the term "waked directions" is inappropriate.

   Now rephrased.

7. Page 14, line 285: It is understood that, following the binning process, the average wind direction was computed - was a vector average used?

   The available yaw angles were simply split into 3 subsets with the simple goal of obtaining a more yaw-specific power curve and compromise with the limited number of data points available. Hence each of these 3 subsets were represented by the average of the yaw angles for each subset, intended as scalar average. This allowed to estimate an "average" power curve for each correspondent "average" yaw setting.

8. Page 15, first para: It is helpful to present the relation for estimating the wake skew angle here. How does the formula correct for the variation of the wake skew angle with CT? Past works have shown that this follows a quasi-linear relationship.

   The formulation follows Eq. 6.12 in Bastankhah 2016 and a direct reference has been added to the text. The function expressed in this formula is non-linear, however a simple test can show this can be approximated to a linear function where "skew= a*CT+c", at least for a certain range of CT values. However, what is described in this paragraph is that previous calibration exercises carried out by NREL have suggested that a skew angle function multiplied by a factor of 2 better fits field experiments, and such an option was available in FLORIS platform.

9. Page 15, line 326: Explain what these parameters stand for.

   Now added "Those are principally the same model-parameters as described for the other participants in Table 6, described in detail in (Bastankhah and Porté-Agel, 2016), but implemented in a different framework than FLORIS."

10. Page 17, line 373: To what extent has this transfer function been validated?

    The transfer function calculated was fitted and compared with normal operation data. Below, a comparison with and without the transfer function is added for evaluation.

[Figure]

[Figure]

(a) Without the transfer function      (b) With the transfer function

11. Page 17, line 380: The numerical technique applied for filtering has to be stated.

    It is the filtered data according to the procedure described in Section 2.1 (checking if the turbines are operational at the same time, no curtailment and the wind direction sector indicated in Table 2) - now rephrased for further clarity as "The final data set within that sector for wake steering consists of 216 data points ..."

12. Page 22, Section 3.1: Details about the operating conditions of the rotor are lacking (e.g. tip speed ratio)

13. Page 22, Section 3.1: the Reynolds number matching is difficult to achieve in the wind tunnel tests given that the scale ratio is far too small. The paper should elaborate about this limitation.

14. Page 26, line 578: in yawed rotors, the disk-averaged induction at the rotor becomes a function of time. This violates the assumption for the simplistic wake losses models that are based on the linear momentum equation applied for stead flow conditions. A remark on this matter should be included.

    As stated earlier, Section 3 Blind Test #2: CL-Windcon wind tunnel is now excluded from this article (Part I of the benchmark results), and transferred to the subsequent paper (Part II of the benchmark results). Your input is to be kept in mind for that submission.

15. Page 37: The paper should explain briefly the numerical verification work undertaken to ensure that numerical errors are negligible. It should be ensured that the uncertainty arising from numerical spatial and temporal discretization does not mask the differences in power estimated with and without WFFC.

    The verification tests performed within C-Windcon project are presented in Section 3.5 in Doekemeijer et al. 2018 (`http://www.clwindcon.eu/wp-content/uploads/2017/03/CL-Windcon-D1.2-Wind-farm-models.pdf`) – now cited also under Blind Test #3 of the article.

**Minor Comments**

All the minor comments are addressed.